# Time-restricted feeding promotes muscle function through purine cycle and AMPK signaling in Drosophila obesity models

Christopher Livelo [1,5], Yiming Guo [1,5], Farah Abou Daya [1], Vasanthi Rajasekaran[1], Shweta Varshney[2,3], Hiep D. Le[2], Stephen Barnes [4], Satchidananda Panda[2] & Girish C. Melkani [1,3] ✉

Obesity caused by genetic and environmental factors can lead to compromised skeletal muscle function. Time-restricted feeding (TRF) has been shown to prevent muscle function decline from obesogenic challenges; however, its mechanism remains unclear. Here we demonstrate that TRF upregulates genes involved in glycine production (*Sardh* and *CG5955*) and utilization (*Gnmt*), while *Dgat2*, involved in triglyceride synthesis is downregulated in *Drosophila* models of diet- and genetic-induced obesity. Muscle-specific knockdown of *Gnmt*, *Sardh*, and *CG5955* lead to muscle dysfunction, ectopic lipid accumulation, and loss of TRF-mediated benefits, while knockdown of *Dgat2* retains muscle function during aging and reduces ectopic lipid accumulation. Further analyses demonstrate that TRF upregulates the purine cycle in a diet-induced obesity model and AMPK signaling-associated pathways in a genetic-induced obesity model. Overall, our data suggest that TRF improves muscle function through modulations of common and distinct pathways under different obesogenic challenges and provides potential targets for obesity treatments.

Obesity is a global and public health problem linked to various comorbidities[1]. Major contributors to obesity include living a lifestyle comprised of high-caloric diets and having a genetic predisposition to the disease[2–4]. The skeletal muscle plays a crucial role in metabolism as it is the major tissue responsible for glucose uptake from the blood[5]. Muscle dysfunction due to obesity can lead to insulin resistance and reduced energy levels[6]. Indeed, intramyocellular lipids or intramyocellular triglycerides (IMCL/IMTG) catalyzed by diacylglyceride acyltransferase 2 (DGAT2) deposited within skeletal muscle cells can be harmful if not routinely depleted as observed in athletes[7]. In addition, truncal adiposity has been associated with increased levels of S-adenosylmethionine (SAM) in overfed humans[8]. SAM is a universal methyl donor involved in various physiological processes and increased levels have been observed to be a pathogenic catalyst that

requires regulation from entities such as glycine N-methyltransferase (GNMT). GNMT converts SAM to sarcosine with the help of glycine, which can be produced via sarcosine dehydrogenase (SARDH)[9].

The primary driving force behind muscle metabolism relates to supplying energy needed for muscular contractions. Adenosine triphosphate (ATP) helps maintain muscle fiber contraction and ATP is regulated by metabolic pathways such as AMPK-signaling and the purine cycle. AMPK generally acts as a central sensor of energy status (AMP/ATP and ADP/ATP ratios) and maintains energy balance by regulating downstream anabolic and catabolic pathways[10]. In skeletal muscle, activation of AMPK has been shown to improve glucose uptake and insulin sensitivity[11] under obesogenic pressure[12], while chronic activation of AMPK increases muscle fiber oxidative capability by enhancing mitochondrial biogenesis[13]. Meanwhile, the purine cycle

[1]Department of Pathology, Division of Molecular and Cellular Pathology, Heersink School of Medicine, University of Alabama at Birmingham, Birmingham, AL 35294, USA. [2]Regulatory Biology Laboratory, Salk Institute for Biological Studies, La Jolla, CA 92037, USA. [3]Department of Biology, Molecular Biology Institute, San Diego State University, San Diego, CA 92182, USA. [4]Department of Pharmacology and Toxicology, University of Alabama at Birmingham, Birmingham, AL 35294, USA. [5]These authors contributed equally: Christopher Livelo, Yiming Guo. ✉e-mail: girishmelkani@uabmc.edu

helps maintain appropriate energy levels during exercise through ATP formation in the adenylate kinase reaction, enhancement of glycolysis, and anaplerosis through the production of fumarate[14]. Insight into muscle function, metabolism, and energy production continues to be a growing topic of interest as new studies continue to emerge[15,16].

*Drosophila melanogaster* (fruit fly) is an amenable model for studying human metabolic diseases as mechanisms associated with nutrient sensing, energy utilization, and energy storage are mostly conserved[17]. We have previously studied obese *Drosophila* by using a high-fat diet-induced obesity model (HFD) and a genetic-induced obesity model (flies lack *sphingosine kinase 2; Sk2* mutant)[18]. Both obesity models displayed skeletal muscle dysfunction, accumulation of aberrant lipids, insulin resistance as well as mitochondrial defects[18]. An intervention known as time-restricted feeding (TRF) has been shown to regulate gene expression and gene rhythmicity leading to the amelioration of obesity and metabolic dysfunction[18,19]. Imposing TRF on *Drosophila* subjected to obesogenic challenges attenuated the adverse effects of obesity shown by improved muscle performance, reduced intramuscular fat, lowered phospho-AKT levels, in addition to the reduction in a marker of insulin resistance[18]. A recent human study of 11 men with obesity in a randomized cross-over design demonstrated that short-term TRF was sufficient to modulate rhythmic metabolism of lipids, amino acids and improve nocturnal glucose levels and insulin profiles in skeletal muscle during daytime[20,21]. This study indicates that TRF is potentially impactful in managing pathologies related to metabolism and obesity while providing a natural and affordable form of alternate therapy. TRF has proved to be beneficial in various animal models of obesity shown in mouse liver, *Drosophila* heart, and muscle[18,19,22]. However, there is little information regarding the mechanistic impacts of TRF on skeletal muscle in different obesity models.

This present study investigates the mechanistic basis for TRF improvement in skeletal muscle by assessing transcriptomic data of WT, HFD, and *Sk2* models under TRF. We demonstrate that the expression levels of genes related to glycine production (*Sardh* and *CG5955*) and utilization (*Gnmt*) are upregulated under TRF in WT, HFD, and *Sk2* models. Furthermore, the expression level of a key enzyme involved in triglyceride synthesis (*Dgat2*) was downregulated in all TRF conditions. Interestingly, TRF induces upregulation in genes and increases in metabolites related to the purine cycle in HFD model. On the other hand, upregulation of genes and increases in metabolites relating to glycolysis, glycogen metabolism, tricarboxylic acid cycle (TCA), and electron transport chain (ETC) connected by AMP kinase (AMPK) signaling are observed under TRF in *Sk2* model. We further performed muscle functional assessments, cytological and biochemical assays, and metabolomic analyses to validate the pathways and their biological significance in muscle function. Taken together, this study elucidates potential mechanisms behind TRF's protective properties against skeletal muscle dysfunction and metabolic impairment induced by obesity, which may pave the way for future TRF studies in muscle.

## Results
### Common differentially expressed genes (DEGs) were identified under TRF in WT and both obesity models
To gain insight into the molecular mechanisms in skeletal muscle by which TRF protects against obesity-induced metabolic and functional changes, we examined the temporal transcriptomic profiles of indirect flight muscle (IFM) from obese flies under ALF and TRF. Feeding regimens and food type were assigned on day 4 such that flies under ALF had food access for 24 h, while flies under TRF had 12 h food access in daytime only (ZT0 – ZT12) (Fig. 1a). Wild-type *Drosophila* (Canton S) were subjected to either a regular diet (referred to as WT flies) or a high-fat diet (referred to as HFD flies) representing the diet-induced obesity model. *Sphingosine kinase 2* mutant flies were fed with a

regular diet (referred to as *Sk2* flies) representing the genetic-induced obesity model. IFMs were collected at week 3 from 10 female flies per cohort every 4 h over a 24-h period, and poly (A+) RNA-seq analysis was performed (Fig. 1b) (see Supplementary Data 1 for normalized read counts). A glossary of terms and abbreviations can be found in Supplementary Data 8.

Transcriptomic data were analyzed using two approaches. First, time-series samples were treated as replicates to identify DEGs between groups using DESeq2[23] (Supplementary Data 2–4). As TRF is known to induce moderate changes in gene expression[22], the differential gene expression threshold was set as fold change ≥1.25 and a $p$-value ≤ 0.05 in the analysis. Using these criteria, we identified 143, 270, and 408 significantly upregulated genes, and 236, 636, and 579 significantly downregulated genes under TRF versus ALF in WT, HFD, and *Sk2* IFM tissues respectively (Fig. 1c, d). Principal component analysis (PCA) revealed gene expression differences between WT and obesity models, while the effect of feeding pattern (TRF versus ALF) was moderate (Supplementary Fig. 1a).

Secondly, we used empirical JTK_CYCLE[24] to identify genes with a 24-h period cycling pattern of expression. Rhythmic genes in different groups were identified (Supplementary Fig. 1b) using criteria shown in the method section. Notably, more periodic transcripts were identified under TRF in WT and HFD, except *Sk2*. Core clock genes *Clk*, *Cyc*, *Per*, and *Tim*, were examined under TRF (Supplementary Fig. 1c). Interestingly, there were no significant expression differences between ALF and TRF for any core clock genes in the IFMs of WT and obesity models. *Clk* and *Tim* exhibited periodic oscillations under both ALF and TRF in WT, HFD, and *Sk2* IFMs. *Cyc* did not show consistent rhythmicity, which was aligned with the previous finding that *Cyc* mRNA does not cycle[25]. *Per* was arrhythmic under ALF but restored rhythmicity under TRF in HFD IFM tissue.

We examined the common DEGs and found that 5 genes (*CG6188/Gnmt*, *CG6385/Sardh*, *CG5955*, *CG6806/Lsp2, and CG5896/Grass*) were commonly upregulated and 3 genes (*CG1942/Dgat2*, *CG7997*, and *CG13992*) were commonly downregulated under TRF across WT, HFD and *Sk2* flies (Fig. 1c–g). To confirm TRF-mediated transcriptional changes (Supplementary Fig. 2a, b), we performed independent quantitative real-time PCR (qRT-PCR) analysis to validate these gene expression changes, which were found to be mostly consistent with our transcriptomic data (Supplementary Fig. 2c).

### Muscle-specific knockdown of *Gnmt*, *Sardh*, or *CG5955* leads to progressive skeletal muscle dysfunction, ectopic lipid accumulation, and abolishes TRF-mediated benefits
Three out of five common upregulated genes (*Gnmt*, *Sardh*, *and CG5955*) stood out as they were orthologous to human genes (*Gnmt*, *Sardh*, and *Tdh*), and furthermore, shared a role in glycine utilization and production (Fig. 2a, b, Supplementary Fig. 2a). Therefore, we measured glycine levels under ALF and TRF in WT, HFD, and *Sk2* models (Fig. 2c). We found that HFD-ALF flies had significantly reduced glycine levels as previously observed from HFD-induced obesity in humans[26], while *Sk2*-ALF flies had significantly increased glycine compared to WT-ALF flies. Interestingly, HFD-TRF had a modest but not statistically significant increase while *Sk2*-TRF displayed a significant increase in glycine levels compared to their respective ALF conditions (Fig. 2c). As glycine serves as an input for multiple metabolic pathways, such as glutathione synthesis and regulation of one-carbon metabolism, one of the possibilities is that TRF modulates glycine utilization differently between HFD and *Sk2* flies, masking the TRF-mediated changes on glycine levels. To examine the role of *Gnmt*, *Sardh*, and *CG5955* in skeletal muscle, we performed knockdown (KD) using UAS-RNA interference (RNAi) of *Gnmt*, *Sardh*, and *CG5955* with an IFM-specific driver (*Act88F-Gal4*)[27] and three independent RNA lines per gene. Gene KD efficiency was assessed using qRT-PCR (Supplementary Fig. 3a). We measured muscle performance in 1-, 3- and 5-week-old

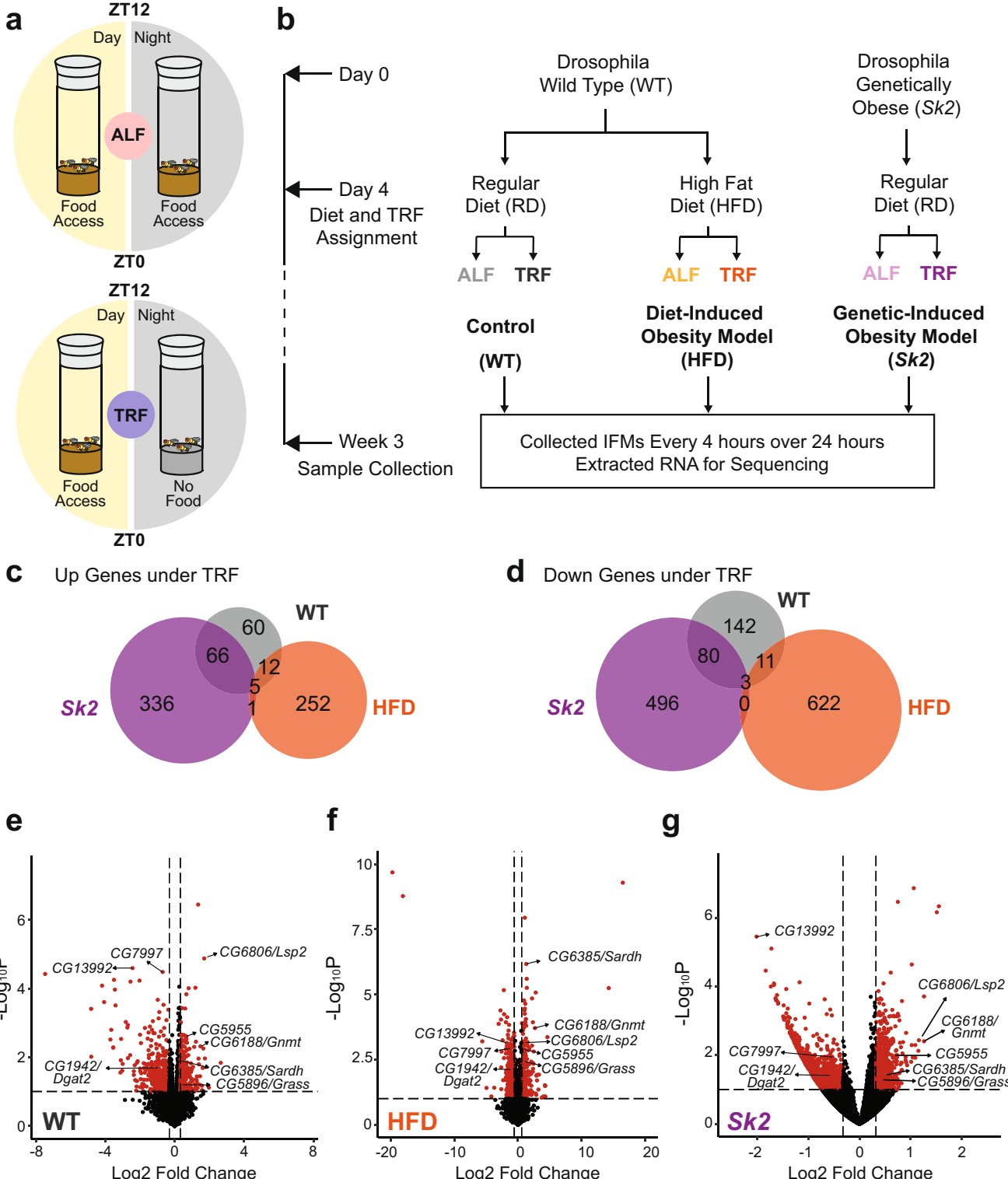

**Fig. 1 | Common differentially expressed genes identified under TRF versus ALF in WT, HFD, and *Sk2* flies. a** Schematic depicting the timing of food access in ALF (*ad libitum* feeding) and TRF (time-restricted feeding). In ALF, flies have unrestricted access to food during the day (yellow) and night (gray). In TRF, flies consume food only during the 12 h of daytime. **b** Flow chart depicting the experimental setup. Wild-type and genetically obese mutant *Sk2* flies were separated into respective food types either a regular diet (RD) or high-fat diet (HFD) to establish WT, HFD, and *Sk2* conditions. Flies were further separated into either ALF or TRF. IFMs were collected at 3 weeks of age every 4 h totaling 6-time points for RNA sequencing. **c, d** Venn diagram of significantly upregulated genes (**c**) and significantly downregulated genes (**d**) performed using DeSeq2 (*p*-value ≤ 0.05, fold change ≥1.25) under TRF versus ALF in WT, HFD, and *Sk2* models. **e–g** Volcano plots showing the expression profiles of the common DEGs under TRF versus ALF in WT (**e**), HFD (**f**), and *Sk2* (**g**) flies. Red dots represent differentially expressed genes under TRF (*p*-value ≤ 0.05, Fold change ≥1.25). Arrows indicate the common DEGs under TRF.

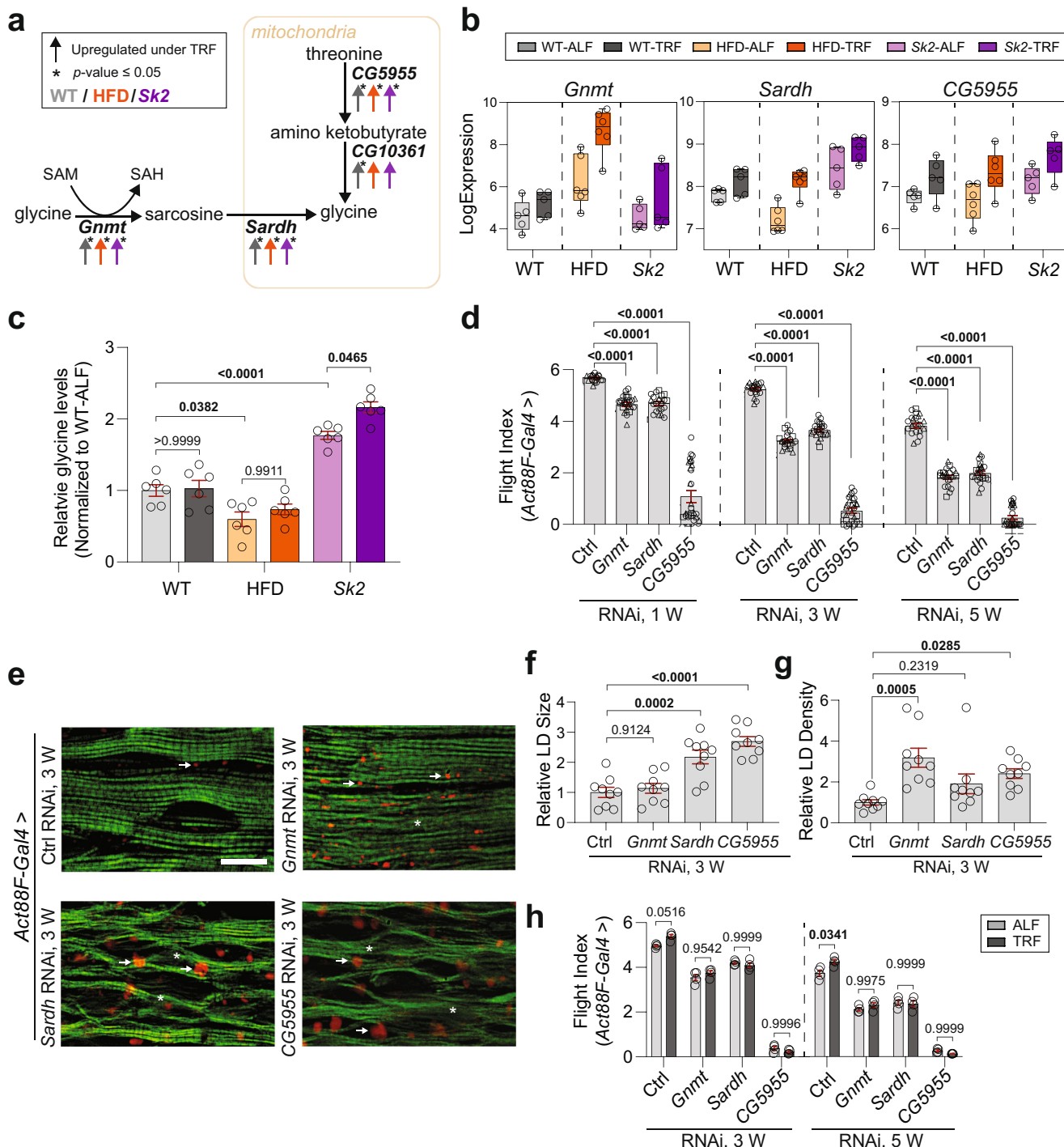

female flies (Fig. 2d, Supplementary Fig. 3b, c). Reduced flight performance was detected and progressed with aging during IFM-specific *Gnmt*, *Sardh*, and *CG5955* suppression in female flies, while a greater reduction was observed in male flies (Supplementary Fig. 3d). For the rest of the study, we only focused on the female progenies when using *Act88F-Gal4* driver.

In addition to muscle dysfunction, IFMs from 3-week-old female flies upon IFM-specific KD of *Sardh* and *CG5955* showed a significant increase of aberrant lipid accumulation (both size and density, Nile red staining, see arrow) (Fig. 2e). Furthermore, IFM-specific KD of *Sardh* and *CG5955* resulted in disorganization of actin-containing myofibrils (Phalloidin staining, see asterisk) compared to age-matched control myofibrils (Fig. 2e). IFM-specific KD of *Gnmt*

exhibited only a moderate increase in lipid droplets size; however, the lipid density was significantly enhanced, compared to age-matched control tissue (Fig. 2e–g). Overall, we observed a reduction in muscle performance and an increase in ectopic lipid deposition and myofibril disorganization upon IFM-specific KD of *Gnmt*, *Sardh*, and *CG5955*. Abdomen lipid staining was also performed in the fat body and no significant differences were observed (Supplementary Fig. 5a–c). To test whether *Gnmt*, *Sardh*, and *CG5955* contribute to TRF-mediated muscle improvement, we tested the flight ability of 3-, and 5-week-old female flies with IFM-specific KD of *Gnmt*, *Sardh*, and *CG5955* under ALF and TRF. It was found that flight indices upon IFM-specific KD of *Gnmt*, *Sardh*, and *CG5955* failed to improve under TRF (Fig. 2h).

**Fig. 2 | Functional and cytological assessments of *Gnmt*, *Sardh*, and *CG5955* KD using *Act88F-Gal4*. a** Schematic representation of *Gnmt*, *Sardh*, and *CG5955* connection with metabolite intermediates and important genes (italic bold) encoding enzymes. The color indicates the condition (gray WT, orange HFD, purple *Sk2*). Up arrows indicate upregulation of gene expression under TRF versus ALF (fold change ≥1.25). Asterisks indicate *p*-value ≤ 0.05. **b** Expression level of gene *Gnmt*, *Sardh*, and *CG5955* under ALF and TRF in WT, HFD, and *Sk2* models. $N = 5$ time points in WT and *Sk2*. $N = 6$ time points in HFD. **c** Relative glycine levels under ALF and TRF in 3-week-old female WT, HFD, and *Sk2* thoraces. $N = 6$ biologically independent replicates. Mean ± SEM. Two-way ANOVA with Sidak post hoc tests. **d** Flight performance of female flies with *Act88F-Gal4*-driven KD of *Gnmt*, *Sardh*, or *CG5955* at 1, 3, and 5 weeks of age. Two independent control RNAi lines were used (Ctrl). Three independent RNAi lines per gene were tested. Flight indices plotted as cohorts of 10–20 flies from independent RNAi lines were indicated with the symbol circle, triangle, or square. # of cohorts per RNAi line per age group = 7–9; # of flies

per RNAi line per age group = 100–170. Details of precise *N* are in Source Data. Mean ± SEM. One-way ANOVA with Sidak post hoc tests. **e** Fluorescence images of the cryosectioned IFMs from 3-week-old female flies with *Act88F-Gal4*-driven KD of *Gnmt*, *Sardh*, and *CG5955* upon staining with phalloidin (green) and Nile Red (red puncta). More accumulation of lipid (arrows) and actin-containing myofibrillar disorganization (asterisks) was detected in IFMs of KD of *Gnmt*, *Sardh*, and *CG5955*, compared with age-matched control. The scale bar is 20 μm. **f, g** Intramuscular lipid quantification (lipid droplet size (**f**) and density (**g**)). $N = 9$ from three flies' IFM per genotype. Mean ± SEM. One-way ANOVA with Sidak post hoc tests. **h** Flight performance of 3- and 5-week-old female flies with *Act88F-Gal4*-driven KD of *Gnmt*, *Sardh*, and *CG5955* under ALF and TRF. As the three independent lines showed similar results in **d**, one RNAi line per gene was used. Flight indices were plotted as cohorts of 10–20 flies. # of cohorts per age group = 4; # of flies per age group = 53–83. Details of precise *N* are in Source Data. Mean ± SEM. Two-way ANOVA with Sidak post hoc tests. Source data are provided as a Source Data file.

In order to examine the role of *Gnmt*, *Sardh*, and *CG5955* in IFM during the adult phase, we utilized a *DJ694-Gal4* driver, an adult-muscle driver that initiates upon eclosion and remains active during the whole adult life span[27–30]. KD of *Gnmt*, *Sardh*, and *CG5955* using *DJ694* driver was carried out and KD efficiency in IFM was validated using qRT-PCR (Supplementary Fig. 3e). We measured muscle performance at 4-day- and 3-week-old female and male flies. While modest, or no significant differences in flight index were observed on day 4, a significant decline in flight performance was observed from KD of all three genes in 3-week-old female flies (Fig. 3a and Supplementary Fig. 3f), suggesting the important roles of *Gnmt*, *Sardh*, and *CG5955* in muscle function and maintenance. We also tested IFM-specific KD of all three genes in the adults using the *Act88F-GS-Gal4* driver[31]. Similar muscle functional decline was found in 3-week-old female flies (Supplementary Fig. 3g). Notably, less severe muscle function decline from *CG5955* KD using *DJ694-Gal4* or *Act88F-GS-Gal4* compared to *Act88F-Gal4*, suggesting a potential role of *CG5955* during the developmental phase. IFM upon *Gnmt*, *Sardh*, and *CG5955* KD using *DJ694* driver were examined in 3-week-old female flies. Aberrant lipid accumulation and myofibrillar disorganization were observed (Fig. 3b–d) as previously shown using *Act88F* driver (Fig. 2e–g). Abdomen lipid staining was also performed in the fat body; no significant differences were observed (Supplementary Fig. 5g–i). To assess the roles of *Gnmt*, *Sardh*, and *CG5955* on TRF-mediated benefits in muscle during the adult stage, we tested the flight performance upon KD of those genes using *DJ694* under ALF and TRF. Consistent with results from *Act88F* driver, TRF-mediated muscle improvements were abrogated (Fig. 3e), although mild expression increases were detected upon TRF within the corresponding KD (Supplementary Fig. 3h).

Furthermore, we overexpressed *Gnmt*[32] using *Act88F-Gal4* driver to examine whether muscle function could be retained. Interestingly, overexpression of *Gnmt* preserved flight performance in 5-week-old female flies (Supplementary Fig. 3i), with modestly reduced lipid droplet size in IFMs (Supplementary Fig. 3j, k). No significant changes were observed in the abdomen fat body Nile red staining (Supplementary Fig. 5a–c). We also used *DJ694* and *Act88F-GS* with titrated doses of RU486 to perform overexpression of *Gnmt* in the adult phase. Both *DJ694* and *Act88F-GS* drivers increased expression levels of *Gnmt* that are comparable with TRF-induced *Gnmt* upregulation in the IFM (Fig. 3f and Supplementary Fig. 3l). Flight performance of 3-week-old female flies improved, especially when feeding with HFD (Fig. 3g and Supplementary Fig. 3m, n). Previous studies have shown two possible upstream transcription factors, which modulate GNMT activities: cAMP-regulated transcription coactivator (CRTC) and Forkhead Box O (FoxO)[33,34]. Interestingly, *Crtc* expression levels were slightly higher under TRF in WT and HFD flies, while *FoxO* expression level was higher under TRF in *Sk2* flies (Supplementary Fig. 3o), possibly suggesting obese-specific mediation of *Gnmt* expression under TRF. Taken together, TRF-mediated upregulation of *Gnmt*, *Sardh*, and *CG5955* may

account for at least a part of the beneficial effect of TRF in skeletal muscle.

## Muscle-specific knockdown of *Dgat2* retains age-dependent decline of skeletal muscle function

DGAT2, known as one of the two acyl-CoA: diacylglycerol O-acyltransferase enzymes, catalyzes the final step in de novo TG synthesis (Fig. 4a)[35,36]. From our transcriptomic data, the expression levels of *Dgat2/CG1942* were downregulated under TRF in WT and obesity models (Fig. 4b and Supplementary Fig. 2a–b). We also examined the expression of upstream enzymes associated with lipid droplet de novo TG synthesis[35,36]. Interestingly, *Gpat4* and *Agpat3* also showed decreased but not significant expression levels under TRF (Fig. 4a, b). To investigate the role of *Dgat2* in skeletal muscle function, we performed *Dgat2* KD using *Act88F-Gal4* driver and a UAS-RNAi line with KD efficiency validated (Supplementary Fig. 4a). We measured muscle performance in 1-, 3-, 5-, and 7-week-old female flies. Interestingly, IFM-specific suppression of *Dgat2* significantly improved flight ability with age when compared to age-matched control (Fig. 4c). Lipid staining in IFMs from *Act88F*-driven KD of *Dgat2* was examined at 3-week-old female flies (Fig. 4d). A significant reduction of lipid droplet size in IFM was observed in *Dgat2* KD flies compared to the control while no change in lipid density was observed (Fig. 4e, f). Abdomen lipid staining was also performed in the fat body and no significant differences were observed (Supplementary Fig. 5a–c). We also examined muscle performance and muscle cytology of flies with *DJ694*-driven *Dgat2* KD with KD efficiency validated (Fig. 4g–j and Supplementary Fig. 4b). Similarly, improved flight performance (Fig. 4g) and less lipid droplet were observed with no change to density compared to the age-matched control (Fig. 4h–j). Abdomen lipid staining was also performed in the fat body; no significant differences were observed (Supplementary Fig. 5g–i).

To explore the contribution of *Dgat2* downregulation to TRF-mediated muscle benefits, we performed TRF on female flies upon *DJ694*-driven *Dgat2* KD. Interestingly, 5-week-old flight performance was further improved under HFD-TRF upon *Dgat2* KD (Supplementary Fig. 4c). An additional reduction of *Dgat2* expression levels was observed under TRF while suppressing *Dgat2* via RNAi (Supplementary Fig. 4d). To further confirm the role of *Dgat2* in muscle, we overexpressed human *Dgat2* (h*Dgat2*) in IFM, and as expected, a decline on muscle performance was seen in 3-week-old flies (Supplementary Fig. 4e) with an increase on lipid droplet density in IFM even at 1 week of age (Supplementary Fig. 4f, g) but not in the abdominal tissue (Supplementary Fig. 5d–f). We also performed TRF on female flies upon *DJ694*-driven h*Dgat2* OE. 3-week-old flight performance was improved under RD-TRF and HFD-TRF upon h*Dgat2* OE (Supplementary Fig. 4h). It could potentially be explained by the TRF-mediated reduction of endogenous *Dgat2* levels (Supplementary Fig. 4i), which could potentially counter the increased levels of h*Dgat2*.

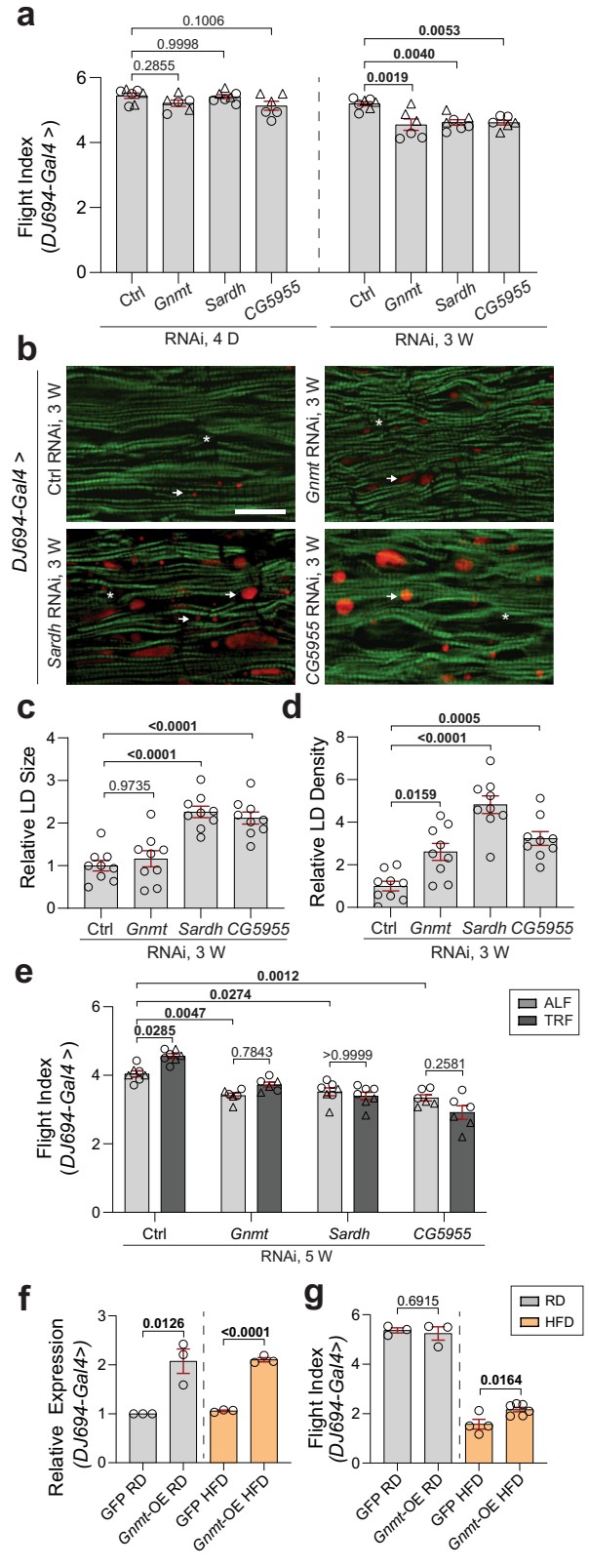

**a**

**b**

**c**

**d**

**e**

**f**

**g**

**Fig. 3 | Functional and cytological validation of *Gnmt*, *Sardh*, and *CG5955* KD using *DJ694-Gal4*. a** Flight performance of female flies with *DJ694-Gal4*-driven KD of *Gnmt*, *Sardh*, or *CG5955* at 4 days and 3 weeks of age. Two independent RNAi lines were used for control and each gene. Flight indices plotted as cohorts of 10–20 flies from independent RNAi lines were indicated with the symbol circle or triangle. # of cohorts per RNAi line per condition = 3; # of flies per genotype per age group = 86–130. Details of precise *N* are in Source Data. Mean ± SEM. One-way ANOVA with Sidak post hoc tests. **b** Fluorescence images of the IFMs from 3-week-old female flies with *DJ694-Gal4*-driven KD of *Gnmt*, *Sardh*, and *CG5955* upon staining with phalloidin (green) and Nile Red (red puncta). More accumulation of lipid (arrows) and actin-containing myofibrillar disorganization (asterisks) was detected in IFMs upon KD of *Gnmt*, *Sardh*, and *CG5955*, compared with age-matched control. The scale bar is 20 μm. **c**, **d** Intramuscular lipid quantification (lipid droplet size (**c**) and density (**d**)). *N* = 9 from three flies' IFM per genotype. Mean ± SEM. One-way ANOVA with Sidak post hoc tests. **e** Flight performance of 5-week-old female flies with *DJ694-Gal4*-driven KD of *Gnmt*, *Sardh*, and *CG5955* under ALF and TRF. Two independent lines were tested. Flight indices were plotted as cohorts of 10–20 flies. Results from independent RNAi lines were indicated with symbol circle or triangle. # of cohorts per RNAi line per condition = 3–4; # of flies per genotype = 67–98. Details of precise *N* are in Source Data. Mean ± SEM. Two-way ANOVA with Sidak post hoc tests. **f** Relative expression of *Gnmt* in 3-week-old female flies with *Gnmt* overexpression using *DJ694* driver. Experimental flies were fed with RD or HFD. *N* = 3 biological replicates. Mean ± SEM. Two-sided unpaired *t*-tests. **g** Flight performance in 3-week-old female flies with *Gnmt* overexpression driven by *DJ694*. Experimental flies were fed with RD or HFD. Flight indices were plotted as cohorts of 10–20 flies. # of cohorts per condition = 3–5; # of flies per genotype per age = 40–100. Details of precise *N* are in Source Data. Mean ± SEM. Two-sided unpaired *t*-tests. Source data are provided as a Source Data file.

important role in muscle function and lipid regulation. *CG7997*, another downregulated gene, is orthologous to human *Gla* (galactosidase alpha) (Supplementary Fig. 4l). However, little is known about *CG7997* in *Drosophila*. We tested muscle performance on flies with IFM-specific KD of *Gla* using *Act88F* driver; flight ability was also significantly improved with aging (Supplementary Fig. 4m). Further studies will be needed to understand the role of *CG7997* in maintaining muscle function.

## TRF increases folate regulation and purine cycle in the HFD model

We examined gene expression changes by performing Gene Ontology and Reactome pathway enrichment analysis for HFD and *Sk2* DEGs separately. Genes that were significantly upregulated under TRF versus ALF in HFD were enriched in immune response, proteolysis regulation, purine cycle, and amino acid metabolism (Fig. 5a, b), while downregulated genes were involved in neuronal signaling (Supplementary Fig. 6a, b). As most of the hits were related to the purine cycle, we examined the gene expression changes within the folate cycle and purine cycle (Fig. 5c). The purine cycle involves a 10-reaction 6-enzyme process that generates IMP from PRPP[37]. Strikingly, mRNA expression levels of all 6 genes (*Prat2*, *Gart*, *Pfas*, *Paics*, *AdSL*, and *Atic*) encoding enzymes were significantly upregulated under TRF in HFD (Fig. 5c, d, and Supplementary Fig. 7a). *Prps*, located upstream of de novo IMP biosynthesis, as well as *Nmdmc* and *Pug* within the folate cycle, were significantly upregulated (Fig. 5c, d). As shown in Supplementary Fig. 7b, 8 genes were rhythmic over 24 h under TRF; among them, 7 genes oscillated in both ALF and TRF with higher amplitudes under TRF, while *Atic* gained rhythmicity under TRF. All cycling genes reached the peak expression at the end of the eating period. Notably, non-significant increases in the expression of these genes under TRF versus ALF were observed in WT and *Sk2* flies, although a similar peak pattern was shown in *Sk2* flies (Supplementary Fig. 7b).

As glycine is a key amino acid utilized in the purine cycle, TRF-induced upregulation of the purine cycle could potentially mask the increased glycine generated by upregulated *Sardh* and *CG5955* (Fig. 2c). To evaluate this possibility, we suppressed *Gart*, a gene

We also examined the expression levels of *Dgat2* paralogs *CG1941* and *CG1946*. Slight reductions in expression levels were observed under TRF versus ALF, except *CG1941* in HFD (Supplementary Fig. 4j). *CG1941* KD driven by *DJ694-Gal4* improved flight performance at week 7, while no significant improvement was observed upon *CG1946* KD (Supplementary Fig. 4k). Overall, using loss-of-function and gain-of-function validation approaches, our data suggest that *Dgat2* plays an

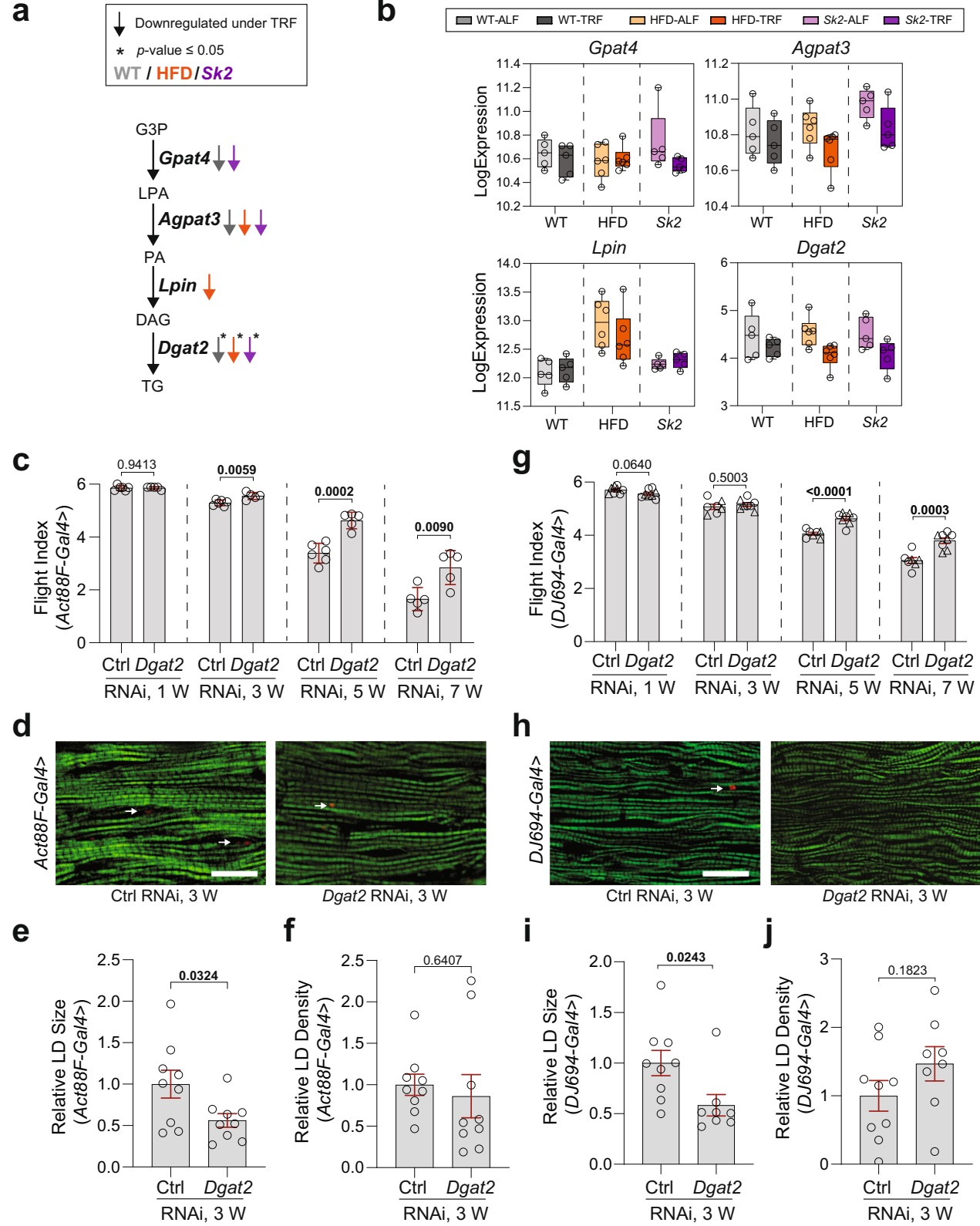

encoding enzyme which incorporates glycine in the purine cycle and examined glycine levels under TRF in HFD (Supplementary Fig. 7c). Interestingly, *Gart* suppression led to a significant increase in glycine level, supporting that the non-significant increase of glycine levels could be potentially due to the elevation of glycine utilization under HFD-TRF. We attempted to conduct *Gart* and *Sk2* double KD to examine whether additional increase of glycine level can be detected.

Unfortunately, *Sk2* knockdown efficiency was only ~25% and couldn't recapitulate the increased basal level of glycine in *Sk2* null mutant.

To understand the roles of genes involved in folate cycle and purine cycle on skeletal muscle function, we performed KD of *Nmdmc* and *AdSL* using *Act88F* and *DJ694* drivers with two RNAi lines per gene. While *Act88F*-driven KD of *Nmdmc* or *AdSL* showed reduced flight performance which was progressively worsened with age (Fig. 5e and

**Fig. 4 | Gene expression and functional validation of *Dgat2* in skeletal muscle in WT and obesity models. a** Schematic representation of de novo triacylglycerol synthesis with metabolite intermediates and important genes (italic bold) encoding enzymes. Color indicates condition (gray WT, orange HFD, purple *Sk2*). Down arrows indicate downregulation of gene expression (fold change ≥1.25). Asterisks indicate *p*-value ≤ 0.05. **b** Expression level of gene *Dgat2* under ALF and TRF in WT and obese models. *N* = 5 time points in WT and *Sk2*. *N* = 6 time points in HFD. **c** Flight performance of female flies with *Act88F-Gal4*-driven *Dgat2* KD at 1, 3, 5, and 7 weeks of age. One RNAi line was tested. Flight indices were plotted as cohorts of 10–20 flies. # of cohorts per RNAi line per age group = 5–6; # of flies per genotype per age = 65–102. Details of precise *N* are in Source Data. Mean ± SEM. Two-sided unpaired *t*-tests. **d** Fluorescence images of the IFMs from 3-week-old female flies with *Act88F-Gal4*-driven KD of *Dgat2* upon staining with phalloidin (green) and Nile Red (red puncta). Smaller sizes of lipids (arrows) were detected in IFMs of flies with

*Dgat2* KD, compared with age-matched control. The scale bar is 20 μm. **e**, **f** Intramuscular lipid quantification of 3-week-old females (lipid droplet size (**e**) and density (**f**)). *N* = 9 from three flies' IFM per genotype. Mean ± SEM. Two-sided unpaired *t*-tests. **g** Flight performance of female flies with *DJ694-Gal4*-driven *Dgat2* KD at 1, 3, 5, and 7 weeks of age. Two independent fly lines were tested. Flight indices were plotted as cohorts of 10–20 flies. Results from independent RNAi lines were indicated with symbol circle or triangle. # of cohorts per RNAi line per condition = 3–4; # of flies per genotype per age group = 95–137. Details of precise *N* are in Source Data. Mean ± SEM. Two-sided unpaired *t*-tests. **h** Fluorescence images of the IFMs from 3-week-old female flies with *DJ694-Gal4*-driven *Dgat2* KD upon staining with phalloidin (green) and Nile Red (red puncta). The scale bar is 20 μm. **i**, **j** Intramuscular lipid quantification of 3-week-old females (lipid droplet size (**i**) and density (**j**)). *N* = 9 from three flies' IFM per genotype. Mean ± SEM. Two-sided unpaired *t*-tests. Source data are provided as a Source Data file.

Supplementary Fig. 7d, e), *DJ694*-driven KD only had moderate performance reduction but failed to show improved flight ability under TRF (Supplementary Fig. 7f). We also subjected *Act88F*-driven *Nmdmc* or *AdSL* KD flies into HFD-ALF or HFD-TRF regimens, and no improvements were observed under TRF (Supplementary Fig. 7g).

To further validate whether TRF activates purine cycle, we performed an untargeted metabolome analysis on 3-week-old HFD-ALF and HFD-TRF IFMs. Adenine nucleotides (ATP, ADP, AMP) are essential for providing the energy necessary for muscular contracture[38] and accumulation of ATP degradation products (inosine, hypoxanthine xanthine, and uric acid) have been observed under atrophic muscle conditions[38]. Interestingly, we observed hypoxanthine, xanthine, and inosine were reduced under HFD-TRF (Fig. 5f and Supplementary Fig. 10). Meanwhile, ADP, in addition to fumarate, a TCA entry point, and its product malic acid, were increased under HFD-TRF (see Supplementary Fig. 9 for the differential analysis of metabolites in WT-ALF and WT-TRF). As the purine cycle plays an important role in energy balance, ATP levels were measured from the thoraces of 3-week-old HFD female flies under ALF and TRF. The relative ATP levels increased by 20% under TRF compared to ALF in the HFD flies, while no changes were observed in WT flies (Supplementary Fig. 7h).

Folic acid, also known as vitamin B9, can be reduced to THF, and then enter the folate cycle in support of the purine cycle. Given the significant upregulation of *Gnmt*, *Sardh*, *CG5955*, and genes associated with purine synthesis under HFD-TRF, we tested the impact of folic acid supplementation on muscle function under a high-fat diet challenge. A dose-dependent effect of folic acid on muscle performance is shown in Supplementary Fig. 7i, j, where a dose higher than 4 mM is deleterious. Therefore, a concentration of 4 mM was used in the following folic acid supplementation. In addition, we tested the effects of folic acid supplementation in WT and *Sk2* flies; however, no improvements in flight and climbing abilities were found (Supplementary Fig. 7k, l). As folic acid supplementation improved muscle performance in HFD flies, the effects of folic acid supplementation in HFD were also examined in flies with *Act88F*-driven KD of *Gnmt* and *Nmdmc*. Interestingly, a significant improvement in flight performance was observed in *Nmdmc* KD flies under a high-fat diet supplemented with folic acid, but not in flies with *Gnmt* KD (Fig. 5g). Regarding ATP levels, we detected a significant increase in ATP in both *Gnmt* KD and *Nmdmc* KD HFD flies supplemented with folic acid (Fig. 5h). While we have not directly measured the effects of folic acid feeding on metabolites, we have shown increases in metabolites related to purine cycle under HFD-TRF. Metabolites such as oxypurines (hypoxanthine, xanthine) were decreased in HFD-TRF. It is important to note that decreased oxypurines were also observed from folic acid treated ischemic patients in another study[39]. Furthermore, an in silico study predicted that folate depletion leads to a reduction of ATP pools[40]. Altogether, our results suggest that TRF modulates purine cycle in HFD IFMs and

this in turn ameliorates obesity-related muscle declines via increased ATP levels.

## TRF upregulated genes linked with AMPK and downstream pathways in *Sk2* mutant, a genetic obesity model

Based on the GO and Reactome analysis, genes that were significantly upregulated under TRF in *Sk2* flies were enriched in four major groups: (1) Glycolysis; (2) Glycogen metabolism; (3) TCA cycle; (4) Mitochondrial ETC (Fig. 6a, b). In addition, GO and Reactome analyses revealed significant downregulation of genes involved in cell division, DNA replication, and DNA repair (Supplementary Fig. 6c, d). Interestingly, these four metabolic pathways enriched by upregulated genes under TRF can be connected by the activation of AMPK (Fig. 6c, d). Moreover, the mRNA levels of the AMPKα subunit and genes associated with the AMPK downstream pathways were significantly increased under TRF in *Sk2* flies; while, this trend was not consistently seen in WT or HFD flies (Supplementary Fig. 8a–c). Previous studies have revealed rhythmicity in gene expression for key enzymes of glycolysis in mice skeletal muscle[41], and glycogen metabolism in rat liver[42]. Therefore, we examined whether the significantly upregulated genes under TRF also have rhythmic expression in *Sk2* flies. As expected, most genes encoding enzymes in glycolysis and glycogen metabolism had rhythmicity under TRF (Supplementary Fig. 8d), where *Pgi*, *Pgk*, and *Eno* gained rhythmicity under *Sk2*-TRF compared to *SK2*-ALF. Elevated amplitudes were detected under TRF among genes that oscillate in both ALF and TRF. On the other hand, most of the genes associated with TCA cycle and ETC did not show rhythmicity, aligning with the previous observation where the TCA cycle and ETC components exhibited circadian changes in protein but not mRNA levels[25,43]. Notably, expression levels of the upregulated genes associated with AMPK signaling downstream pathways were unchanged, if not decreased, under TRF in WT and HFD flies (Supplementary Fig. 8d, e).

To understand the role of genes associated with glycolysis, glycogen metabolism, TCA, and ETC on skeletal muscle function, we performed KD of selected genes using the *Act88F* and *DJ694* drivers. While significant reductions of flight performance in 1- and 3-week-old flies were observed upon *Act88F*- driven KD of *Ampkα*, *GlyS*, *Pfk*, *Ald1*, *mAcon1*, *Ogdh*, *SdhD* and *Sicily* (Fig. 6e, f, and Supplementary Fig. 8f, g), only moderate performance reduction were observed in 5-week-old flies upon *DJ694*-driven KD of *Ampkα* and *SdhD* (Supplementary Fig. 8h). It is to note that *DJ694*-driven *mAcon1* and *Ogdh* KD showed flight performance impairment, suggesting their potential roles in muscle maintenance. We subjected these *DJ694*-driven KD flies to ALF and TRF regimens. TRF failed to improve muscle performance upon suppression of *Ampkα*, *Ogdh*, and *SdhD*, but not *mAcon1* (Supplementary Fig. 8h).

To assess the *Sk2*-specific roles of AMPK downstream pathways, we tested the flight performance of flies with *Act88F*-driven *GlyS* or *mAcon1* KD accompanied with *Sk2* RNAi. While *mAcon1* KD itself led to flight performance reduction, *mAcon1* and *Sk2* double KD caused a

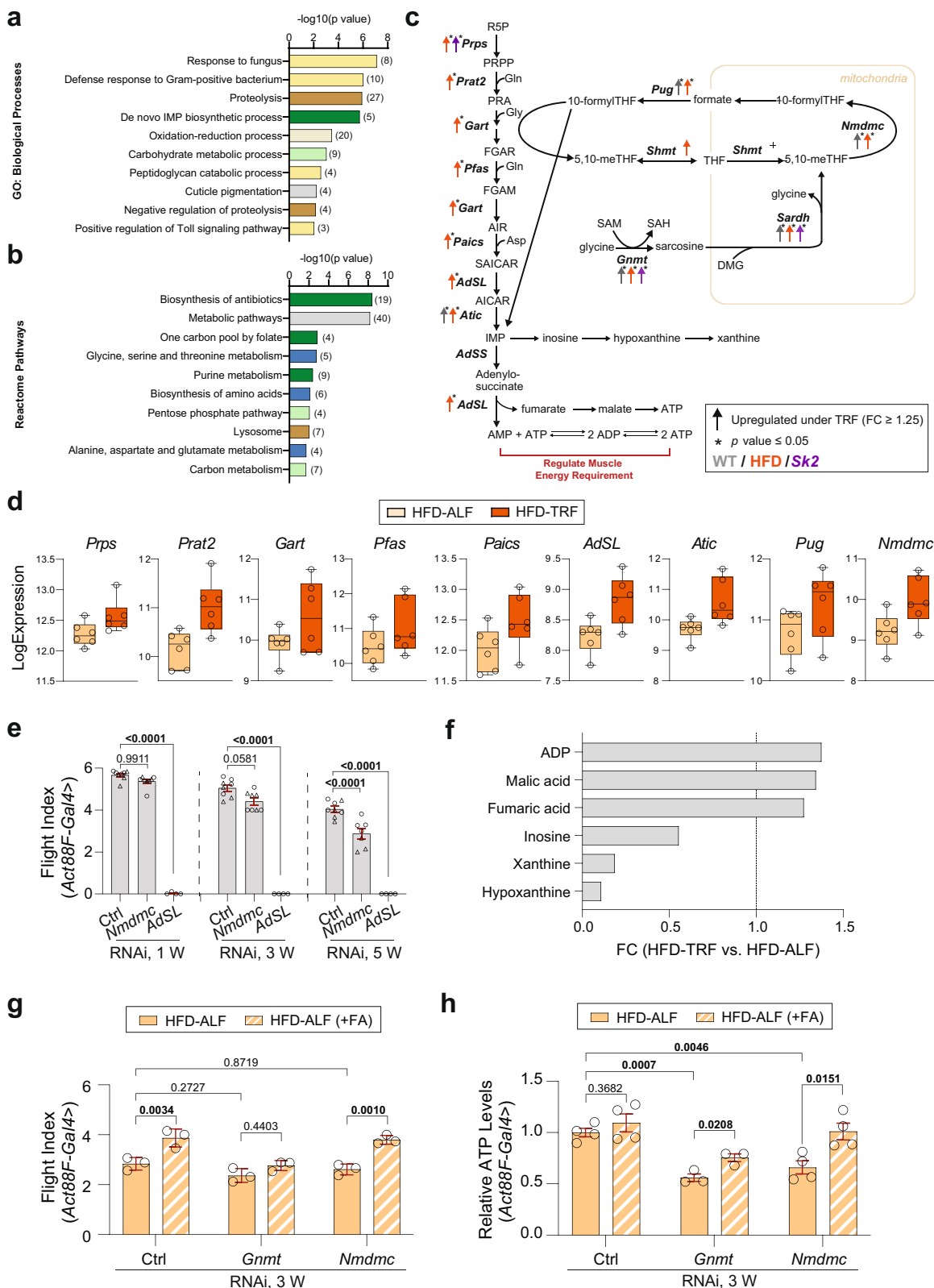

further reduction in flight index (Supplementary Fig. 8i). A similar trend was also observed in *GlyS* KD. To examine whether AMPK-associated pathways contribute to TRF-mediated muscle improvement in *Sk2* flies, we tested the flight performance upon *GlyS* or *mAcon1* KD accompanied with *Sk2* KD. While TRF-mediated muscle improvements were observed in *mAcon1* KD flies, TRF failed to improve flight ability in the suppression of *mAcon1* accompanied with

*Sk2* KD. Moreover, while *GlyS* KD flies did not respond to TRF, *GlyS,* and *Sk2* double KD flies showed a further decline in muscle performance under TRF (Supplementary Fig. 8i). Altogether, these results suggest that basal or increased activity of AMPK-associated pathways supports TRF-mediated muscle benefits in *Sk2* flies.

To further test if TRF activates AMPK signaling in *Sk2* flies, metabolites were analyzed under ALF and TRF. We observed di- and tri-

**Fig. 5 | TRF upregulated genes linked with purine biosynthesis and folate cycle in the HFD model. a** GO analysis and **b** Reactome pathway analysis of 270 genes that were significantly upregulated under TRF in HFD flies. Bar charts represent the −log10 (p-value) of each enriched GO term and pathway. Related pathways are colored similarly. The number of genes identified in each GO term and pathway is shown in parentheses. **c** Schematic representation of purine cycle and folate cycle with metabolic intermediates and important genes (italic bold) encoding enzymes possibly involved in energy regulation in muscle. The color indicates condition (gray WT, orange HF, purple *Sk2*). Up arrows indicate upregulation of gene expression (fold change ≥1.25). Asterisks indicate p-value ≤ 0.05. **d** Expression level of significantly upregulated genes in the purine cycle and folate cycle under TRF versus ALF in the HFD model. $N = 5$ time points in WT and *Sk2*. $N = 6$ time points in HFD. **e** Flight performance of control, *Nmdmc* KD, and *AdSL* KD using *Act88F-Gal4* at 1, 3, and 5 weeks of age in female flies. Two independent lines were tested. Results from *AdSL* RNAi #1 are shown in Fig. 5e and results from *AdSL* RNAi #2 are shown separately in Supplementary Fig. 7e due to varying severity of phenotypes and

different KD efficiency verified by qRT-PCR (Supplementary Fig. 7d). Flight indices were indicated as cohorts of 10–20 flies. Results from independent RNAi lines were indicated with the symbol circle or triangle. # of cohorts per RNAi line per age group = 4; # of flies per genotype per age group = 51–143. Details of precise N are in Source Data. Mean ± SEM. One-way ANOVA with Sidak post hoc tests. **f** Metabolites (fold change ≥1.25 under HFD-TRF versus HFD-ALF) associated with the purine cycle. $N = 3$ biologically independent replicates. **g** Flight performance of 3-week-old female flies with *Act88F-Gal4*-driven *Gnmt* KD and *Nmdmc* KD under HFD-ALF and HFD-ALF(+FA). FA, folic acid. # of cohorts per genotype per diet = 3; # of flies per genotype per diet = 44–56. Details of precise $N$ are in Source Data. Mean ± SEM. Two-way ANOVA with Sidak post hoc tests. **h** Relative ATP level measurement in 3-week-old female flies with *Act88F-Gal4*-driven *Gnmt* KD and *Nmdmc* KD under HFD-ALF and HFD-ALF(+FA). $N = 3$–4 biologically independent replicates per genotype per diet. Mean ± SEM. Two-sided unpaired t-tests. Source data are provided as a Source Data file.

saccharides melezitose and melibiose reduced under *Sk2*-TRF suggesting higher rates of metabolism of glycogen (Fig. 6g). TCA components citrate and malate, in addition to L-carnitine, acetylcarnitine and glutamine, which are essential for TCA[44], were found to be increased under *Sk2*-TRF condition. Furthermore, beta-nicotinamide adenine dinucleotide (β-NAD⁺) and nicotinamide (NADH), which are co-factors essential for ETC, were found to be increased under *Sk2*-TRF (Fig. 6g and Supplementary Fig. 11). These results support the TRF-mediated involvement of glycogen metabolism, TCA, and ETC pathways, which are well-known pathways in regulating ATP levels[45].

To determine whether AMPK contributes to TRF-dependent skeletal muscle benefits, we examined the expression of *Ampkα* (gene encoding the catalytic subunit of AMPK in *Drosophila*[46]) (Supplementary Fig. 8a, b), as genes encoding the beta and gamma subunits were not found in our transcriptomic data. While AMPKα protein levels were unchanged under TRF compared to the ALF counterparts (Supplementary Fig. 8j, k), increased p-AMPKα level was observed under *Sk2*-TRF versus *Sk2*-ALF (Fig. 6h, i, and Supplementary Fig. 8l), suggesting the TRF-mediated activation of AMPK signaling, especially in *Sk2* flies. We further evaluated p-AMPKα levels at different time points across the day in *Sk2* flies. Interestingly, a phase shift of p-AMPKα was observed under *SK2*-TRF compared to *Sk2*-ALF, which peaks at the end of the eating period (Fig. 6j, k). As AMPK is known as an energy sensor, ATP levels were also measured under ALF and TRF in 3-week-old *Sk2* female flies. The relative ATP levels were reduced by 32% in *Sk2*-ALF compared to WT-ALF, and as expected, TRF restored relative ATP levels by 20% compared to ALF in the *Sk2* flies (Supplementary Fig. 7h). Taken together, our results suggest that TRF activates AMPK signaling and its downstream pathways through increased gene expression in an *Sk2*-specific manner. Consequently, this change in gene expression leads to increased ATP levels and in turn helps ameliorate obesity-related muscle phenotypes.

## Discussion

The prevalence of obesity continues to be a worldwide growing issue associated with crippling healthcare and economic burdens[47]. The skeletal muscle plays a primary role in energy and protein metabolism, glucose uptake and storage, and essential daily physiological tasks such as breathing and locomotion[48]. Interestingly, studies have demonstrated that TRF, a natural non-pharmaceutical intervention, protects against obesity, aging, and circadian disruption in peripheral tissues such as the skeletal muscle[18,49]. This study explores potential mechanisms responsible for TRF-mediated improvement of muscle function under conditions of obesity (HFD and *Sk2*). From transcriptomic analyses, common up/downregulated genes under TRF having orthologs to humans (Supplementary Fig. 2a) were found to be related to glycine production (*Sardh*, and *CG5955*)[50,51], SAM regulation (*Gnmt*)[9], and triglyceride synthesis (Dgat2)[52]. In addition, we found that

flies under HFD-TRF predominantly showed upregulated genes related to the purine cycle. In contrast, *Sk2*-TRF flies showed upregulation of the gene encoding the catalytic subunit of AMPK (AMPKα) and downstream pathways involved in glycolysis, glycogen metabolism, TCA cycle, and ETC.

Previous studies have observed that GNMT allows the universal methyl donor, SAM, to be converted to SAH by transferring a methyl group to glycine[53]. Interestingly, higher SAM levels in older adults correlated with increased fat mass and truncal adiposity, suggesting a role in obesity. Furthermore, a related study observed that SAM was increased in overfed humans[8,54]. In our study, we observed TRF-mediated upregulation of *Gnmt*, together with *Sardh* and *CG5955*, glycine producers that can assist SAM catabolism[9,32] (Fig. 2a). We found that measured glycine levels were significantly increased in *Sk2*-TRF while HFD-TRF and WT-TRF did not mirror the same level of increase. We hypothesize that glycine increases in HFD-TRF may not be measurable because of HFD-TRF-induced purine cycle activation, where glycine is consumed by *phosphoribosylglycinamide transformylase* (encoded by *Gart*). Both essential purine cycle genes *Gart* and *Nmdmc* were found to be significantly upregulated in HFD-TRF but not in *Sk2*-TRF, suggesting that glycine consumption through the purine pathway occurred only in HFD. Interestingly, inducing *Gart* KD in HFD-TRF led to glycine levels being significantly increased compared to HFD-ALF. While glycine levels were largely unchanged in the WT condition, we are uncertain if further aging is required to show the effects of TRF on glycine levels in WT flies which are generally healthier compared to HFD and *Sk2* (Fig. 2c). IFM-specific KD of *Gnmt*, *Sardh* and *CG5955* using *Act88F* displayed a significant reduction in muscle performance (Fig. 2d). Furthermore, cytology of *Gnmt*, *Sardh*, and *CG5955* KD flies displayed ectopic infiltration of lipids in the skeletal muscle (Fig. 2e–g). Interestingly, upon IFM-specific KD of *Gnmt*, *Sardh*, and *CG5955*, previously observed TRF-mediated muscle improvements were abolished signifying the three genes' importance in TRF-mediated muscle improvement (Fig. 2h).

*Dgat2* downregulation was observed in all TRF conditions. A recent study found that overexpression of *Dgat2* in glycolytic type 2 muscle led to increased lipid accumulation and insulin resistance in mice[55]. This aligns with our observation that muscle performance during aging was retained in IFM-specific *Dgat2* KD flies (Fig. 4c), and significantly less lipid accumulation in muscle was seen (Fig. 4d–f). Moreover, IFM-specific overexpression of human *Dgat2* (h*Dgat2*) resulted in reduced muscle performance (Supplementary Fig. 4e) and increased lipid accumulation in muscle (Supplementary Fig. 4f, g), signifying a conserved role of *Dgat2* and its translational potential in humans. Interestingly, subjecting *Dgat2* KD flies to RD- and HFD-TRF showed additional improvements in muscle performance, accompanied by a further reduction of *Dgat2* levels (Supplementary Fig. 4c, d). Moreover, flies with h*Dgat2* overexpression demonstrated

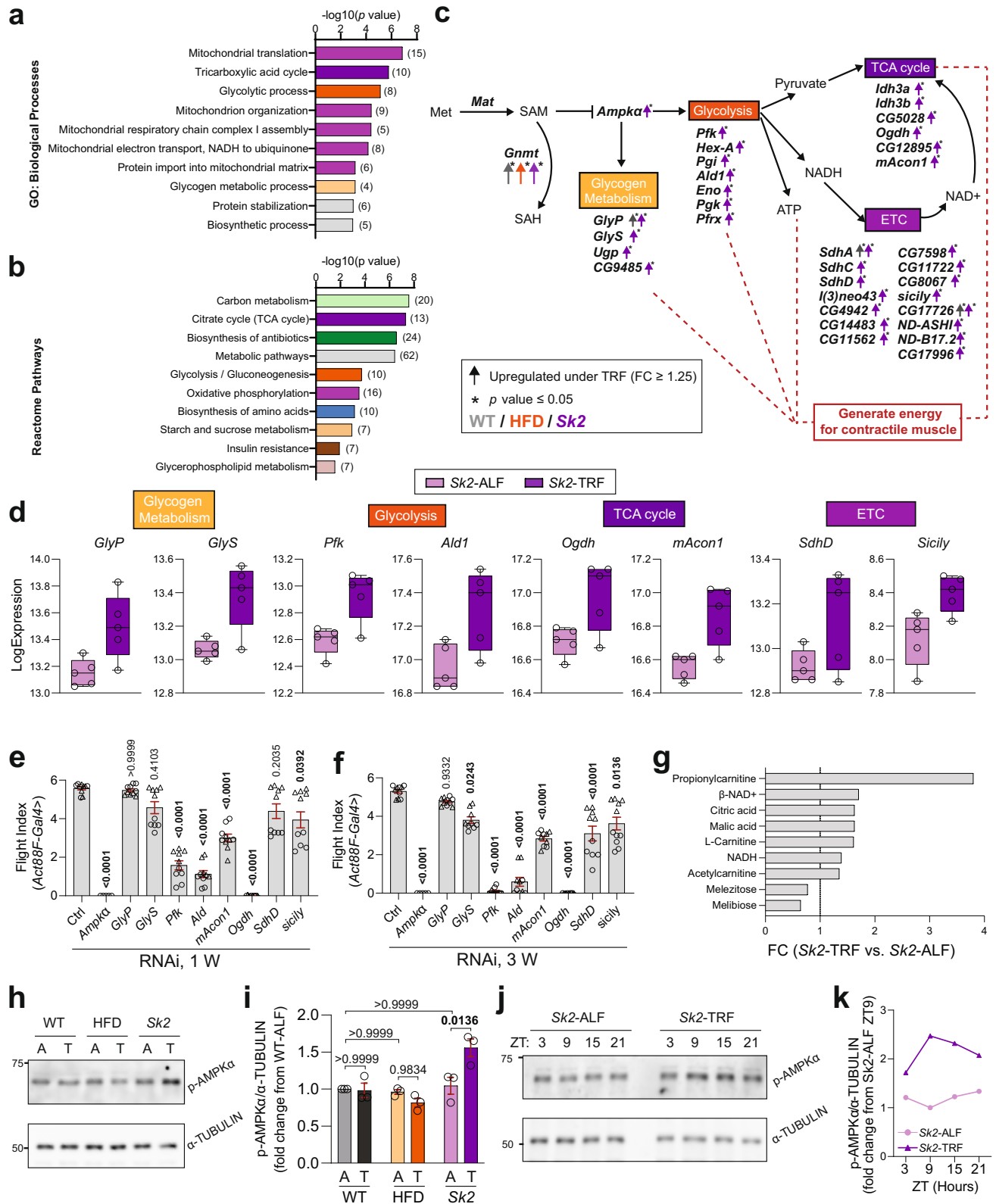

improved muscle performance under TRF compared to their ALF counterparts, and interestingly, with reduced expression of endogenous *Dgat2* (Supplementary Fig. 4h, i). Taken together, our results suggest the possibility that TRF-mediated reduction of *Dgat2* provides benefits to muscle performance in both *Dgat2* KD and h*Dgat2* overexpression flies. This does not however, preclude any other pleiotropic effects of TRF which may have played contributing roles in the

observed muscle improvement. It is known that *Dgat2* knockdown increases de novo synthesis of fatty acids from glucose towards a TAG pool, which is simultaneously hydrolyzed, yielding fatty acids for mitochondrial oxidation[56]. This may suggest TRF's ability to mediate glucose metabolism in muscle stems from *Dgat2* reduction.

Evaluating upregulated genes predominantly in HFD-TRF, we observed genes with functions relating to the purine cycle and

**Fig. 6 | TRF upregulated genes associated with AMPK and downstream pathways in the *Sk2* model. a** GO analysis and **b** Reactome pathway analysis of 408 genes that were significantly upregulated under TRF in *Sk2* flies. Bar charts represent the -log10 (*p*-value) of each enriched GO term and pathway. Related pathways are colored similarly. The number of genes identified in each GO term and pathway is shown in parentheses. **c** Schematic representation of the connection between *Gnmt*, *Ampkα*, and downstream of AMPK signaling with important genes (italic bold) encoding enzymes. The color indicates condition (gray WT, Orange HFD, purple *Sk2*). Up arrows indicate upregulation of gene expression (fold change ≥ 1.25). Asterisks indicate *p*-value ≤ 0.05. **d** Expression level of significantly upregulated genes in glycolysis, glycogen metabolism, TCA cycle, and electron transport chain under TRF versus ALF in *Sk2* flies. *N* = 5 time points in WT and *Sk2*. *N* = 6 time points in HFD. **e, f** Flight index of *Act88F-Gal4*-driven KD of representative genes associated with AMPK downstream pathways at 1 (**e**) and 3 (**f**) weeks of age in female flies. Two independent lines per gene were tested. Flight indices were indicated as cohorts of 10–24 flies. Results from independent RNAi lines were indicated with the symbol circle or triangle. Results from *Ampkα* RNAi #1 are shown in Fig. 6e, f and results from *Ampkα* RNAi #2 are shown separately in Supplementary Fig. 8g due to varying severity of phenotypes and different KD efficiency verified by PCR (Supplementary Fig. 8f). # of cohorts per RNAi line per condition = 4–5; # of flies per genotype per age group = 112–174. Details of precise *N* are in Source Data. Mean ± SEM. One-way ANOVA with Sidak post hoc tests. **g** Metabolites (fold change ≥ 1.25 under *Sk2*-TRF versus *Sk2*-ALF) associated with the AMPK signaling pathway. *N* = 3 biologically independent replicates. Decreased di- and trisaccharides are potentially broken down in glycogen metabolism. Increased metabolites are related to TCA, and ETC pathways. **h** Representative western blot of p-AMPKα levels (top) and α-TUBULIN (bottom) from 3-week-old female IFMs in WT, HFD, and *Sk2* flies under ALF (A) and TRF (T). Samples were collected at ZT 9. (i) Ratios of p-AMPKα/α-TUBULIN normalized to WT-ALF. *N* = 3 biological independent samples. Mean ± SEM. Two-way ANOVA with Sidak post hoc tests. **j** Representative western blot of p-AMPKα levels (top) and α-TUBULIN (bottom) from 3-week-old female thoraces at ZT 3, 9, 15, and 21 in *Sk2*-ALF and *Sk2*-TRF flies. **k** Ratios of p-AMPKα/ α-TUBULIN (normalized to trough of *Sk2*-ALF value) from 3-week-old female fly thoraces at ZT3, 9, 15, and 21 in *Sk2*-ALF and *Sk2*-TRF flies. Source data are provided as a Source Data file.

immune-related response (Fig. 5a, b). Although involved in immune function, the purine cycle also helps balance energy requirements potentially needed for muscular contraction through several ways, including replenishment of the TCA intermediate of fumarate and increasing flux for adenylate kinase[57]. A key gene for both TCA anaplerosis and adenylate kinase flux, *AdSL*, was significantly upregulated along with most upstream enzymes (Fig. 5c, d, and Supplementary Data 5). Interestingly, KD of *adenylosuccinate lyase* (*AdSL*) using *Act88F* driver showed impairment of flight index, while using *DJ694* driver showed normal flight performance but lost the TRF benefits in muscle (Fig. 5e, Supplementary Fig. 7f). In this study, comparing collected metabolites from HFD-TRF to HFD-ALF flies, we found an increase in ADP (a precursor to ATP), fumarate (an entry point for enriching the TCA), and malate (a subsequent product of fumarate upon entering TCA). Furthermore, HFD-ALF flies were found to have higher levels of inosine, hypoxanthine, and xanthine (Fig. 5f, and Supplementary Fig. 10), which are markers of ATP catabolism and ATP exhaustion[58]. This corroborates our observation of reduced ATP levels in HFD-ALF compared to WT-ALF and HFD-TRF displayed increased levels of ATP (Supplementary Fig. 7h). This may suggest that ATP is a key component modulated by TRF and involved in mediating muscle improvement under HFD conditions. Although our results indicate purine involvement in HFD-TRF, the results are limited in its ability to show dynamics and changes in metabolite levels over time. To show dynamics, more sophisticated methodology requiring C13 labeling would be required. Interestingly, a recent human study comparing 11 human individuals with obesity under TRF (8-h eating window) and extending feeding (EXF; 15-h eating window) showed purine cycle genes upregulated as well in TRF[20] (Supplementary Fig. 12a, b, and Supplementary Data 7). These findings provide support for the significance of the purine cycle in humans, however, the cause of obesity from this study was undisclosed and differences existed in TRF duration and eating window period. Another study has shown that ADSL plays a role in ATP generation and a new role of ADSL has been recently uncovered as an insulin secretagogue leading to insulin release and glucose uptake[59]. Taken into conjunction with the downregulation of *Dgat2*, *AdSL* may also aid in TRF's ability to combat insulin resistance under HFD conditions.

Furthermore, an entry point of the purine cycle, 10 formyl tetrahydrofolate (10-formylTHF) has been negatively correlated with insulin resistance and obesity[60]. We found *Pug* and *Nmdmc* are significantly upregulated, which helps produce 10-formylTHF (Fig. 5c, d, Supplementary Data 5). Folate is a crucial component for the final production of 10-formylTHF, which leads to the activation of purine cycle. Folate deficiency can lead to cardiovascular disease, muscle weakness, and difficulty in walking[61,62]. In addition, folate supplementation demonstrated flight improvement and increased relative ATP levels (Fig. 5g,

h). This may suggest that the maintenance of optimal folate levels are crucial for purine cycle activation and may subsequently help to improve muscle performance. However, increased ATP levels in *Gnmt* KD from folic acid supplementation did not mirror the same magnitude of flight improvement seen in *Nmdmc* KD. In addition, control flies with folic acid supplementation demonstrated flight improvement with only modest increases in ATP levels (Fig. 5g, h). This may suggest that not just overall levels of ATP are important but potentially other factors such as ATP flux may play a role in improving flight performance. Interestingly, literature suggests that GNMT may promote purine-related pathways through its aid in the production of 5, 10-methylene tetrahydrofolate with SARDH, which is the entry point of folate into mitochondria[63]. Furthermore, GNMT also modulates purine expression through its translocation to the nucleus in folate-depleted conditions such as HFD[9,63].

We observed DEGs found in *Sk2*-TRF compared to *Sk2*-ALF to be predominantly involved in TCA, glycogen metabolism, glycolysis, and mitochondrial ETC (Fig. 6a, b, and Supplementary Data 6). These pathways are connected through AMPK signaling[64], further the catalytic domain of AMPK was also observed to be significantly upregulated in *Sk2*-TRF (Fig. 6c). It is well known that AMPK acts as an energy sensor able to sense high ratios of AMP/ATP and help regulate lipid metabolism[65]. AMPK is also linked to catabolism and energy production for muscle contraction through ATP production in glycolysis, TCA, and ETC, mediating glucose uptake in tissues such as muscle and mediating fatty acid oxidation[66]. Here, our results showed consistent upregulation of ETC-related genes (Supplementary Data 6), genes encoding TCA key enzymes (aconitase, isocitrate dehydrogenase, α-keto glutarate dehydrogenase) and genes associated with glycolytic and glycogen metabolism (Fig. 6d). We found that metabolites increased under *Sk2*-TRF were L-carnitine, propionyl-carnitine, and acetylcarnitine (Fig. 6g and Supplementary Fig. 9, 11), key components for assisting the production of acetyl-CoA needed for TCA[44]. We also found citric acid and malic acid, products of TCA cycle increased under TRF in *Sk2*. Further, increases in NADH suggest increased NAD+ consumption in TCA cycle while production of NAD+ may suggest increased activity of ETC. In addition, we found melezitose and melibiose, a trisaccharide and disaccharide increased under *Sk2*-ALF, suggesting that these two are metabolized in TRF via glycogen metabolism which is also associated with AMPK signaling. The findings of these metabolites support the involvement of AMPK signaling under *Sk2*-TRF however, is limited in its ability to elucidate the dynamics of these metabolites and changes over time. A previous study has shown that GNMT assists AMPK activation indirectly as SAM's methylation of protein phosphatase 2A (PP2A) leads to inhibition of AMPK activation, therefore, GNMT reduction of SAM may lead to AMPK activation[67].

As GNMT plays a profound role in the proposed pathways, *Crtc* and *FoxO* expression were examined under ALF/TRF conditions in WT, HFD, and *Sk2* as potential transcription factors which mediate *Gnmt* gene expression[33,34]. In *Drosophila*, fasting conditions can trigger activation of CRTC, which has been found to stimulate *Gnmt* and *Sardh* while also upregulating the expression of purine cycle genes (*Ade2*, *AdSL*)[33]. Due to this reason, we hypothesized that CRTC may mediate *Gnmt* expression under HFD-TRF. Indeed, we found that *Crtc* displayed upregulation under TRF compared to ALF in WT and HFD flies, potentially supporting CRTC involvement under TRF in both WT and HFD (Supplementary Fig. 3o). While the above changes were not found in *Sk2*, *FoxO* showed upregulation (5 out of 6-time points) under TRF in *Sk2* only (Supplementary Fig. 3o). Interestingly, a previous study regarding *FoxO* has shown that immune responses in *Drosophila* alter SAM metabolism through dFoxO activation and *Gnmt* upregulation[34].

Our results have shown that TRF-mediated benefits in IFM were abrogated upon KD of *AdSL*, *Nmdmc*, *Ampkα*, *Ogdh*, and *SdhD* (Supplementary Figs. 7f, g and 8h). It is to note that *AdSL*, *Nmdmc*, *Ampkα*, *Ogdh*, and *SdhD* have been demonstrated to be important for muscle development using *Mef2-Gal4* driver[68]. Therefore, the abrogation of TRF benefits could be potentially due to the knockdown of genes essential for muscle function and not necessarily due to the absence of upregulation induced by TRF. In an attempt to rule out this possibility, we used *DJ694-Gal4*, an adult-muscle driver, to induce modest suppression of target gene expression. Interestingly, *DJ694*-driven KD of *AdSL*, *Nmdmc*, *Ampkα*, and *SdhD* didn't lead to any significant decline in flight performance at the age of week 5 (Supplementary Figs. 7f and 8h), suggesting the abrogation of TRF benefits was not caused by any muscle function decline from suppression of tested genes. Although *DJ694* driver expresses in the oenocytes and salivary glands during the developmental phase and abdominal muscle during the adult phase, *DJ694* driver allows to modestly manipulate candidate genes in the adult IFM because of the absence of expression in muscle before the adult stage[27]. However, potential effects on flight performance from *DJ694*-driven KD occurring outside of IFM cannot be assessed by our methods. Further assessment will also be needed to investigate the individual contribution of candidate genes in TRF-mediated benefits.

In summary, TRF previously exhibited improvement in metabolic and skeletal muscle function[17] in *Drosophila*, and this current study provides a potential mechanistic basis for the TRF-mediated benefits. We identified that *Gnmt*, *Sardh*, *CG5955*, and *Dgat2* were modulated across all conditions under TRF. IFM-specific KD of these genes impact ectopic lipid deposition and muscle performance. TRF-mediated benefits in IFM are abrogated upon suppression of *Gnmt*, *Sardh*, and *CG5955*, indicating that TRF-mediated upregulation of *Gnmt*, *Sardh*, and *CG5955* may account for at least a part of the TRF beneficial effect observed in muscle. Furthermore, our transcriptomic and metabolomics analyses demonstrated that distinct pathways were modulated under TRF in both HFD and *Sk2* obese models. While HFD-TRF displayed activation of the purine cycle, *Sk2*-TRF displayed activation in AMPK signaling and downstream pathways. In addition, KD of genes associated with the purine cycle and AMPK signaling led to impaired muscle function. As both the purine cycle and AMPK signaling can modulate ATP levels, our results suggest that TRF modulates the purine cycles in HFD and AMPK signaling in *Sk2*, leading to changes in energy balance and subsequent improvement of muscle function. Overall, our results assess the functional importance of purine cycles and AMPK downstream signaling within the skeletal muscle in different obesity models under TRF potentially initiated by CRTC and FOXO (Fig. 7).

## Methods

### *Drosophila* models, diets, and feeding fasting regimens

Canton-S and Sphingosine kinase 2 (*Sk2*, BDSC: 14133) were obtained from Bloomington *Drosophila* Stock Center (BDSC). *Sk2* is the

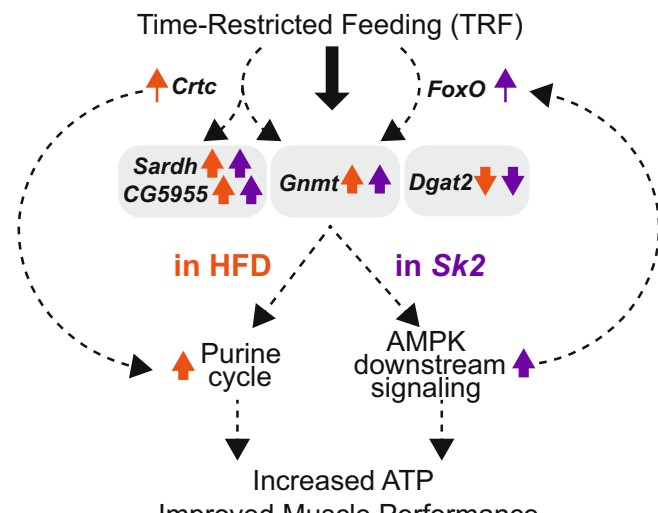

**Fig. 7 | Proposed mechanism of TRF in *Drosophila* skeletal muscle under obesogenic challenges.** 12 h feeding: 12 h fasting mediated TRF intervention modulates common and distinct pathways in the diet- and genetic-induced obesity models. TRF induces the upregulation of Gnmt, Sardh, and CG5955 and the downregulation of Dgat2 in both HFD and Sk2 obesity models. In HFD model, TRF induces upregulation in genes and increases in metabolites related to the purine cycle. Upregulations in the gene expression of Gnmt, Sardh and purine-cycle associated genes are potentially promoted by Crtc. In Sk2 model, upregulation of genes and increases in metabolites relating to glycolysis, glycogen metabolism, tricarboxylic acid cycle (TCA), and electron transport chain (ETC) connected by AMP kinase (AMPK) signaling are observed under TRF. Upregulation in the gene expression of Gnmt is potentially promoted by FoxO, while activation of AMPK may sustain the activation of FOXO. TRF-mediated activation of the purine cycle and AMPK signaling result in higher ATP levels which potentially lead to improved muscle performance in obesity models.

*Drosophila* ortholog of human *Sphingosine kinase 1*. *Sk2* mutants show a hallmark accumulation of ceramide, which has been implicated as a contributor to obesity[69]. Standard regular diet (RD): agar 11 g/L, active dry yeast 30 g/L, yellow cornmeal 55 g/L, molasses 72 mL/L, 10% nipagen 8 mL/L, propionic acid 6 mL/L. High-fat diet (HFD): standard diet supplemented with 5 % coconut oil[17]. For the WT model, CS flies were fed with RD. To model the diet-induced obesity, CS flies were fed with HFD. To test the impact of folic acid, 4 mM of folic acid was added either to a high-fat diet or a standard diet. 4 mM concentration of folic acid was determined for use through titration of different folate concentrations (0.1, 1, 4, 8, and 16 mM) where 4 mM showed the most beneficial impact in muscle performance (Supplementary Fig. 7i). To model genetic-induced obesity, *Sk2* flies were fed with RD. Male and female adult flies were collected upon eclosion and maintained on a standard diet in groups of 20–25 for 3 days. Vials were assigned a diet and feeding regimen on the fourth day. Flies were transferred onto fresh media every 3 days. ALF and TRF flies were subjected to vials with assigned diet at zeitgeber time zero (ZT0) 9 AM (lights on) and were switched to either a food media vial (for ALF) or a 1.1% agar vial (for TRF) at 9 PM (lights off). All flies were kept at 22 °C, 50% humidity in a 12 h light/12 h dark (LD) cycle[18].

To evaluate skeletal muscle-specific function of the identified genes, the following UAS-RNAi transgenic stocks were obtained from Bloomington Drosophila Stock Center (BDSC) and Vienna Drosophila Resource Center (VDRC): *Gnmt*-RNAi (BDSC: 42637; VDRC: 110623; VDRC: 25983), *Sardh*-RNAi (BDSC: 51883; VDRC: 108873, VDRC: 27601), *CG5955*-RNAi (BDSC: 64566, VDRC: 102443; VDRC:15838), *Dgat2*-RNAi (VDRC: 107788; VDRC: 7942), *Nmdmc*-RNAi (BDSC: 62268; VDRC: 5706), *AdSL*-RNAi (BDSC: 34347; VDRC: 6343), *Ampkα*-RNAi (BDSC: 57785; BDSC: 35137), *GlyS*-RNAi (BDSC: 50956; VDRC: 35137), *GlyP*-

RNAi (VDRC: 27928; BDSC: 33634), *Pfk*-RNAi (BDSC: 34336; VDRC: 105666), *Ald1*-RNAi (BDSC: 26301; BDSC: 65884), *Ogdh*-RNAi (BDSC: 33686; VDRC: 50393), *mAcon1*-RNAi (BDSC: 34028; BDSC: 24756), *SdhD*-RNAi (BDSC: 65040; VDRC: 26776), *Sicily*-RNAi (BDSC: 55442; VDRC: 103029), *Sk2*-RNAi (BDSC: 35570), and control RNAi lines (BDSC: 36303, BDSC: 36304). The following overexpression stocks were obtained from BDSC: UAS-hDgat2 (BDSC: 84854) and UAS-GFP (BDSC: 5431). UAS-Gnmt overpression stock was obtained from the laboratory of Masayuki Miura[32]. Data from RNAi lines were combined if more than one RNAi lines were used unless otherwise indicated. *Act88F-Gal4* and *DJ694-Gal4* were obtained from the laboratory of Richard Cripps[27]. *Act88F-GS-Gal4* was obtained from the laboratory of Fabio Demontis[31]. For experiments using the *Act88F-GS-Gal4* driver, flies after day 4 post-eclosion were fed using food supplemented with titrated concentrations of RU486 (mifepristone; Calbiochem #475838) (10, 50, and 100 nM) unless otherwise indicated. RU486 was dissolved using ethanol. The corresponding control flies were fed using food supplemented with the same volume of ethanol alone. Ethical approval was not required for the experiments performed in this study given the exclusive use of fruit flies.

## RNA-Seq expression profiling
Similar to our previous paper[22], after 3 weeks of age on the assigned diet and feeding regimen, indirect flight muscle (IFM) was collected from 10 female flies every 4 h over 24 h. Samples were flash-frozen and homogenized using a pellet pestle. Total RNA was prepared using the RNeasy kit (QIAGEN). Libraries were prepared according to Illumina protocols for single-end 50 bp mRNA sequencing and sequenced on an Illumina HiSeq 2500. Reads were aligned to the *Drosophila* melanogaster genome (dm6) using STAR (v2.5.0e). Only reads aligning uniquely to the genome were considered. Normalized gene expression values were found by counting alignments overlapping with exons from RefSeq gene models. WT-TRF-ZT11 and *Sk2*-ALF-ZT23 were detected as outliers, therefore, data from these two time points were removed during analysis. The two outliers and their corresponding ALF/TRF counterparts were not shown in the expression figure panels.

Analysis of differential expression was carried out using DESeq2 (v1.8)[23], with designs that accounted for models (WT, HFD, and *Sk2*) and feeding group (ALF and TRF). Statistical significance was assessed using a negative binomial Wald test. The threshold of differential gene expression was set as fold change $\geq$1.25 and a *p*-value $\leq$ 0.05. For visualizations, a regularized logarithm transformation[23] was applied to the gene-level read counts. Gene expression heat maps were generated from these transformed values following mean-centering. Principal component analysis was performed using the 500 genes with the highest variance across all samples. Gene Ontology Biological Process analysis and Kyoto Encyclopedia of Genes and Genomes (KEGG) pathway enrichment analysis were performed by the Database for Annotation, Visualization and Integrated Discovery (DAVID) v6.8. The GO terms were manually curated so that there were no more than three genes that were shared between any two GO terms. To analyze the temporal dynamic of gene expression, we used empirical JTK_CYCLE with asymmetry search[40]. Periodic patterns were determined using duplicated data i.e. two cycles were artificially created from the one actual cycle of experimental data[18]. Transcripts with a maximum/minimum fold-change $\geq$1.3 and a Benjamini-Hochberg corrected *p*-value $\leq$ 0.05 were considered periodic. The expression level (normalized read count, log2) of genes are plotted as box plots. The horizontal line in each box represents the median value. The endpoints of the box represent the 25th–75th percentiles. The whiskers extend to the lowest and highest values within 1.5 times the interquartile range (IQR) from the 25th–75th percentiles.

## Muscle performance
To evaluate the roles of identified genes in muscle performance, a flight test was conducted using flies with KD or overexpression of selected genes driven by *Act88F-Gal4*, *DJ694-Gal4* or *Act88F-GS-Gal4*. Briefly, this methodology[18,70] involves the release of control and experimental adult flies into the center of a Plexiglas box with a light source positioned at the top. Flies were released as cohorts containing 10–20 flies. Based upon each animal's ability to fly up [6.0], horizontally [4.0], down [2.0], or not at all [0.0], we assign flight indices (FI) for different ages and genotypes of flies. Details of fly ages, genotypes, conditions, cohort numbers, total fly numbers, and flight index per cohort are documented in Source Data.

## Negative geotaxis
As previously reported[18] flies were transferred to a new vial (10–13 flies per experiment) and allowed to rest for 2 min to acclimatize. The vial was then sharply tapped three times to stimulate a negative geotaxis response. The climbing ability of the flies was video-recorded and saved for data analysis. At 10-s intervals, the fraction of flies that climbed to the 7 cm mark was calculated.

## Cytological analysis of muscle and abdomen in fat body tissues
As previously established[18] for cytological analysis, samples were prepared and fixed in 4% PFA, washed with PBS, and then aligned longitudinally (thoraces) in a cryomold filled with Tissue-Tek OCT (Sakura), and then flash frozen on dry ice. 30 μm thickness cryosectioned samples were washed and stained with 1× fluorescence dye 488-I with Phalloidin conjugate (1 μg/ml) for the structural aberrations, and Nile red (1 μg/mL) in 1× PBS for the lipid staining. Fluorescence images were taken using an Olympus BX-63 microscope coupled with cell-tracking software, and quantification of lipid droplets (size and density) was performed using Python scripting language (CITE) and consisted of images simple binary and Otsu thresholding and watershed segmentation.

## Western blot analysis
Five thoraces were added to 75 μl of lysis buffer (62 mM Tris pH7.5, 0.1% SDS with protease and phosphatase inhibitors). The samples were boiled for 5 min at 95 °C and centrifuged to collect the supernatant. Sample loading buffer was added, and the samples were then electrophoresed on 10% Mini-PROTEAN TGX precast protein gels (Bio-Rad). The protein bands were transferred to the nitrocellulose membrane (Bio-Rad) and incubated with AMPK-alpha antibody (Cat.: ab80039, 1:3000, Abcam), Phospho-AMPK-alpha (Thr172) antibody (Cat.: 2535 S, 1:1000, Cell Signaling), alpha Tubulin antibody (Cat.: ab52866, 1:3000, Abcam). For the secondary antibody, horseradish peroxidase-conjugated Goat anti-rabbit IgG (ab6721, Abcam) and Rabbit anti-mouse IgG (ab6728, Abcam) were used at 1:10,000 dilution. Clarity Max or Clarity Western ECL substrate (Bio-Rad) was used to detect the signal, and the corresponding band was quantified using ImageJ software.

## Glycine measurement
Ten thoraces from 3-week-old female flies per condition were isolated and glycine levels were measured using a fluorescence-based Glycine Assay Kit (Fluorometric), (Abcam, ab211100). The method for glycine measurement was followed according to the manufacturer's protocol. A small volume of the homogenate was used for detecting protein levels using a Bradford assay (Bio-Rad) for the normalization of samples.

## ATP quantification
ATP levels in the thoraces of 3-week-old flies were quantified with a luciferase-based ATP kit (Thermo Fisher). The collection of ATP was followed according to a previous study[71]. Briefly, fly thoraces were homogenized using a homogenization buffer (6 M guanidine HCl, 100 mM Tris (pH 7.8), 4 mM EDTA) and a pellet pestle. A small volume

of the homogenate was used for detecting protein levels using a Bradford assay (Bio-Rad). The remaining homogenate was lysed via boiling for 5 min and was centrifuged at maximum speed for 3 min at 4 °C. Dilution of the luciferin and luciferase was made per the instruction of the ATP kit and ATP-dependent luminescence was measured at 560 nm. ATP measurements were taken from triplicates per condition and were averaged after a total of 3 sequential readings, which were normalized according to protein levels; both protein and ATP were measured using the EPOCH Biotek plate reader.

### Metabolite sample preparation for LC-MS/MS analysis

Using 3-week-old ALF and TRF flies from WT, HFD, and Sk2 models, an aliquot (100 µL) of 10 homogenized IFMs (3 replicates per condition) in PBS were treated with ice-cold 80% methanol (400 µL) for 30 min to extract metabolites. The extractions were centrifuged at $3000 \times g$ for 10 min at 4 °C, and the supernatant was transferred to a new test tube and dried to completion under nitrogen gas. The samples were then reconstituted in 100 µL of 0.1% formic acid in ddH$_2$O for mass spectrometry analysis. An aliquot (10 µL) of each sample was loaded onto a Phenomenex 2.1 × 100 mm, 2.7 µm Luna Omega, 80 Å reverse-phase column (Torrance, CA). Metabolites were resolved using a linear gradient (2–50%) of acetonitrile in 0.1% formic acid for 5 min and then a 50–98% acetonitrile gradient until 6.0 min with a 1 min further hold at 98% acetonitrile. The column was then re-equilibrated at initial conditions (0.1% formic acid) for 3 min. The mobile phase flow rate (500 µL/min) was generated by an Exion UHPLC (SCIEX, Toronto, Ontario). The SCIEX 5600 TripleTOF mass spectrometer (SCIEX, Toronto, Canada) was used to analyze the metabolite profile. The Ion Spray voltages for positive and negative modes were +/−5000/4500 V and the declustering potentials were +/−80 V, respectively. Ionspray™, GS1/GS2, and curtain gases were set at 40 psi and 25 psi, respectively. The interface heater temperature was 400 °C. For all samples, sample pools, and blanks, MS1 spectra were collected as time-of-flight survey scans from $m/z$ 50–1000. For the pooled samples, a 0.5 s duty cycle consisted of a 100 ms time-of-flight survey scan followed by eight 50 ms product ion time-of-flight scans over the range from $m/z$ 50–1000 using a collision energy spread of 15 eV with a set collision point of 35 eV. Spectra were centroided and de-isotoped by Analyst software, version 1.81 TF (SCIEX).

### Data analysis and metabolite identification

LC-MS data were processed using MS-Dial (RIKEN Center, Yokohama City, Kanagawa) (version 4.80) to identify ion features occurring across all samples, to align them and to determine the peak areas and their retention times. After removing ions coming from the blank extracts, as well as those that were poorly reproducible, an Excel.csv file was created containing the $m/z$, peak area and retention time of each metabolite ion. This file was uploaded to MetaboAnalyst 5.0 (https://www.metaboanalyst.ca/) for univariant and multivariant statistical evaluations and the generation of heat maps. Putative annotations of metabolites were based on a comparison with the public product ion libraries provided by MS-DIAL. Further assessments were made using the IROA Metabolite Standards compound library (IROA Technologies, Sea Girt, NJ).

### Real-time quantitative PCR

Thoraces or IFMs from 3-week-old flies were collected and flash frozen. RNA was extracted using Zymo Research Quick-RNA Microprep Kit with on-column DNase I digestion. Quantitative PCR was performed using SsoAdvanced Universal SYBR Green supermix (Bio-Rad) in a BIO-RAD CFX Opus Real-Time PCR System. Expression was normalized with 60S ribosomal protein (RPL11). Primers for qPCR are listed below: *Gnmt*-F: ACACGGCGGACATCAAG; *Gnmt*-R: TCGATCAGGTGAATGT AGAAG; *Sardh*-F: CTGTCACACACTGTATCACCTG; *Sardh*-R: CGACGT-GAATTAGCCAGCAG; CG5955-F: AGCCCAGTCAGTCGGTACT; *CG5955*-R: CAGTCAATGCGATGGTCCACA; *CG6806*-F: CACCAACCCTTCACCGA

TGAC; *CG6806*-R: GGCGAAGTAGAAGAGACGCAC; *CG5896*-F: CCTCAG CCAAAGAGTGTCCAA; *CG5896*-R: CTCCGCCGCAAAGGAATCT; *Dgat2*-F: TGTCCAAGTTGTTGGTGCTC; *Dgat2*-R: GGCACTCTTCGAATTCTC CA; *CG7997*-F: ATGGCCTGGCACTAAAGCC; *CG7997*-R: TCCGCATGTC TTTGGAAAAGTT; *CG13992*-F: GCAGCAAACGCACCATGAC; *CG13992*-R: CGGCACATATAATACGGCTCATC; *Nmdmc*-F: GCACCCAACTAGCA-CACGA; *Nmdmc*-R: CGACCATCTTGTTAGCCACATAC; *Ampkα*-F: TGG GCACTACCTACTGGG; *Ampkα*-R: ATCTGGTGCTCGCCGATCTT; *AdSL*-F: TGATCTGATTGTACTGAGGGACG; *AdSL*-R: ACTGGCTTAGACGAGC TATCAC; *Rpl11*-F: CGATCTGGGCATCAAGTACGA; *Rpl11*-R: TTGCGCTT CCTGTGGTTCAC. Results are presented as $2^{-\Delta\Delta Ct}$ values normalized to the expression of Rpl11 and control samples. All reactions were performed in triplicate.

### Statistical analysis

Significance in flight muscle performance and climbing ability was determined using one-way ANOVA or two-way ANOVA with posthoc Sidak's method for multiple comparisons or two-sided unpaired $t$-tests. Lipid droplet size and density differences were performed by one-way ANOVA with post hoc Sidak's method or two-sided unpaired $t$-tests. For western blot analysis, differences between samples were determined using two-way ANOVA with posthoc Sidak's method. Relative ATP quantification analyses were made using two-sided unpaired $t$-tests. Bar graphs show mean ± SEM. All statistical analyses were performed with GraphPad Prism 9.

### Reporting summary

Further information on research design is available in the Nature Portfolio Reporting Summary linked to this article.

### Data availability

The RNA-seq data generated in this study have been deposited in the Gene Expression Omnibus (GEO) database under accession number GSE205334. Genome assembly used in this paper (dm6) is publicly available. Uncropped immunoblots and source data underlying graphs are presented in the "Source Data" file. The reporting summary and supplementary data 1–8 for this article are available under supplementary information. Source data are provided with this paper.

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

## Acknowledgements

This work was supported by the National Institutes of Health (NIH) grants AG065992 to G.C.M. and AG068550 to G.C.M. and S.P., UAB Startup funds 3123226 and 3123227 to G.C.M., and Wu-Tsai Human Performance Alliance and the Wu-Tsai Foundation grant to S.P. The mass spectrometer used in these studies was purchased from an NIH Shared Instrumentation Grant S10 RRO27822 to S.B. We thank Salk Institute Bioinformatics core for the RNA seq and data analysis. We thank Anju Melkani, UCSD dissection/isolation of muscle tissue, Carlyssa Boyd, UAB, and Jesús E. Villanueva, SDSU for implementing TRF. We also thank Landon Wilson from the Targeted Metabolomics and Proteomics Lab (TMPL) at UAB for helping us with our metabolite analysis, Aniket Pant, UAB School of Medicine, for his help with lipid quantification, Dr. Masayuki Miura for the *UAS-Gnmt* overexpression line (UTokyo) and Dr. Fabio Demontis for the *Act88F-Geneswitch-Gal4* line (St. Jude Children's Research Hospital). We also would like to thank Dr. Ralph Sanderson, Dr. Rakesh Patel, Dr. Shannon Bailey, and Dr. Jonathan Roth, UAB Heersink School of Medicine, for their editorial comments on the manuscript. Stocks obtained from the Bloomington Drosophila Stock Center, and Vienna Drosophila Resource Center were used in this study.

## Author contributions

G.C.M. designed the experiments with the help of S.P., C.L., and Y.G. G.C.M., C.L., and H.D.L. prepared sequencing samples. Y.G., H.D.L., and C.L. analyzed transcriptomic data. C.L., Y.G., S.V., and V.R. performed WT, HFD, and *Sk2* model experiments. F.A. performed the acquisition of cytological images. G.C.M. and S.B. designed the metabolomics analyses and S.B. and C.L. performed the resulting data analysis. Y.G., C.L., and G.C.M. prepared the paper with S.P.'s input. All authors provided feedback on the manuscript.

## Competing interests

The authors declare no competing interests.
