## [Peer Review File · Nature Communications]

Time-restricted feeding promotes muscle function through purine cycles and AMPK signaling in *Drosophila* obesity modelsREVIEWER COMMENTS

Reviewer #1 (Remarks to the Author):

This study identifies several genes related to the muscle function (flight ability) of adult flies. The authors first performed a transcriptomic analysis of the indirect flight muscle with or without time-restricted feeding.

They included two obesity models to give a broader view of the regulatory genes of muscle function. These conditions are based on the previous literature and therefore well established and characterised.

The analysis identified *Gnmt* and many other genes that might be key to maintain muscle function under normal as well as time-restricted feeding.

While the study is obviously important, the authors did not address the deeper mechanistic analysis of their findings. Instead, it has listed up genes potentially related to the TRF-mediated improvement of muscle function. Also, they did not discriminate muscle homeostasis (during ageing) from development. Some additional experiments would help to improve the manuscript.

Major comments,

1, In this study, the authors use Act88F-Gal4 to drive knockdowns of the genes of their interest. To my knowledge, Act88F is also expressed during development. If this is the case, it would be better to use Act88F with tub-Gal80ts or Act88F-GeneSwitch, which enables to manipulate only in the adult flies. Also, it would be better to show whether muscles are not affected in early young adults (such as day2~day5). GeneSwitch would also allow precise control of gene dosage, i.e. to block the gene upregulation by TRF. Strong knockdowns sometimes simply induce unhealthy organ development (of muscles). The authors also can test gain of function would result in the opposite phenotype to further strengthen the gene function.

2, Relatedly, the authors should always show, side-by-side, that whether the manipulation (of any genes) influences the muscle function in the normal condition or it abolishes TRF-mediated improvement. For example, in Fig. 2b-d, *Gnmt* was upregulated by TRF, and the knockdown abolished the change of the flight index by TRF. However, there is no control here. The data would be better to present as in Fig. S6c. Actually, the authors did not control the genetic background of the lines. The proper control might be required, especially when the basal Flight index can be different. For example, in Fig. S6c, *Nmdmc*-RNAi has higher Flight index at the basal line and this does not further increase by TRF. This would lead to a different conclusion.

3, While the transcriptional change is clear, the authors did not address the upstream pathways. How and which transcription factors are involved in up- or down-regulation of the genes by obesity and TRF?

4, The authors highlight several metabolic enzymes/pathways. However, unfortunately, no metabolic analysis has been done. It is possible that the gene expression change is due to the adaptation of metabolic alteration. So looking at mRNA level is usually not enough to discuss whether it is a cause or a consequence. In line with this, the authors should analyse whether IMP and related purine/THF metabolites are increased by feeding folic acid. The phenotypes in Fig. 4g-h are beautifully demonstrated but only measuring ATP would not tell how metabolism is affected.

Minor comments,

5, It is not elucidated whether increased lipid in muscle is a cause of the phenotype in Fig. 2e-g. The authors may test *Gnmt*-, *Sardh*-, or *CG5955*-RNAi together with *DGAT2*-RNAi to see whether it rescues the decreased flight index.

6, Fig. 2b is not mentioned in the text.

7, Fig. S4c,d in the text is actually Fig. S6c,d.

8, Is food intake affected by the gene manipulations, e.g. *Gnmt*-RNAi?

Reviewer #2 (Remarks to the Author):

Livelo and colleagues are investigating how time restricted feeding (TRF) of different diets does impact muscle function during adult age in a *Drosophila* model. The authors feed flies for only 12h per day compared to ad libitum feeding and age them, while monitoring muscle function by performing a flight assay.

How TRF positively affects organism health in general or individual organs in particular during ageing is an exciting field of study, which still lacks mechanistic insight, specifically in the muscle.

To unveil a mechanism, the authors performed transcriptomics analysis of entire thoraces (not indirect flight muscles, as stated incorrectly in the abstract) and identified some metabolic enzymes displaying changed expression upon TRF on various diets or genetics models.

Interestingly, knock-down of one of them, *Dgat2*, appears to result in TRF-like improvement of flight muscle function during ageing despite ad libitum feeding.

In general, the paper appears to superficially follow various candidate genes that are not obviously related. Hence, it has an unclear title and a not very specific abstract. Many of the data presented appear preliminary of results are often over-interpreted.

I am mainly commenting on the transcriptomics and genetic studies of the paper.

1. The authors used entire thoraces as source for the transcriptomics study. However, in the results and in the abstract the authors claim to have collected indirect flight muscles (IFMs). While the thorax is largely consisting of IFMs it contains various other muscles and digestive organs, hence this statement is incorrect.

2. The authors have isolated RNA samples every 4 hours over the course of one day, meaning 6 samples for each of the various feeding conditions. These samples were pooled to perform differential expression analysis using DE-Seq2, which requires replicates. The authors have not collected true replicates for any of the samples to verify the validity of this simplification procedure. As expression for many genes cycles during the day, this simplification is questionable. Similarly worrying is that the Suppl Table 1 is missing several of the samples entirely.

Together with the low threshold of a FC 1.25 (not log₂FC, which is standard) chosen by the authors, the data presented in Figure 1 are at least questionable and would require more detailed verifications for each presented gene. As only 8 genes were followed up, this verification would be doable by qPCR or tagged protein quantification.

3. The authors test flight muscle function in a quite tedious version of the flight assay that can identify small differences between the different conditions following flight muscle specific gene knock-down (Figure 2). This reviewer would like to see an individual results table to verify the statistics. However, neither the number of flies tested in the various conditions, nor their individual performance was provided in the Supplementary Tables. It is also unclear how often this experiment was repeated. It is well known that high GAL4 expression levels present in Act88F-GAL4 can impact flight. Was the 'wild type' control containing Act88F-GAL4? This is unclear from the figure legends or methods.

4. How does *Gnmt*, *Sardh* and CG5955 function relate to TRF or different diets? Why is CG5955 called TdH only in Figure 2g?

5. A potential beneficiary role of *Dgat2* knock-down in IFMs is interesting. It would be important to verify this result by either a second RNAi line or some other genetic tools to rule out off-target effects (Figure 3).

6. Also Figure 4 is missing second RNAi lines for all of the genes shown.

7. The authors find that AMPK phosphorylation is reduced in the model of sphingosine kinase 2 mutants (Figure 5). However, there is no difference in AMPK-phospho levels comparing TRF with ad libitum feeding. Hence, what have we learnt about the TRF protecting mechanism? It is trivial that knock-down of any of the essential glycolysis, TCA or ETC enzymes compromise flight activity. Again, there is no link to the feeding scheme provided.

Reviewer #3 (Remarks to the Author):

Livelo et al. present a mechanistic approach to explain the benefits of time-restricted feeding (TRF) to skeletal muscle function in obesity using the *Drosophila* model organism. To this aim they apply a comparative circadian transcriptome analysis between skeletal muscle-enriched thorax material from wild type (WT) flies and diet- (DIO) or genetically-induced (GIO) fly obesity models under TRF or ad libitum fed (ALF) conditions.

By in silico pathway analysis the authors identify three genes involved in glycine production/usage to be up-regulated under TRF in all three models. Tissue-specific knockdown of all three individual genes impairs flight performance, blunts the TRF benefits in this assay and promotes structural muscle degeneration and ectopic lipid accumulation in aging fly indirect flight muscle.

Along these lines the authors identify fly *Dgat2* as the only gene specifically down-regulated in all models under TRF vs. ALF. Flight muscle-specific knockdown of this gene improved flight performance and suppressed lipid accumulation under ALF in aging WT flies.

In the following the authors use the same strategy to identify TRF-dependent transcriptional pathways specific for their HFD DIO and their *Sk2* mutant GIO model, respectively.

For the DIO model, they propose that TRF operates via purin biosynthesis and folate cycle to potentially improve muscle ATP availability. In support of this interpretation, they demonstrate that folic acid supplementation reverses some effects of HFD diet on physical performance and ATP levels in flies with genetically impaired folate cycle.

Similarly, TRF up-regulates the AMPK α gene and a broad range of genes in AMPK downstream signaling pathways to improve ATP availability in the *Sk2* mutant GIO model. In support of the functional implication of these transcriptional regulations, tissue-specific knockdown of some of these genes selectively impaired physiological performance in an age-dependent manner.

In general, this is an interesting study which targets the mechanism(s) of the evolutionarily conserved health benefits of TRF in obesity. The comparative transcriptome analysis followed by gene ontology and reactome analyses is straightforward and provides a wealth of data on candidate genes.

Comparing the TRF-response of different genotypes and diet conditions is an elegant way to focus on important pathways.

However, the functional data of this study are much less convincing due to a lack of metabolite analyses in support of the physiological consequences of the described gene regulations. Also, compensatory gene regulations by RNAi-mediated gene knockdown are poorly controlled leaving it largely open whether they recapitulate TRF-dependent physiological phenotypes or more fundamental impairments of the underlying metabolic pathways. The *Dgat2* analysis lacks a compensatory gain-of-function approach.

This study would benefit from focusing on the more extended shared pathway analysis. Functional analysis of the individual DIO and GIO models is not very elaborate and suffers from the question how representative HFD and global *Sk2* mutants are for DIO and GIO, respectively.

Major points:

1) Transcriptional fold-changes of ≥ 1.25 appear fairly subtle. In particular, compared to the impact of tissue-specific gene knockdown used for functional interventions. Flightless flies caused by some of the RNAi knockdowns suggest a strong overcompensation of the TRF-induced up-regulation of the genes of interest or even developmental effects. The authors have to take measures and demonstrate that the knockdown is timely and compensates but not overcompensates the TRF-induced up-

regulation.

- 2) Does the glycine pool change as predicted in response to the transcriptional changes?
- 3) Several aspects of Fig. 2d-g are questionable. *Sardh* and CG5966 RNAi have very different impact on flight performance but muscle degeneration and ectopic lipid storage are comparable (and differ from the control). This challenges the view that we are looking at mechanistic rather than correlative changes. Moreover, how does a glycine pool reduction cause TAG accumulation?
- 4) *Dgat2* is down-regulated under TRF. The authors need to demonstrate that TRF benefits (partially) vanish in response to compensatory *Dgat2* overexpression adjusting the gene expression to *Dgat2* levels under ALF. The functional *Dgat2* data rely on a single RNAi construct (knockdown efficiency?), which is weak evidence. In particular, as *Dgat2* is one of three paralogous genes of the *Dgat2* family in flies.
- 5) Fig. 3d-f is not convincing. Size and abundance of the LDs appear too small for robust conclusions about possible changes. Also, these parameters need to be shown as scatter plots to more readily display the distributions.
- 6) Fig 2c, 3c: WT is an inappropriate control, which does not address driver line effects. Also, in 2c the very same control seems to be shown three times in all subpanels. Are they identical to the data shown in Fig. S4g?
- 7) FA supplementation is promising. However, how do transcriptional changes translate into changes in FA and FA-related metabolites? Direct measurements are necessary.
- 8) The flight index data of "Ctrl RNAi" in Fig. 4f and "Ctrl" in Fig. 5h are remarkably similar. Given that they are independent, why do "WT" flight indices vary so much between Fig. 2c or 3c and Fig. S6f?
- 9) The broad transcriptional up-regulation in GIO in response to TRF is interesting. However, how this translates to metabolic outcome is insufficiently addressed. The pAMPKalpha response is at the significance limit and the individual, strong knockdown of selected genes shown in Fig. 5h presents no conclusive picture. With other words without a complementary metabolite profile the GIO data remain preliminary.
- 10) Fig. S3b shows identical plots assigned to different genes.
- 11) Fig. S3c: "Ctrl" is "WT" according to text.
- 12) Transcriptional regulation of the central genes of interest derived from RNAseq data requires qRT PCR validation.
- 13) "... IFM-specific RNAi KD of *Gnmt*, *Sardh* and CG5955 resulted in a cytotoxic effect in the adipose tissue ..." These data need to be presented (in the supplement).

Minor points:

- 14) The title sounds very general referring to DIO and GIO models, while a mild HFD diet and the Sk2 mutant are single models of unknown representative value for DIO and GIO. Given the wealth of transcriptome data the finding of "shared and unique pathways" sounds trivial. In particular, as it remains open if the unique pathways are unique for HFD diet and Sk2 mutants or for DIO and GIO.
- 15) Fig. 2f,g uses once CG5955 and once *Tdh* for the same gene.
- 16) Some SEMs are missing e.g. Fig. 3c *Dgat2* RNAi, Fig. 4h Ctrl RNAi
- 17) Fig 5c Ampk or Ampkalpha? "Generate" needs attention.
- 18) The Actin88F-Gal4 driver needs more description concerning the developmental time- and tissue-specificity.
- 19) For RNAseq analysis, which samples were "ground to fine powder in liquid nitrogen" and which tissues were treated with the "Polytron homogenizer"?
- 20) The description of the p-AMPKalpha data is misleading as a single of the tested conditions is statistically significant.
- 21) Page 7: Supplementary Fig. 4c and 4d should read Supplementary Fig. 6c and 6d.
- 22) "Abdomen lipid staining..." Which abdominal tissue has been analyzed in Fig. S4c-e? Fat body? Muscle? The quality of Fig. S4c is low but does not support the quantifications in S4d,e as the density is clearly larger in *Dgat2* RNAi while the LD area appears smaller.
- 23) Where are the RNAseq data deposited?

Reviewer #4 (Remarks to the Author):

NATCOM-21-32964

Time-restricted feeding promotes skeletal muscle function in diet- and genetic-induced obesity through shared and unique pathways

C Livelo, Y Guo, S Varshney, F Abou Daya, HD Le, S Panda & GC Melkani

This is a very interesting study, providing more insight in the molecular mechanisms that may be responsible for the beneficial health effects of time-restricted feeding (TRF). Especially testing the functional importance of the genes identified by the transcriptomics by the muscle specific knock down and following flight assays, is a strong point of this study.

Nevertheless, I have a couple of remarks:

At several occasions the authors mention the importance of muscle tissue for glucose metabolism. However, in their Discussion it is not discussed whether the current results provide any insight in how TRF could improve glucose metabolism.

At the beginning of their Introduction the authors mention 3 major contributors to the current obesity epidemic, genetic predisposition, calorie dense diets and chronodisruption. However, in their study they only use a DIO and GIO, but no CIO model. Although, I do not know whether chronodisruption also results in obesity in *Drosophila*, it would have been nice to discuss this aspect.

Related to this, in rats and mice the TRF strategy has often been used as a model for chronodisruption by restricting food access to the normal sleep phase. Again I do not know whether this works in *Drosophila*, but it would be nice to know whether the changes in muscle tissue observed in the current study are opposite to those described in the rat and mice studies employing TRF during the sleep phase. I think at least the authors should mention/discuss this aspect.

Figure S1 shows that more rhythmic genes were found during TRF than ALF in the WT and DIO groups, which is what was to be expected based on previous studies. However, in the GIO this pattern was reversed, i.e. more rhythmic genes in the ALF than in the TRF group. Misleading in this figure is that circle size is not representing the number of genes, in the WT and DIO groups the difference between ALF and TRF is approx. 100 genes, but in the GIO group this difference is >2000 genes!

Typo's:

Third line in the 3rd paragraph of the Introduction, "Recent" does not need a capital to start with.

First line in the final paragraph on page 5, "the accumulation" can be removed.

Second line on page 6, what is meant with the "cytotoxic effect in adipose tissue"?

Final lines of the 2nd paragraph on page 6, what is meant with "on aging"? Probably it should read "with aging"?

Response to the reviewers

Reviewer #1 (Remarks to the authors):

This study identifies several genes related to the muscle function (flight ability) of adult flies.

The authors first performed a transcriptomic analysis of the indirect flight muscle with or without time-restricted feeding.

They included two obesity models to give a broader view of the regulatory genes of muscle function.

These conditions are based on the previous literature and therefore well established and characterised.

The analysis identified *Gnmt* and many other genes that might be key to maintain muscle function under normal as well as time-restricted feeding.

While the study is obviously important, the authors did not address the deeper mechanistic analysis of their findings. Instead, it has listed up genes potentially related to the TRF-mediated improvement of muscle function. Also, they did not discriminate muscle homeostasis (during ageing) from development. Some additional experiments would help to improve the manuscript.

Major points:

1, In this study, the authors use *Act88F-Gal4* to drive knockdowns of the genes of their interest. To my knowledge, *Act88F* is also expressed during development. If this is the case, it would be better to use *Act88F* with *tub-Gal80ts* or *Act88F-GeneSwitch*, which enables to manipulate only in the adult flies. Also, it would be better to show whether muscles are not affected in early young adults (such as day2~day5). *GeneSwitch* would also allow precise control of gene dosage, i.e. to block the gene upregulation by TRF. Strong knockdowns sometimes simply induce unhealthy organ development (of muscles). The authors also can test gain of function would result in the opposite phenotype to further strengthen the gene function.

Response: To address this important question raised by the reviewer, we have used two additional drivers for muscle-specific knockdown of *Gnmt*, *Sardh*, and *CG5955*: *DJ694-Gal4* (Bryantsev et al, 2012, PMID: 22008792) and *Act88F-GeneSwitch-Gal4* (*Act88F-GS-Gal4*) (Robles-Murguía et al, 2019, PMID: 31123597). *DJ694-Gal4* driver expresses in adults within the indirect flight muscle post eclosion, and *Act88F-GS-Gal4* driver only initiates when RU486 is introduced (Osterwalder et al, 2001, PMID: 11675495; Roman et al, 2001, PMID: 11675496) allowing precise timing of gene expression. We have examined the flight ability (muscle performance) of flies with IFM-specific *KD* of *Gnmt*, *Sardh*, and *CG5955* using a *DJ694* driver. No significant impairment of flight index was observed in 4-day-old flies, while a significant reduction in flight performance was observed at 3 weeks of age (Fig. 3a). Similarly, upon IFM-specific *KD* using *Act88F-GS* driver, flight performance was not affected on day 4 and reduced at 3 weeks (RU486 was supplemented on day 4 post eclosion) (Fig. S3f). These results suggest the crucial roles of *Gnmt*, *Sardh*, and *CG5955* for muscle maintenance. We also performed IFM-specific *KD* of *Dgat2* using *DJ694* driver, and flight performance improvements were observed at 5 and 7 weeks (Fig. 4g).

We have acquired *UAS-Gnmt*, an overexpression fly stock from Dr. Masayuki Miura's lab (Obata et al, 2015, PMID: 26383889), and performed *Gnmt* overexpression using *Act88F* driver. Aligned with our hypothesis, we observed significant muscle improvement compared to age-matched control in 5-week-old flies (Fig. S3g). Furthermore, we also acquired *UAS-hDgat2* fly line from Bloomington Drosophila Stock Center (BDSC: 84854) and performed human *Dgat2* overexpression using *Act88F* driver. Aligned with our hypothesis, we observed significantly reduced flight performance from 3-week-

old flies (Fig. S4c). Overall, as suggested by reviewer 1, we have addressed the roles of *Gnmt*, *Sardh* and *CG5955* on muscle function maintenance by performing gene *KD* during the adult stage. Additionally, we have evaluated the “gain-of-function” of *Gnmt* and *hDgat2* in skeletal muscle, which supports our hypothesis.

Fig. 3a) Flight performance upon muscle-specific *KD* of *Gnmt*, *Sardh*, and *CG5955* using *DJ694* driver. Fig S3f.) Flight performance upon muscle-specific *KD* of *Gnmt*, *Sardh*, and *CG5955* using *Act88F-GS* driver. Fig. 4g) Flight performance upon IFM-specific *KD* of *Dgat2* using *DJ694* driver. Fig. S3g) Overexpression of *Gnmt* helps retain flight index in aging flies shown in 5-week-old female flies. Fig. S4c) Overexpression of *hDgat2* led to impairment of flight index in 3-week-old flies.

2, Relatedly, the authors should always show, side-by-side, that whether the manipulation (of any genes) influences the muscle function in the normal condition or it abolishes TRF-mediated improvement. For example, in Fig. 2b-d, *Gnmt* was upregulated by TRF, and the knockdown abolished the change of the flight index by TRF. However, there is no control here. The data would be better to present as in Fig. S6c. Actually, the authors did not control the genetic background of the lines. The proper control might be required, especially when the basal Flight index can be different. For example, in Fig. S6c, *Nmdmc*-RNAi has higher Flight index at the basal line and this does not further increase by TRF. This would lead to a different conclusion.

Response: Thank you for pointing this out. We have added the proper control in Fig. 2b (now Fig.2h) side-by-side. While performing RNAi knockdown experiments, we have used two independent controls. 1) We crossed control RNAi lines with *Act88F-Gal4* and used *Act88F>Ctrl* RNAi progeny as the control (plotted side-by-side with gene *KD* in the revised manuscript). 2) We also tested *Act88F/+* (progeny of *Act88F-Gal4* with *W¹¹¹⁸*), and similar flight indexes were observed between *Act88F>Ctrl* RNAi and *Act88f/+* (Fig S3b). Flight indexes are now presented as an average of independent RNAi lines; for example, cohorts from two independent control RNAi lines are indicated as symbol circles or triangles. Flight index shows significant flight reduction upon knockdown of commonly upregulated genes,

demonstrating their role in muscle function, and is seen throughout 1, 3, and 5 weeks of age in female flies (Fig. 2d).

We have shown that *KD* of *Gnmt*, *Sardh*, and *CG5955* using *Act88F* driver lost TRF mediated benefits, while TRF improved muscle performance in control flies at 3 and 5 weeks of age (Fig. 2h). For the flight index, each cohort of 10-20 flies is now individually plotted as a circle in all figures. Regarding figure S6c. there was an error made in plotting this figure. We attempted to compare flight ability upon knockdown of *Gnmt* and *Nmdmc* under HFD-ALF and HFD-TRF, however, we mistakenly plotted flight index from *Act88F>Nmdmc-RNAi* under standard diet. We apologize for this confusion. We have corrected Fig.S6c (now Fig.S7e), which contains all lines under a high-fat diet condition. The data now demonstrates that both *Nmdmc* and *AdSL* *KD* flies under HFD-TRF have no improvement in muscle performance compared to HFD-ALF.

Fig. 2d) Flight performance of female flies with IFM-specific *KD* of *Gnmt*, *Sardh* or *CG5955* using *Act88F* driver at 1, 3, and 5 weeks of age. Fig. 2h) Flight index showing loss of TRF-mediated benefits upon knockdown of common upregulated genes using *Act88F-Gal4* driver in 3- and 5-week-old female flies. Fig. S3b) Flight index of *Act88F/+* compared to *Act88F>* two independent control RNAi lines showed no difference. Fig. S7e) The corrected flight index of 3-week-old female flies with IFM-specific *KD* of *Nmdmc* and *AdSL* using *Act88F* driver under HFD-ALF and HFD-TRF.

3, While the transcriptional change is clear, the authors did not address the upstream pathways. How and which transcription factors are involved in up- or down-regulation of the genes by obesity and TRF?

Response: We appreciate the reviewer's comment regarding upstream pathways. Previous findings have found two possible upstream transcription factors which potentially modulate GNMT activities: cAMP-

regulated transcription coactivator (CRTC) and Forkhead Box O (FOXO). We examined their gene expression from our transcriptome data. Interestingly, we found that *Crtc* expression levels were increased modestly under TRF (4 out of 6 time points) in WT and HFD flies, while *FoxO* expression level was significantly increased under TRF (5 out of 6 time points) in *Sk2* flies (Fig. S3k). It has been shown that CRTC promotes gene expression associated with 1C metabolism pathway (and *Gnmt/Sardh*) and purine cycles leading to energy balance (Wang et al, 2021, PMID: 33723074). As we did not observe increases of purine cycle-related genes with the same magnitude in *Sk2*-TRF flies compared to HFD-TRF flies, one possibility is that CRTC promotes *Gnmt* expression under HFD-TRF, but not under *Sk2*-TRF. FOXO isoforms 1/3 have been found to play a role in regulating muscle energy homeostasis through control of glycolytic flux and mitochondrial metabolism. FOXO can be activated by a multitude of factors such as oxidative stress and inflammation, which in turn activates *Gnmt*, which depletes S-adenosylmethionine (Obata et.al, 2014, PMID: 24746817), leading to AMPK activation (Yalcin et al, 2008, PMID: 18424439). Furthermore, AMPK signaling can preserve FOXO signaling potentially leading to sustained FOXO activation (Sanchez et al, 2012, PMID: 22006269). This study did not explore immune response and may need further investigation on what may have led to FOXO activation in muscle. However, this at least gives a direction of transcription factors, which may be involved in TRF-mediated *Gnmt* expression changes in muscle under HFD and *Sk2*.

S3k

Fig S3k) mRNA expression level (normalized read count, log₂) of *Crtc* and *Foxo* under ALF and TRF in WT, HFD and *Sk2* flies.

4, The authors highlight several metabolic enzymes/pathways. However, unfortunately, no metabolic analysis has been done. It is possible that the gene expression change is due to the adaptation of metabolic alteration. So looking at mRNA level is usually not enough to discuss whether it is a cause or a consequence. In line with this, the authors should analyse whether IMP and related purine/THF

metabolites are increased by feeding folic acid. The phenotypes in Fig. 4g-h are beautifully demonstrated but only measuring ATP would not tell how metabolism is affected.

Response: We thank the reviewer for this suggestion. We analyzed untargeted metabolites under ALF and TRF from IFMs in WT, HFD, and *Sk2* flies by the Targeted Metabolomics Proteomics Laboratory (TMPL) at UAB. Metabolite analyses showed alteration of important metabolites related to the purine cycle, which support our transcriptomic data and functional data obtained with feeding folic acid. As shown in Fig. 5f, we found higher amounts of inosine, hypoxanthine, and xanthine under HFD-ALF than HFD-TRF. As inosine, hypoxanthine, and xanthine are products of ATP catabolism within the purine cycle (Farthing et al, 2015, PMID: 25956679), this may lead to a greater reduction of ATP levels under HFD-ALF.

Interestingly, under HFD-TRF, we found increased levels of fumaric acid, an intermediary product of the purine cycle. Fumaric acid can enter the TCA cycle and subsequently be converted to malic acid (also increased) which follows a path toward generating ATP. Overall, these metabolomic results support our hypothesis that TRF upregulated the purine cycle pathway, which generated more ATP compared to ALF in HFD flies.

We also analyzed metabolites in the *Sk2* model in which we found TRF-mediated increases in NAD/NADH, which is necessary for TCA/ETC; Citrate/malic acid, acetylcarnitine, and L-carnitine, which are important for TCA cycle were also increased (Schroeder et al, 2012, PMID: 22238215). Furthermore, we found that melezitose and melibiose (di- and tri-saccharide) were both decreased in *Sk2*-TRF (Fig. 6g), suggesting activation of glycogenolysis. Analyses of additional metabolites are included in (Fig. S9-11). Though these results are in alignment with purine and AMPK involvement under TRF conditions, these results are limited in their ability to assess dynamic levels of metabolites. These results show only a snapshot rather than temporal changes which would require more sophisticated methods such as utilization of C13 labelling.

Fig. 5f) Metabolites (under HFD-TRF versus HFD-ALF; Fold change ≥ 1.25) associated with purine cycle. Metabolites downregulated are associated with ATP breakdown products; upregulated metabolites are associated with ATP production through purine cycle and TCA. Fig. 6g) metabolites (under *Sk2*-TRF versus *Sk2*-ALF; Fold change ≥ 1.25) associated with AMPK signaling pathway. Decreased metabolites are di- and trisaccharides, which are potentially broken down in glycogen metabolism. Increased metabolites are related to TCA and ETC pathways.

Minor points:

5, It is not elucidated whether increased lipid in muscle is a cause of the phenotype in Fig. 2e-g. The authors may test *Gnmt*-, *Sardh*-, or *CG5955*-RNAi together with *DGAT2*-RNAi to see whether it rescues the decreased flight index.

Response: To the best of our knowledge, the interplay between *Gnmt*, *Sardh*, *CG5955* and *Dgat2* is currently unknown. We believe that it is likely that both lipid accumulation in muscle and ATP levels play a role in muscle performance. We performed *Gnmt*, *Sardh* or *CG5955* KD along with *Dgat2* KD, and no significant differences were observed in flight performance. This will need to be further studied in the future.

6, Fig. 2b is not mentioned in the text.

Response: Figure 2b has now been included in the text.

7, Fig. S4c,d in the text is actually Fig. S6c,d.

Response: Thank you, this has now been corrected and properly cited in the manuscript.

8, Is food intake affected by the gene manipulations, e.g. *Gnmt*-RNAi?

Response: We haven't noticed any differences in food intake with any RNAi driven by *Act88F* driver, compared to control. More importantly, IFM-specific knock-down of any genes was not associated with food intake.

Reviewer #2 (Remarks to the authors):

Livelo and colleagues are investigating how time restricted feeding (TRF) of different diets does impact muscle function during adult age in a *Drosophila* model. The authors feed flies for only 12h per day compared to ad libitum feeding and age them, while monitoring muscle function by performing a flight assay.

How TRF positively affects organism health in general or individual organs in particular during ageing is an exciting field of study, which still lacks mechanistic insight, specifically in the muscle.

To unveil a mechanism, the authors performed transcriptomics analysis of entire thoraces (not indirect flight muscles, as stated incorrectly in the abstract) and identified some metabolic enzymes displaying changed expression upon TRF on various diets or genetics models.

Interestingly, knock-down of one of them, *Dgat2*, appears to result in TRF-like improvement of flight muscle function during ageing despite ad libitum feeding.

In general, the paper appears to superficially follow various candidate genes that are not obviously related. Hence, it has an unclear title and a not very specific abstract. Many of the data presented appear preliminary of results are often over-interpreted.

I am mainly commenting on the transcriptomics and genetic studies of the paper.

1. The authors used entire thoraces as source for the transcriptomics study. However, in the results and in the abstract the authors claim to have collected indirect flight muscles (IFMs). While the thorax is

largely consisting of IFMs it contains various other muscles and digestive organs, hence this statement is incorrect.

Response: Thank you for pointing out this important issue. This error/inconsistency of using IFMs has been addressed for the transcriptomic study and other assays. We now have specified in Fig. 1b legend and method section that IFMs have been used in our transcriptomic study. In addition, IFMs were also used for qRT-PCR validation of TRF-regulated genes and metabolomic study.

2. The authors have isolated RNA samples every 4 hours over the course of one day, meaning 6 samples for each of the various feeding conditions. These samples were pooled to perform differential expression analysis using DE-Seq2, which requires replicates. The authors have not collected true replicates for any of the samples to verify the validity of this simplification procedure. As expression for many genes cycles during the day, this simplification is questionable. Similarly worrying is that the Suppl Table 1 is missing several of the samples entirely. Together with the low threshold of a FC 1.25 (not log₂FC, which is standard) chosen by the authors, the data presented in Figure 1 are at least questionable and would require more detailed verifications for each presented gene. As only 8 genes were followed up, this verification would be doable by qPCR or tagged protein quantification.

Response: This is a reasonable question raised by the reviewer. We identified two samples that were outliers (see methods). Therefore, we removed them from downstream analysis and Supplementary Table 1. In the revised manuscript, the two outliers are indicated in Fig. S1a with a red dash box. The two outliers are now included in Supplementary Table 1 and marked as outliers in the column name.

In general, fold-change ≥ 1.25 and $p \leq 0.05$ may not have been used conventionally in most of the transcriptomic studies, however, TRF mediated fold changes are modest compared to other modulations, such as starvation, diet restriction, etc (see our previous study, Gill et al., 2015, PMID: 25766238, transcripts showed a similar magnitude between ALF and TRF from heart, head, and periphery of 5-week-old flies.). Moreover, fold-change ≥ 1.25 has been used in multiple circadian studies and showed valuable findings (Fonseca Costa et al., 2017, PMID: 28068403; Ma et al., 2021, PMID: 33438579). Additionally, a recent human TRF study on skeletal muscle (Lundell et al., 2020, PMID: 32938935) also reported gene expression changes of similar magnitude. Here, the threshold of fold-change ≥ 1.25 was chosen in our manuscript to identify candidate genes regulated by TRF in both WT and obesity model flies. This threshold led to the identification of 8 candidate genes (5 up and 3 down), with most of the fold changes ≥ 1.5 (Fig. S2a). Within the 5 up genes, *Gnmt*, *Sardh*, and *CG5955* appear to have functional associations, sharing a role in glycine utilization and production. Altogether, the fold-change ≥ 1.25 threshold in our transcriptome data allowed us to identify generic TRF-mediated gene changes with functional association in WT and obesity model flies.

In agreement with the reviewer on further validation, we have performed qRT-PCR to validate the temporal expression of the 7 common DEGs (*CG13992* has low expression and can not be detected robustly using qRT-PCR). As shown in Fig. S2b-c, increased mRNA levels of *Gnmt*, *Sardh*, *CG5955*, *CG6806*, and *CG5896* were observed in at least 3 out of 4 time points in all TRF conditions, demonstrating similar trends as those seen in the RNA-seq data. (Fig. S2b for RNA seq; Fig. S2c for qRT-PCR). *Dgat2* and *CG7997* also showed similar trends, with most time points exhibiting reduced expression in all TRF conditions resembling the same trend seen in RNA-seq.

Fig S2b) Temporal expression level (normalized read count, log₂) of TRF-mediated upregulated and downregulated genes under ALF and TRF in WT, HFD and *Sk2* flies. Fig S2c) qRT-PCR validation of temporal expression of genes from Fig S2b.

3. The authors test flight muscle function in a quite tedious version of the flight assay that can identify small differences between the different conditions following flight muscle specific gene knock-down (Figure 2). This reviewer would like to see an individual results table to verify the statistics. However, neither the number of flies tested in the various conditions, nor their individual performance was provided in the Supplementary Tables. It is also unclear how often this experiment was repeated.

It is well known that high GAL4 expression levels present in Act88F-GAL4 can impact flight. Was the 'wild type' control containing Act88F-GAL4? This is unclear from the figure legends or methods.

Response: In agreement with the reviewer, we have added more details regarding the flight muscle assay methodology and results in the method section and supplementary table of the manuscript. Previously, we plotted the mean flight indices per condition using the total # of flies with no indication of cohorts, N# of cohorts, total N#. This has now been changed to include each cohort fly muscle performance using 10-20 flies per cohort (shown as a circle, triangle or square if more than one RNAi line) and mean was calculated from all cohorts for each condition. All three parameters regarding the individual cohort performance, the number of cohorts and the number of total flies have been included in an individual results table (Source data). Moreover as indicated under reviewer 1 comment #2, while performing RNAi knockdown experiments, we have used two independent control RNAi lines crossed with *Act88F-Gal4* and used *Act88F>ctrl* RNAi progeny as the control. Additionally, we have also tested *Act88F/+*, and similar flight index were observed between *Act88F>Ctrl* RNAi and *Act88F/+* in female progeny (Fig S3b).

Fig. S3b) Flight index of *Act88F/+* compared to *Act88F>ctrl* RNAi #1 and *Act88F>ctrl* RNAi #2 at 1, 3, 5, and 7 weeks.

4. How does *Gnmt*, *Sardh* and *CG5955* function relate to TRF or different diets? Why is *CG5955* called *TdH* only in Figure 2g?

Response: Under HFD, *Gnmt*, *Sardh* and *CG5955* KD flies (left panel) showed lower flight ability compared to their flight ability under normal diet (right panel). Moreover, HFD flies with *Gnmt*, *Sardh* and *CG5955* KD did not show TRF-mediated benefits (left panel), while age-matched control flies showed improved flight ability under TRF (left panel). Also, in the revised manuscript we have corrected *TdH* with *CG5955* for consistency.

Left panel) Flight performance of 3-week-old female flies with IFM-specific KD of *Gnmtd*, *Sardhd* and *CG5955* using *Act88F* driver under HFD-ALF and HFD-TRF. Right panel) Flight performance of 3- and 5-week-old female flies with IFM-specific KD of *Gnmtd*, *Sardhd* and *CG5955* using *Act88F* driver under standard diet ALF and TRF.

5. A potential beneficiary role of *Dgat2* knock-down in IFMs is interesting. It would be important to verify this result by either a second RNAi line or some other genetic tools to rule out off-target effects (Figure 3).

Response: We thank the reviewer for their interest in this finding. As addressed in reviewer #3's 4th comment, we have included two RNAi lines for *Dgat2* knockdown using *DJ694* driver. *Dgat2* mRNA expression upon KD using *Act88F* and *DJ694* driver are shown in Fig. S4a-b with ~75% and ~60% reduction respectively. We found muscle improvements induced by *Dgat2* KD using both drivers were observed at week 5 and 7 of age (Fig. 4c, g). Overall, we were able to support the important role of *Dgat2* in skeletal muscle using two independent drivers (*Act88F* and *DJ694*) demonstrated by the protective effects on flight performance in old age (5 and 7 weeks). Additionally, we have employed an overexpression of human *Dgat2* using *Act88F* driver under ALF and have shown impaired flight performance (Fig. S4c) in addition to lipid increases in the first week of age (Fig. S4d-f), indicating potential human translatability in addition to providing evidence for *Dgat2*'s role in muscle performance.

We have also investigated the paralogs for *Dgat2* (*CG1941* and *CG1946*) with two independent RNAi line KDs using *DJ694* driver. Their transcription levels remained unchanged under TRF, except *CG1946* was upregulated under *Sk2*-TRF (Fig. S4g). Upon IFM-specific KD of *CG1941* or *CG1946* using *DJ694* driver, muscle improvement was observed upon *CG1941* KD at week 7 but wasn't as pronounced as the age-matched *Dgat2* KD flies (Fig. S4h).

Fig. S4a-b) Relative mRNA expressions of *Act88F-Gal4* or *DJ694-Gal4* drive KD of *Dgat2* in 3-week-old female flies were quantified with qRT-PCR. Fig. 4c) Flight index showing *Dgat2* KD using *Act88F-Gal4* leads to improved flight performance at 5 and 7 weeks of age. Fig. 4g) Flight index showing similar benefits of *Dgat2* KD via *DJ694* driver. Fig. S4c) Flight performance under *hDgat2* overexpression using *Actin88F-Gal4* at 1 week of age. Fig. S4d) lipid stain using Nile red showing moderate increases in lipid droplet size and density under *hDgat2* overexpression. Fig S4e-f) Quantification of lipid droplet area and lipid droplet density. Fig. S4g) Expression levels of *CG1941* and *CG1946*. Fig. S4h) Flight performance of 1-, 3-, 5- and 7-week-old female flies with *DJ694-Gal4* drive KD of *Dgat2* predicted paralogs *CG1941* and *CG1946*.

6. Also Figure 4 is missing second RNAi lines for all of the genes shown.

Response: Second RNAi lines are now added to Fig. 4f (now Fig. 5e). The two RNAi lines' results were indicated by either circles or triangles. Similar flight performance from the two independent RNAi lines were observed for *Gnmt* and *Nmdmc* (Fig.2d and 5e), therefore, we performed folic acid supplementation on IFM-specific knockdown using one RNAi line (Fig. 5g-h).

7. The authors find that AMPK phosphorylation is reduced in the model of sphingosine kinase 2 mutants (Figure 5). However, there is no difference in AMPK-phospho levels comparing TRF with ad libitum feeding. Hence, what have we learnt about the TRF protecting mechanism?

It is trivial that knock-down of any of the essential glycolysis, TCA or ETC enzymes compromise flight activity. Again, there is no link to the feeding scheme provided.

Response: We appreciate the comments. Our previous protein collection from different conditions was not controlled for time, we have re-collected protein samples from WT, HFD, and *Sk2* ALF and TRF flies at for one time point (ZT9). While AMPK α protein levels were unchanged under TRF compared to the ALF counterparts (Fig. S8i, j), increased p-AMPK α level was observed under *Sk2*-TRF versus *Sk2*-ALF (Fig. 6h, i, and Fig. S8h), which implies the potential upregulation of TRF-mediated AMPK activation, especially in *Sk2* flies. To fully evaluate the protein levels of AMPK α under TRF in comparison to ALF, we collected thorax protein samples from *Sk2* flies at ZT3, 9, 15, and 21 (Fig. 6j, k). We found increased phosphorylated-AMPK α in *Sk2*-TRF compared to *Sk2*-ALF. Altogether, these results support our initial claim that TRF activates AMPK-associated pathways in *Sk2* flies.

Our transcriptomic data showed that genes associated with glycolysis, glycogen metabolism, TCA, and ETC pathways were upregulated under *Sk2*-TRF. Given TRF results in muscle improvement, we first tested *KD* of these genes to validate their roles in muscle function. We found muscle function decline upon the *KD* of genes associated with these pathways (Fig. 6e, f). Therefore, increasing of glycolysis, glycogen metabolism, TCA, and ETC activity may account for at least a part of the beneficial effect of TRF. It is likely that not all enzymes within these pathways are essential for muscle functions. For example, in our previous study on *Drosophila* heart, we found TRF-mediated downregulation of genes associated with ETC. However, *KD* of 3 genes of ETC complex I components in heart showed cardioprotection, while *KD* of 1 gene showed no effect on heart function (Gill et al., 2015, PMID: 25766238). To further test the roles of these genes in *Sk2*-TRF mediated benefits, knockdown of *mAcon1* or *GlyS* accompanied with *Sk2* was made and we measured their flight performance under ALF and TRF (Fig. S8f). While TRF led to muscle improvements on *mAcon1* *KD* flies, TRF failed to improve muscle performance on flies with *mAcon1* and *Sk2* double *KD*. Moreover, while *GlyS* *KD* flies didn't respond to TRF, *GlyS* and *Sk2* double *KD* flies showed a further decline on muscle performance under TRF. Altogether, these results suggest that increasing basal activity of these AMPK associated pathways support TRF-mediated muscle benefits in *Sk2* flies.

Fig. 6h) Representative western blot of p-AMPKα levels (top) and α-TUBULIN (bottom), from 3-week-old female fly IFMs in *Sk2* TRF and *Sk2* ALF flies. Fig. 6i) Ratios of p-AMPKα/ α-TUBULIN (normalized to trough of ALF value). Fig. 6j, k) p-AMPKα protein levels at ZT3, 9, 15, and 21 in *Sk2*-ALF and *Sk2*-TRF flies. Fig. S8h) Flight index of 3-week-old flies with indicated genotypes.

Reviewer #3 (Remarks to the authors):

Livelo et al. present a mechanistic approach to explain the benefits of time-restricted feeding (TRF) to skeletal muscle function in obesity using the *Drosophila* model organism. To this aim they apply a comparative circadian transcriptome analysis between skeletal muscle-enriched thorax material from wild type (WT) flies and diet- (DIO) or genetically-induced (GIO) fly obesity models under TRF or ad libitum fed (ALF) conditions.

By in silico pathway analysis the authors identify three genes involved in glycine production/usage to be up-regulated under TRF in all three models. Tissue-specific knockdown of all three individual genes impairs flight performance, blunts the TRF benefits in this assay and promotes structural muscle degeneration and ectopic lipid accumulation in aging fly indirect flight muscle.

Along these lines the authors identify fly *Dgat2* as the only gene specifically down-regulated in all models under TRF vs. ALF. Flight muscle-specific knockdown of this gene improved flight performance and suppressed lipid accumulation under ALF in aging WT flies.

In the following the authors use the same strategy to identify TRF-dependent transcriptional pathways specific for their HFD DIO and their *Sk2* mutant GIO model, respectively.

For the DIO model, they propose that TRF operates via purin biosynthesis and folate cycle to potentially improve muscle ATP availability. In support of this interpretation, they demonstrate that folic acid supplementation reverses some effects of HFD diet on physical performance and ATP levels in flies with genetically impaired folate cycle.

Similarly, TRF up-regulates the AMPKalpha gene and a broad range of genes in AMPK downstream

signaling pathways to improve ATP availability in the Sk2 mutant GIO model. In support of the functional implication of these transcriptional regulations, tissue-specific knockdown of some of these genes selectively impaired physiological performance in an age-dependent manner.

In general, this is an interesting study which targets the mechanism(s) of the evolutionarily conserved health benefits of TRF in obesity. The comparative transcriptome analysis followed by gene ontology and reactome analyses is straightforward and provides a wealth of data on candidate genes. Comparing the TRF-response of different genotypes and diet conditions is an elegant way to focus on important pathways.

However, the functional data of this study are much less convincing due to a lack of metabolite analyses in support of the physiological consequences of the described gene regulations. Also, compensatory gene regulations by RNAi-mediated gene knockdown are poorly controlled leaving it largely open whether they recapitulate TRF-dependent physiological phenotypes or more fundamental impairments of the underlying metabolic pathways. The *Dgat2* analysis lacks a compensatory gain-of-function approach.

This study would benefit from focusing on the more extended shared pathway analysis. Functional analysis of the individual DIO and GIO models is not very elaborate and suffers from the question how representative HFD and global Sk2 mutants are for DIO and GIO, respectively.

Major points:

1) Transcriptional fold-changes of ≥ 1.25 appear fairly subtle. In particular, compared to the impact of tissue-specific gene knockdown used for functional interventions. Flightless flies caused by some of the RNAi knockdowns suggest a strong overcompensation of the TRF-induced up-regulation of the genes of interest or even developmental effects. The authors have to take measures and demonstrate that the knockdown is timely and compensates but not overcompensates the TRF-induced up-regulation.

Response: As indicated from our response to reviewer 2 comment #2, in general, fold-change ≥ 1.25 and $p \leq 0.05$ may not be used conventionally in most transcriptomics studies, however, TRF mediated fold changes are modest compared to other modulations, such as starvation, diet restriction, etc (see our previous study, Gill et al., 2015, PMID: 25766238, transcript showed fold change of similar magnitude between ALF and TRF from heart, head and periphery of 5-week-old flies.). Moreover, fold-change ≥ 1.25 has been used in multiple circadian studies and showed valuable findings (Fonseca Costa et al., 2017; Ma et al., 2021). Additionally, a recent human TRF study on skeletal muscle (Lundell et al., 2020, PMID: 32938935) also reported gene expression changes of similar magnitude. Here, the threshold of fold-change ≥ 1.25 was chosen in our manuscript to identify candidate genes regulated by TRF in both WT and obesity model flies. This threshold led to the identification of 8 candidate genes (5 up and 3 down) with most of the fold changes ≥ 1.5 (Fig. S2a). Within the 5 up genes, *Gnmt*, *Sardh*, and *CG5955* appear to have functional association, sharing a role in glycine utilization and production. Altogether, the fold-change ≥ 1.25 threshold in our transcriptome data allowed us to identify generic TRF-mediated gene changes with a functional association in WT and obesity model flies.

We now have also incorporated IFM-specific knockdown with *DJ694-Gal4* driver (Bryantsev et al, 2012, PMID: 22008792), which expresses only in adults within the indirect flight muscle post eclosion. We observed a less severe but still significant reduction in-flight performance at week 3 with *DJ694* driver (Fig. 3a). We validated KD levels at week 3 using qRT-PCR and showed $\sim 50\%$ reduction in mRNA

levels of *Gnmt*, *Sardh*, and *CG5955* (Fig. S3d), which should not overcompensate the TRF-induced up-regulation. Similarly, muscle-specific knockdown of *Gnmt*, *Sardh*, and *CG5955* using *DJ694* driver abolished TRF-mediated benefits on muscle performance (Fig. 3e).

Fig. 2d) Flight performance upon muscle-specific KD of *Gnmt*, *Sardh*, and *CG5955* using *Act88F* driver. Fig. 3a) Flight performance upon muscle-specific KD of *Gnmt*, *Sardh*, and *CG5955* using *DJ694* driver. Fig. 3e) Muscle-specific knockdown of *Gnmt*, *Sardh* and *CG5955* using *DJ694* driver abolished TRF-mediated benefits on muscle.

2) Does the glycine pool change as predicted in response to the transcriptional changes?

Response: We have measured the glycine levels between WT and obesity models and between ALF and TRF using a glycine kit (Glycine is not detected through our metabolite analyses due to its chemistry). When comparing HFD-ALF to WT-ALF, there was a significant reduction in overall glycine levels (Fig. 2c). Referring to Fig. 2b, *Sardh* expression under HFD-ALF seems to be lower providing a potential explanation for this reduction in glycine level compared to WT and additionally, *Gnmt* was also higher potentially leading to greater utilization of glycine. We found that HFD-TRF only induced a non-significant increase in glycine levels compared to HFD-ALF, which may be due to activation of purine cycle, as purine cycle is known to consume glycine (Fig. 2c, and Diagram reference from Zhao et al, 2015, PMID: 25605736). When comparing *Sk2*-ALF to WT-ALF, we found a significantly increased amount of glycine (Fig. 2c), which is likely because of increased levels of *Sardh* and *CG5955* under *Sk2*-ALF versus WT-ALF (Fig. 2b). A further increase on glycine level was observed under TRF in *Sk2* flies, which is likely due to the TRF-induced increases on *Sardh* and *CG5955* levels (Fig. 2c). These increases in glycine levels in *Sk2*-TRF are likely needed in order for GNMT to regulate *S-adenosylmethionine* levels and subsequently lead towards AMPK signaling activation (Zubiete-Franco et al, 2016, PMID: 26394163).

Fig. 2c)

Fig. 2b)

Diagram Reference Zhao et al, 2015, PMID: 25605736)

Fig. 2c) Relative glycine measurements of WT, HFD and *Sk2* under ALF and TRF. Fig.2b) Expression level (normalized read count, log₂) of glycine producers *Sardh* and *CG5955* seem to be increased in ALF conditions in *Sk2* compared to WT possibly explaining increase of glycine in *Sk2* model. Diagram reference from Zhao et al, 2015 PMID: 25605736) showing that glycine is consumed in activation of purine cycle potentially masking TRF induced increases in production of glycine under HFD.

3) Several aspects of Fig. 2d-g are questionable. *Sardh* and *CG5966* RNAi have very different impact on flight performance but muscle degeneration and ectopic lipid storage are comparable (and differ from the control). This challenges the view that we are looking at mechanistic rather than correlative changes. Moreover, how does a glycine pool reduction cause TAG accumulation?

Response: This is an important question raised by the reviewer. It is true that both KD in *Sardh* and *CG5955* have impaired flight performance with *CG5955* KD being the more severe of the two using *Act88F*-Gal4 driver (Fig. 2d) despite similar lipid deposition levels (Fig. 2f and g). This disparity in flight impairment was not observed under the *DJ694* driver between *Sardh* and *CG5955* KD (Fig. 3a). This indicates a potential developmental effect associated with the *CG5955* KD under *Act88F*-Gal4. Interestingly, *CG5955* is known to function as a threonine dehydrogenase leading to the regulation of threonine levels, which subsequently produces glycine. Literature suggests that excess or absence of dietary threonine during development can reduce protein synthesis in skeletal muscle seen in pigs (Wang et al, 2007, PMID: 33723074). This may suggest that the developmental defect of *CG5955* KD causes severe muscle impairment as seen in Fig. 2d. Developmental defects may also extend to lipid metabolism as lipid droplet density was higher in *Act88F* compared to *DJ694* in *CG5955* (Fig. 2e-g and Fig. 3b-d). *CG5955*'s role in the literature extends towards having roles mediating the TCA cycle shown in a diagram reference (Tang et al, 2021, PMID: 34444752). This may suggest that early defects in the TCA cycle could lead to greater sustained impairments in lipid metabolism even throughout adulthood (Diagram reference from Tang et al, 2021 PMID: 34444752). Not much has been explored regarding this and will need further investigation to answer this question.

Regarding glycine levels and TAG accumulation, recent reports demonstrate that circulating glycine levels are associated with a favorable lipid and inflammatory plasma profile (higher HDL-cholesterol and apolipoprotein A1, lower triglycerides, apolipoprotein B and C-reactive protein).

Mechanisms surrounding glycine and lipid accumulation however are not well elucidated currently (Rom et al, 2018, *Curr Opin Lipidol.*, PMID: 30153136). Our data suggest that increased levels of glycine may potentially lead to reduced overall lipid levels under *Sk2*-TRF compared to *Sk2*-ALF, however further insight regarding increases of glycine levels in *Sk2*-ALF compared to WT-ALF will need further study.

Diagram Reference Tang et al, 2021, PMID: 34444752)

Fig. 2d) Flight index of *CG5955* KD compared with *Sardh* KD shows disparity even throughout 3 and 5 weeks of age under *Act88F*-Gal4. Fig. 3a) Flight index of *CG5955* KD compared with *Sardh* KD shows similar flight indices compared to those observed in *Act88F*-Gal4 indicating a potential developmental associated effect. Fig. 2e) shows representative lipid staining with Nile red under *Act88F*-Gal4. Fig. 2f) shows relative lipid droplet size under *Act88F*-Gal4 in 3 weeks of age. Fig. 2g) Shows relative lipid droplet density under *Act88F*-Gal4 in 3 weeks of age. Fig. 3b) Shows representative lipid staining with Nile red under *DJ694* driver. Fig. 3c) Shows relative lipid droplet size under *DJ694* in 3 weeks of age with similar trends as seen in *Act88F* driver. Fig. 3d) Shows notable changes in lipid droplet density under *DJ694* in 3 weeks of age compared to *Act88F* suggesting developmental effects leading to impaired lipid metabolism causing higher density in *Act88F* driver. Diagram reference Tang et al, 2021, PMID: 34444752) Reference showing how “TDH” (*CG5955* in *Drosophila*) leads to activation of TCA. Due to this role, developmental defects in *CG5955* may lead towards impairment of lipid metabolism throughout adults.

4) *Dgat2* is down-regulated under TRF. The authors need to demonstrate that TRF benefits (partially) vanish in response to compensatory *Dgat2* overexpression adjusting the gene expression to *Dgat2* levels under ALF. The functional *Dgat2* data rely on a single RNAi construct (knockdown efficiency?), which is weak evidence. In particular, as *Dgat2* is one of three paralogous genes of the *Dgat2* family in flies.

Response: As responded for reviewer #2 comment #5, we now have two RNAi lines for *Dgat2* knockdown using *DJ694* driver. *Dgat2* mRNA expression upon KD using *Act88F* and *DJ694* driver are shown in Fig. S4a-b with ~75% and ~60% reduction respectively. Muscle improvements induced by *Dgat2* KD using both drivers were observed at week 5 and 7 of age (Fig. 4c, g). We were able to support the important role of *Dgat2* in skeletal muscle using two independent drivers (*Act88F* and *DJ694*) demonstrated by the protective effects on flight performance in old age (5 and 7 weeks). Additionally, we have employed an overexpression of human *Dgat2* using *Act88F* driver under ALF and have shown impaired flight performance (Fig. S4c) in addition to lipid increases in the first week of age (Fig. S4d-f), indicating potential human translatability in addition to providing evidence for *Dgat2*'s role in muscle performance.

The paralogs for *Dgat2* (*CG1941* and *CG1946*) now have been tested with two independent RNAi lines KD using *DJ694* driver. Their transcription levels remained unchanged under TRF, except *CG1946* was upregulated under *Sk2*-TRF (Fig. S4g). Upon IFM-specific KD of *CG1941* or *CG1946* using *DJ694* driver, muscle improvement was observed upon *CG1941* KD at week 7 but wasn't as pronounced as the age-matched *Dgat2* KD flies (Fig. S4h).

Fig. S4a-b) Relative mRNA expressions of *Act88F-Gal4* or *DJ694-Gal4* drive KD of *Dgat2* in 3-week-old female flies were quantified with qRT-PCR. Fig. 4c) Flight index showing *Dgat2* KD using *Act88F-Gal4* leads to improved flight performance at 5 and 7 weeks of age. Fig. 4g) Flight index showing similar benefits of *Dgat2* KD via *DJ694* driver. Fig. S4c) Flight performance under h*Dgat2* overexpression using *Actin88F-Gal4* at 1 week of age. Fig. S4d) lipid stain using Nile red showing moderate increases in lipid droplet size and density under h*Dgat2* overexpression. Fig S4e-f) Quantification of lipid droplet area and lipid droplet density. Fig. S4g) Expression levels of *CG1941* and *CG1946*. Fig. S4h) Flight performance of 1-, 3-, 5- and 7-week-old female flies with *DJ694-Gal4* drive KD of *Dgat2* predicted paralogs *CG1941* and *CG1946*.

5) Fig. 3d-f is not convincing. Size and abundance of the LDs appear too small for robust conclusions about possible changes. Also, these parameters need to be shown as scatter plots to more readily display the distributions.

Response: Thank you for this suggestion. Indeed, the lipid droplets are small at 3-week-old WT flies and *Dgat2* KD flies. We have performed *Dgat2* KD using both *Act88F* and *DJ694* drivers, and lipid droplet area were reduced with both drivers (Fig. 4d-j), although no changes on lipid density (Fig. 4f, j). Abdomen lipid staining was also performed in fat body to ensure the Nile red quantification of lipids

represent the true physiology (Fig. S5a-c). New representative images have been used for Fig 3d (now Fig. 4d). Lipid droplet parameters are now shown as scatter plots.

Fig. 4d) Fluorescence images of the IFMs from 3-week-old female flies with Act88F-Gal4 drive KD of Ctrl, *Dgat2* upon staining with phalloidin (green) and Nile Red (red puncta). Fig. 4h) *DJ694* drivers. Fig. 4e) Relative droplet area in 3 weeks of age shown as scatter plots in both *Act88F* and Fig 4i) *DJ694*. Fig. 4f) Shows relative lipid droplet density in 3 weeks of age as a scatter plot in *Act88F*-Gal4 and Fig. 4j) *Dj694*.

6) Fig 2c, 3c: WT is an inappropriate control, which does not address driver line effects. Also, in 2c the very same control seems to be shown three times in all subpanels. Are they identical to the data shown in Fig. S4g?

Response: Thank you for pointing this out. While performing RNAi knockdown experiments, we have used two independent control RNAi lines crossed with *Act88F-Gal4* and used *Act88F>ctrl* RNAi progeny as the control. Additionally, we have also tested *Act88F/+*, and similar flight index were observed between *Act88F>Ctrl* RNAi and *Act88f/+* (Fig S3b). We have carefully revised the figures and ensured that controls are correctly described in figures and figure legends.

7) FA supplementation is promising. However, how do transcriptional changes translate into changes in FA and FA-related metabolites? Direct measurements are necessary.

Response: As also asked by reviewer 1, we appreciated the reviewer's suggestion on this. In our ongoing study we are analyzing untargeted metabolites under ALF and TRF from IFMs in WT, HFD and *Sk2* flies, with the aid of the Targeted Metabolomics Proteomics Laboratory at UAB. Our metabolome analyses showed alteration of important metabolites related to purine cycle, which support our transcriptomic

data as well as functional data obtained with feeding folic acid. As shown in Fig 5f, we found higher amounts of inosine, hypoxanthine and xanthine under HFD-ALF compared to HFD-TRF. As inosine, hypoxanthine and xanthine are products of ATP catabolism within the purine cycle (Farthing et al, 2015, PMID: 25956679) this may lead to greater reduction of ATP levels. Interestingly, under HFD-TRF, we found increased levels of fumaric acid, an intermediary product of the purine cycle. It is known that fumaric acid can enter in the TCA cycle and subsequently converted to malic acid (also increased) which follows a path towards generating ATP. Overall, these metabolomic results support our hypothesis that TRF upregulated purine biosynthesis pathway which generated more ATP compared to ALF in HFD flies. Though these results are in alignment with purine involvement under TRF conditions, these results are limited in their ability to assess dynamic levels of metabolites. These results show only a snapshot rather than temporal changes which would require more sophisticated methods such as utilization of C13 labelling.

Diagram Reference Ichida et al, 2012, PMID: 23203137)

A) Diagram reference from Ichida et al, 2012, showing purine cycle salvage pathway and ATP production; metabolites from ATP breakdown shown in red circle, ADP accumulation potentially indicating direction towards ATP production circled in green. Fig. 5f) Metabolite analysis from HFD-TRF compared to HFD-ALF. This supports that under HFD-TRF, inosine, hypoxanthine and xanthine are reduced, while fumaric acid and malic acid are increased. These results may indicate greater ATP production as fumaric intermediate from purine can funnel into TCA. Levels of ADP may indicate that cycles move towards the direction of ATP production from AMP. (ATP metabolites are not detected due to their instability in columns).

8) The flight index data of "Ctrl RNAi" in Fig. 4f and "Ctrl" in Fig. 5h are remarkable similar. Given that they are independent, why do "WT" flight indices vary so much between Fig. 2c or 3c and Fig. S6f?

Response: We have now plotted flight index as individual points that each represents a cohort of 10-20 flies, moreover, data from second RNAi lines have been added to Fig. 4f and Fig. 5h (now Fig. 5e and Fig.

6e-f). Parameters regarding the individual cohort performance, the number of cohorts, and the number of total flies have been included in an individual results table (Source Data). We thank the reviewer for this concern and would like to clarify that the control in Fig. 2c and 3c are *Act88F*> Ctrl RNAi flies, while Fig. S6f (now Fig. S7f) WT are Canton S flies. During data collection for flight index, the control group and experimental group were always handled by the same user within a close time period. Therefore, we believe our data have proper controls although there might be batch-to-batch variation.

9) The broad transcriptional up-regulation in GIO in response to TRF is interesting. However, how this translates to metabolic outcome is insufficiently addressed. The pAMPKalpha response is at the significance limit and the individual, strong knockdown of selected genes shown in Fig. 5h presents no conclusive picture. With other words without a complementary metabolite profile the GIO data remain preliminary.

Response: We appreciate this important question brought by the reviewer. We have collected untargeted metabolites under ALF and TRF from IFMs in *Sk2* flies. We analyzed metabolites in the *Sk2* model in which we found TRF-mediated increases in NAD/NADH, which is necessary for TCA/ETC; Citrate/Malic acid, acetylcarnitine, and L-carnitine, which are important for TCA (Schroeder et al, 2012, *Circ Cardiovasc. Imaging*). Furthermore, we found that melezitose and melibiose (di- and tri-saccharide) were both decreased in *Sk2*-TRF (Fig. 6g), suggesting activation of glycogenolysis.

As our previous protein collection from different conditions was not carefully controlled for time, we have re-collected protein samples from WT, HFD, and *Sk2* ALF and TRF flies at the same time (ZT9). While AMPK α protein levels were unchanged under TRF compared to the ALF counterparts (Fig. S8i, j), increased p-AMPK α level was observed under *Sk2*-TRF versus *Sk2*-ALF (Fig. 6h, i, and Fig. S8h), which implies the potential upregulation of TRF-mediated AMPK activation especially in *Sk2* flies. To fully evaluate the protein levels of AMPK α under TRF in comparison to ALF, we collected thorax protein samples from *Sk2* flies at ZT3, 9, 15, and 21 (Fig. 6j, k). We found increased phosphorylated-AMPK α in *Sk2*-TRF compared to *Sk2*-ALF. Altogether, these results support our initial claim that TRF activate AMPK-associated pathways in *Sk2* flies.

Fig. 6h) Representative western blot of p-AMPK α levels (top) and α -TUBULIN (bottom), from 3-week-old female fly IFMs in *Sk2* TRF and *Sk2* ALF flies. Fig. 6i) Ratios of p-AMPK α / α -TUBULIN (normalized to trough of ALF value). Fig. 6j, k) p-AMPK α protein levels at ZT3, 9, 15, and 21 in *Sk2*-ALF and *Sk2*-TRF flies. Fig. 6g) Metabolites (fold change ≥ 1.25 under *Sk2*-TRF versus *Sk2*-ALF) associated with AMPK signaling pathway.

10) Fig. S3b shows identical plots assigned to different genes.

Response: Thank you for pointing this out, we have corrected this plot (Fig. S2b).

11) Fig. S3c: “Ctrl” is “WT” according to text.

Response: This has now been changed in the main text under results section and also seen in the figure legend for S3c

12) Transcriptional regulation of the central genes of interest derived from RNAseq data requires qRT PCR validation.

Response: qPCR validation has now been added (Fig. S2c and S4b)

13) “... IFM-specific RNAi KD of *Gnmt*, *Sardh* and *CG5955* resulted in a cytotoxic effect in the adipose tissue ...” These data need to be presented (in the supplement).

Response: This was a typo and has been corrected to include lipid deposition and not “cytotoxic effects”. We performed adipose tissue only to ensure that Nile red quantification of lipids represented the true physiology. We found no IFM-specific KD effects of these genes in the accumulation of lipid in the adipose tissue (Fig. S5)

Minor points:

14) The title sounds very general referring to DIO and GIO models, while a mild HFD diet and the *Sk2* mutant are single models of unknown representative value for DIO and GIO. Given the wealth of transcriptome data the finding of “shared and unique pathways” sounds trivial. In particular, as it remains open if the unique pathways are unique for HFD diet and *Sk2* mutants or for DIO and GIO.

Response: In agreement with the reviewer, we have changed the title to “Time-restricted feeding attenuates muscle dysfunction through purine cycles and AMPK signaling in Drosophila obesity models” Previously, in the entire manuscript we have used the term “DIO” to refer to a high-fat diet-induced obesity model, however, as the representative value of our high-fat diet model is unknown we have changed this wording to “HFD”. Also, the term “GIO” has also been changed to “Sk2” which was how we previously represented this model in our previous study (Villanueva et al, 2019, PMID: 31221967)

15) Fig. 2f,g uses once CG5955 and once Tdh for the same gene.

Response: “Tdh” has now been changed, this was the human name for the gene which should have been Cg5955. This has now been changed to keep the wording consistent.

16) Some SEMs are missing e.g. Fig. 3c Dgat2 RNAi, Fig. 4h Ctrl RNAi

Response: All of these figures now contain SEM as seen in other figures.

17) Fig 5c Ampk or Ampkalpha? “Gnerate” needs attention.

Response: Ampk α is the proper term and this has now been changed in addition to the correct spelling of “generate”.

18) The Actin88F-Gal4 driver needs more description concerning the developmental time- and tissue-specificity.

Response: Along with the references, we have added more description regarding *Actin88F-Gal4* and have additionally added 2 drivers: *DJ694-Gal4* and *Actin88F-geneswitch* in order to address the developmental effects of using *Actin88F-Gal4*. Tissue specificity description has also been added for more clarity.

19) For RNAseq analysis, which samples were “ground to fine powder in liquid nitrogen” and which tissues were treated with the “Polytron homogenizer”?

Response: All tissues were treated with the polytron homogenizer. The method section has been corrected.

20) The description of the p-AMPKalpha data is misleading as a single of the tested conditions is statistically significant.

Response: As described for reviewer question # 9, we have re-run the assay with different time points and have included this in the figure (Fig. 6k, l).

21) Page 7: Supplementary Fig. 4c and 4d should read Supplementary Fig. 6c and 6d.

Response: Figure orders have been reworked and have been changed to ensure the correct citation of figures. Thank you for noticing this.

22) “Abdomen lipid staining...” Which abdominal tissue has been analyzed in Fig. S4c-e? Fat body? Muscle? The quality of Fig. S4c is low but does not support the quantifications in S4d,e as the density is clearly larger in Dgat2 RNAi while the LD area appears smaller.

Response: We have clarified that abdomen staining is done in the fat body (adipose tissue). New representative images have been used for Fig S4c-e (now Fig S5a-c).

23) Where are the RNAseq data deposited?

Response: The RNA seq data has submitted to Gene Expression Omnibus (GEO) on May 27, 2022, with the following secure token has been created to allow review of record GSE205334 while it remains in private status: Please use the following link to see the RNA seq data deposited to GEO using the token highlighted below.

<https://www.ncbi.nlm.nih.gov/geo/query/acc.cgi?acc=GSE205334>

Ydapckkdbshpip

Reviewer #4 (Remarks to the authors):

NATCOM-21-32964

Time-restricted feeding promotes skeletal muscle function in diet- and genetic-induced obesity through shared and unique pathways

C Livelo, Y Guo, S Varshney, F Abou Daya, HD Le, S Panda & GC Melkani

This is a very interesting study, providing more insight in the molecular mechanisms that may be responsible for the beneficial health effects of time-restricted feeding (TRF). Especially testing the functional importance of the genes identified by the transcriptomics by the muscle specific knock down and following flight assays, is a strong point of this study.

Nevertheless, I have a couple of remarks: At several occasions the authors mention the importance of muscle tissue for glucose metabolism. However, in their Discussion it is not discussed whether the current results provide any insight in how TRF could improve glucose metabolism.

At the beginning of their Introduction the authors mention 3 major contributors to the current obesity epidemic, genetic predisposition, calorie dense diets and chronodisruption. However, in their study they only use a DIO and GIO, but no CIO model. Although, I do not know whether chronodisruption also results in obesity in Drosophila, it would have been nice to discuss this aspect.

Related to this, in rats and mice the TRF strategy has often been used as a model for chronodisruption by restricting food access to the normal sleep phase. Again I do not know whether this works in Drosophila, but it would be nice to know whether the changes in muscle tissue observed in the current study are opposite to those described in the rat and mice studies employing TRF during the sleep phase. I think at least the authors should mention/discuss this aspect.

Response: We greatly appreciate the positive feedback from this reviewer and also for bringing important remarks. In our previous study, we have evaluated the TRF-mediated effects on metabolic parameters including glucose metabolism and insulin resistance (Villanueva et al, 2019, PMID: 31221967). We agree with the importance of discussing glucose metabolism in TRF and have added more insight into the discussion section in paragraph #4. We also mentioned how the purine cycle, namely, *AdSL* is an insulin secretagogue, which may help mediate glucose uptake. Glucose metabolism may be uniquely regulated between HFD and *Sk2* under TRF. For example, in *Sk2*- TRF, AMPK α showed increased expression levels and gene expression upregulation associated with AMPK downstream pathways such as glycolysis. This may suggest that under *Sk2*-TRF that there is greater glucose utilization through glycolysis and glycogen metabolism. These are some of the insights gained from this study that may help explain how glucose metabolism is improved under TRF in the skeletal muscle.

Also, *Drosophila* has been used as a model for chronodisruption, including results from our group shown in a previous study (Villanueva et al, 2019, PMID: 31221967). Previously we have demonstrated the effects of circadian disruption on muscle function with the induction of a light/light (LL) paradigm (well-established chronodisruption model). In addition to muscle dysfunction, we observed ectopic lipid deposition and insulin resistance upon chronodisruption, which was also observed in our obesity models (Villanueva et al, 2019, PMID: 31221967). More interestingly, imposing TRF resulted in attenuated muscle dysfunction, ectopic lipid accumulation, and insulin resistance in the chronodisruption model, similar to obesity models (Villanueva et al, 2019, PMID: 31221967). We are currently under a time-series experiment for collecting transcriptomic data during a 24 h cycle using the chronodisruption model under ALF and TRF. However, transcriptomic changes under TRF in the chronodisruption model are beyond the scope of this manuscript. Since our study is primarily focusing on obesity, we have reworded the manuscript text to discuss mainly obesity as a metabolic challenge.

Figure S1 shows that more rhythmic genes were found during TRF than ALF in the WT and DIO groups, which is what was to be expected based on previous studies. However, in the GIO this pattern was reversed, i.e. more rhythmic genes in the ALF than in the TRF group. Misleading in this figure is that circle size is not representing the number of genes, in the WT and DIO groups the difference between ALF and TRF is approx. 100 genes, but in the GIO group this difference is >2000 genes!

Response: In agreement with the reviewer, we have edited the figure to reflect the differences in order to prevent confusion with our venn diagram (Fig S1b).

Fig. S1b) Periodic skeletal muscle transcripts identified under ALF and TRF in WT, HFD and *Sk2* flies.

Typo's:

Third line in the 3rd paragraph of the Introduction, “Recent” does not need a capital to start with.

Response: This has now been corrected to lower case

First line in the final paragraph on page 5, “the accumulation” can be removed.

Response: This has now been removed.

Second line on page 6, what is meant with the “cytotoxic effect in adipose tissue”?

Response: This was a typo, which has now been removed to include ectopic lipid deposition and myofibrillar disorganization instead.

Final lines of the 2nd paragraph on page 6, what is meant with “on aging”? Probably it should read “with aging”?

Response: This was a typo, “on aging” has also been removed and replaced.

REVIEWER COMMENTS

Reviewer #1 (Remarks to the Author):

In the revised manuscript, the authors have addressed almost all the concerns that were raised from this reviewer and overall, their findings have been solidly confirmed. I can now recommend this study to publish.

Only one minor comment: The authors have assessed the effect of folic acid feeding on flight performance and ATP levels. Although this experiment is not mandatory, one further investigation in how folate feeding affects other metabolites would strengthen their study.

Reviewer #2 (Remarks to the Author):

I am happy to see that this revised manuscript by Livelo and colleagues has now an understandable title and abstract.

The authors suggest that TRF in an aging fly obesity model results in some muscle function improvement by upregulation of genes producing glycine and SAM as well as downregulation of Dgat2, regulating triglyceride synthesis.

As explained in my initial review, I am not too surprised that drastic knock-down of glycine production enzymes Gnmt, Sardh and CG5955 result in a muscle phenotype, this might be interesting as explained by the authors, but the link to TRF rescue of obesity is unclear to me. This would require mild over-expression (1.25-fold?) and assay the effect under fatty diet. The authors did this only for one gene (Gnmt) and the effects seem small. Still the authors conclude that "Together, our results suggest that Gnmt, Sardh, and CG5955 are required for TRF-mediated improvement of skeletal muscle performance (line 269). This conclusion is not justified by the data presented.

Hence the effects seen upon knock-down of Dgat2, in the revised version attempted to be done specifically in adults, could be more interesting related to the mechanism of TRF rescued muscle deterioration.

The wealth of information presented in the manuscript comes with the cost that no gene or mechanism is investigated in detail. I still see many over interpretations of the data and it seems to me that inconclusive results are either ignored or mis-interpreted to fit the authors hypothesis. This is not the level of rigorosity I do expect for a Nat Comm article.

1. I appreciate that the authors have now attempted to verify the transcriptional changes by qPCR now shown in fig S2. In the text they state "validate these gene expression changes, which were found to be consistent". However, I do not see p-values in Figure S2c and it unclear to me why 4/4, 3/4 and 2/4 are shown? Does this mean in case of 2/4 only 2 out of 4 tests were plotted? If this is the case, I question the usefulness of data. Focusing on the important genes that are followed up here and plotting them properly with statistical analysis would be more useful and accurate. A general over-interpretation of the verification result in the text does not help.

2. I am also concerned that "outliers" in the transcriptomics analysis have been removed. What is the justification for this? Did the authors perform a PCA analysis that justified this measure?

3. It is interesting to see that the authors have now measured glycine levels in the thoraces of aged adults raised in the different conditions. However, they only find a small effect of TRF in the Sk2^{-/-} model, not in the wild type and not in high fat diet. Additionally, the Sk2^{-/-} which supposedly mimics the HFD, has a totally different glycine level than the HFD flies. This questions to me many of the findings the authors present here or at least their interpretation that the protective mechanism of TRF is mediated through the modification of glycine levels by the slightly changed expression levels of the enzymes studied.

4. I appreciate that the authors have tested 3 independent RNAi lines in the Act88F induced RNAi experiments. However, the important Figure 2d-e presents only of one? And it is not specified which one. Obviously, testing 3 different ones means the results of each of these needs to be shown at least in one of the functional assays. Only these functional assays can rule out off-target effects (and not the assay testing knock-down efficiency of the on-target).

5. Line 250 "To our knowledge, we report for the first time the muscle-specific requirement of Gnm1, Sardh, and CG5955." A developmental flight muscle function for CG5955 had been reported in PMID 20220848. Developmental knock-down resulted in flightlessness.

6. Flight performance differences for Gnm1, Sardh, and CG5955 RNAi reported here with the supposedly adult specific DJ694 driver appear very minor to me, despite significance (Fig 3A). Is this biologically relevant?

7. line 282: "The expression levels of Dgat2 were reduced under TRF versus ALF at ≥ 4 time points from our transcriptomic data in WT and obesity models (Fig. 4b and Supplementary Fig. 2b)." I do not see this reduction for wild type in Fig 4b.

Were the assays done in Fig4d-j with flies raised on normal food or fatty food? If the first is the case and give the above not difference in wild type I question the usefulness of the finding.

8. In the response to the reviewers' comments the authors write that they now used gene-switch Act88F-GAL4 to induce knock-down in the adult only. It seems for the most interesting experiment, the adult specific knock-down of Dgat2, only the less characterized and likely not muscle specific DJ694-GAL4 line was used. Figures 5 and 6 were only done with developmental knock-down, so no need to comment from me in detail. The obvious developmental effects cannot be ruled out, see next point.

9. Nmdmc and AdSL were already described to be essential for normal muscle function (PMID 20220848). This means both genes are essential for muscle function in general. Hence, it is trivial that a possible improvement induced by TRF does not happen upon knock-down of Nmdmc and AdSL. The same is also true for Ampk α (SNF1A), mAcon1, Ogdh (Nc73EF) and SdhD. These genes were all shown to be developmentally required for muscle function (PMID 20220848).

Reviewer #3 (Remarks to the Author):

The authors present a carefully and comprehensively revised version of this interesting study, which satisfies almost all of my initial concerns.

In view of the remarkable extra experimental evidence added by the authors, it remains unclear, why they refrain from overexpressing human Dgat2 under TRF (and not under ALF) conditions to demonstrate that reverting the (endogenous) Dgat2 down-regulation erases the beneficial effects of TRF on flight performance.

The use of an additional, adult-specific driver for the indirect fly muscles is a real gain and strengthens the conclusion of the authors. Still, it is unclear to me, why the authors determined the knockdown efficiency under ALF and not under TRF to directly address the extent of the compensatory regulation.

"Abdomen lipid staining was also performed in fat body to ensure the Nile red quantification of lipids represent the true physiology (Fig. S5a-c)."

The rationale of this argument is unclear to me as Act88F is introduced as IFM-specific driver. Showing that manipulation of any of the genes of interest in the flight muscles does not cause changes in the abdominal fat body LD population argues in favor of the absence of non-cell autonomous effects. But it fails to prove the analytical value of the method as suggested by this

statement. Any control causing a change in LD size/density is missing here. This needs to be corrected.

All in all I congratulate the authors to an insightful study, which broadens our understanding of the mechanisms underlying the beneficial effects of TRF.

Reviewer #4 (Remarks to the Author):

I thank the authors for their answers and edits in response to my remarks. There's only one remaining issue. In response to my remarks about Suppl Figure S1b the authors adapted the figure itself, but they did not respond to the question that was also in the remark, i.e. why is in the SK2 experiment the number of rhythmic genes (much) higher in the ALF group than in the TRF group? Which is contrary to what is expected and what is found in the WT and HF experiments.

Reviewer #1 (Remarks to the Author):

In the revised manuscript, the authors have addressed almost all the concerns that were raised from this reviewer and overall, their findings have been solidly confirmed. I can now recommend this study to publish. Only one minor comment: The authors have assessed the effect of folic acid feeding on flight performance and ATP levels. Although this experiment is not mandatory, one further investigation in how folate feeding affects other metabolites would strengthen their study.

Response: Thank you for the encouraging words and for the insightful comment. We have not directly measured the effects of folic acid feeding on metabolites; however, we have shown increases in metabolites related to the purine cycle under HFD-TRF. Metabolites such as oxypurines (hypoxanthine, xanthine) were decreased in HFD-TRF compared to ALF. Another study also noted that decreased oxypurines were observed in folic acid-treated ischemic patients (PMID: 18362233). Furthermore, an *in-silico* study predicted that folate depletion leads to a reduction of ATP pools (PMID: 32384607), while we observed increased ATP under HFD-TRF in our study. Taken together, although we have not directly assessed the effects of folic feeding on metabolites, we have some support indicating that HFD-TRF metabolites may likely reflect a similar metabolite output as folic feeding.

Reviewer #2 (Remarks to the Author):

I am happy that this revised manuscript by Livelo and colleagues now has an understandable title and abstract. The authors suggest that TRF in an aging fly obesity model results in some muscle function improvement by upregulation of genes producing glycine and SAM as well as downregulation of *Dgat2*, regulating triglyceride synthesis.

As explained in my initial review, I am not too surprised that drastic knock-down of glycine production enzymes *Gnmt*, *Sardh* and *CG5955* result in a muscle phenotype, this might be interesting as explained by the authors, but the link to TRF rescue of obesity is unclear to me. This would require mild overexpression (1.25-fold?) and assay the effect under fatty diet. The authors did this only for one gene (*Gnmt*) and the effects seem small. Still the authors conclude that “Together, our results suggest that *Gnmt*, *Sardh*, and *CG5955* are required for TRF-mediated improvement of skeletal muscle performance (line 269). This conclusion is not justified by the data presented.

Response: Thank you for this comment. As suggested by the reviewer, we understand the concern that overexpression of *Gnmt* driven by *Act88F* driver may not fully recapitulate the physiological effects of TRF as the overexpression may be much higher than is actually observed under TRF conditions. To address this, we now have performed both *Gnmt* overexpression using the *DJ694-Gal4* and *Act88F-GS-Gal4* drivers with titrated doses of RU486 (no RU, 10nM RU, 50 nM RU and 100 nM RU) in both regular diets (RD) and under fatty diet (HFD). Relative expression levels of *Gnmt* are shown in Fig. 3f and Supplementary Fig. 3i. The levels of overexpression from either driver displayed expression levels of *Gnmt* similar to expression levels seen in TRF conditions while still displaying improvement in muscle performance in both RD and HFD (Fig. 3g and Supplementary Fig. 3m, n). It is to note that we observed a mild reduction in flight performance from RU supplementation alone, therefore, we feel the *DJ694* (Also expressed in the IFM, response for comment #9) driver would be a more amenable choice for most of the validation experiments. Altogether, our results support the idea that *Gnmt* overexpression benefits

muscle performance/muscle maintenance when the expression is modulated to the levels seen in TRF. Transgenic overexpression stocks are not available for *Sardh* and *CG5955*, however, we found KD of all three genes *Gnmt*, *Sardh*, and *CG5955* resulted in impairment of muscle function using the *Act88F* and *DJ694* drivers. Further, loss of TRF beneficial effect on muscle performance was observed upon *Gnmt*, *Sardh*, and *CG5955* KD driven by *DJ694* (with 50-60% knockdown efficiency), suggesting their importance in TRF-mediated improvement of muscle function. As only *Gnmt* overexpression was tested, we have modified our statement to “Taken together, TRF-mediated upregulation of *Gnmt*, *Sardh*, and *CG5955* may account for at least a part of the beneficial effect of TRF in skeletal muscle.”

Fig. 3

Fig. S3

Fig. 3f, g) Relative RNA expression levels (f) and muscle performance (g) of 3-week-old female flies with *Gnmt* overexpression driven by *DJ694* under RD or HFD. Fig. S3l-n) Relative RNA expression levels (l) and muscle performance (m, n) of 3-week-old female flies with *Gnmt* overexpression driven by *Act88F-GS*. RD or HFD are supplemented with no RU, 10nM RU, 50nM RU or 100nM RU. RD: Regular diet; HFD: High-fat diet.

Hence the effects seen upon knock-down of *Dgat2*, in the revised version attempted to be done specifically in adults, could be more interesting related to the mechanism of TRF rescued muscle deterioration. The wealth of information presented in the manuscript comes with the cost that no gene or mechanism is investigated in detail. I still see many over interpretations of the data and it seems to

me that inconclusive results are either ignored or mis-interpreted to fit the authors hypothesis. This is not the level of rigorousness I do expect for a Nat Comm article.

Response: Both *Act88F-GS* (PMID: 31123597; PMID: 32537848) and *DJ694* (PMID: 12882353; PMID: 22008792; PMID: 30239736) are established IFM-specific drivers which can induce expression in the adult phase. For the majority of the experiments, *DJ694* was used as a higher dose of RU486 can have adverse effects on muscle function (PMID: 31123597). Meanwhile, no requirement of the drug was needed for *DJ694*-driving expression.

As we also addressed for reviewer 3, we conducted TRF upon *Dgat2* KD in both RD and HFD. Interestingly, the 5-week-old flight performance was further improved under WT-TRF and HFD-TRF upon *Dgat2* KD driven by *DJ694* (Supplementary Fig. 4g), while additional reduction of *Dgat2* expression levels was observed under TRF in the *Dgat2* KD flies (Supplementary Fig. 4f). In addition, we overexpressed h*Dgat2* using *DJ694* and improvements on muscle function were demonstrated (Supplementary Fig. 4i). It is noted that endogenous *Dgat2* was reduced under TRF in h*Dgat2*-OE flies (Supplementary Fig. 4h). Altogether, our results suggested the possibility that TRF-mediated further reduction of *Dgat2* may account for the observed muscle improvements from *Dgat2* KD and *Dgat2* OE flies under TRF. However, we are not able to preclude any other pleiotropic effects as a result of TRF which may be playing a contributing role in the observed muscle improvement. While we couldn't rule out if these additional improvements were from further lower levels of *Dgat2* or other TRF-mediated changes, our results supported that TRF-mediated downregulation of *Dgat2* is beneficial for muscle function.

Fig. S4

Fig. S4f) Relative expression levels of *Dgat2* under ALF and TRF upon *DJ694* driving *Dgat2* KD from 3-week-old female flies. Fig. S4g) flight performance of 5-week-old female flies under ALF and TRF upon *Dgat2* KD in both RD and HFD. Fig. S4h) Relative expression levels of *Dgat2* under ALF and TRF upon h*Dgat2* OE from 3-week-old female flies. Fig. S4i) Flight performance of 3-week-old female flies upon *Dgat2* O under ALF and TRF in RD or HFD. A: ALF; T: TRF.

See below as well as answers/justification to the additional specific questions/comments made by this reviewer.

1. I appreciate that the authors have now attempted to verify the transcriptional changes by qPCR now shown in fig S2. In the text they state “validate these gene expression changes, which were found to be

consistent". However, I do not see p-values in Figure S2c and it unclear to me why 4/4, 3/4 and 2/4 are shown? Does this mean in case of 2/4 only 2 out of 4 tests were plotted? If this is the case, I question the usefulness of data. Focusing on the important genes that are followed up here and plotting them properly with statistical analysis would be more useful and accurate. A general over-interpretation of the verification result in the text does not help.

Response: We appreciate the suggestion for the usage of p-values and now have incorporated this into Fig. S2c. The numbers "4/4" for example indicate the number of time points where gene expression was increased or decreased under TRF compared to ALF. This was shown as a way to display upregulated or downregulated gene time-series expression patterns in sequencing and qPCR data under ALF and TRF. In order to avoid potential confusion, we have now removed the numbers in the revised manuscript.

Fig. S2c

Fig. S2c) qRT-PCR validation of temporal expression of genes from Fig S2b.

2. I am also concerned that "outliers" in the transcriptomics analysis have been removed. What is the justification for this? Did the authors perform a PCA analysis that justified this measure?

Response: Yes, PCA analysis was included in the previously revised version (Fig. S1a). Outliers are indicated with a dashed red box, the figure has been added below as a reference.

Fig. S1a. PCA plot shown in last revision with red dashed box indicating outliers.

3. It is interesting to see that the authors have now measured glycine levels in the thoraces of aged adults raised in the different conditions. However, they only find a small effect of TRF in the *Sk2*^{-/-} model, not in the wild type and not in high fat diet. Additionally, the *Sk2*^{-/-} which supposedly mimics the HFD, has a totally different glycine level than the HFD flies. This questions to me many of the findings the authors present here or at least their interpretation that the protective mechanism of TRF is mediated through the modification of glycine levels by the slightly changed expression levels of the enzymes studied.

Response: Though we did not measure a significant increase in glycine levels under HFD-TRF despite the upregulation of glycine-producing genes *Sardh* and *CG5955*, we hypothesized that this could be due to the TRF-mediated activation of the purine cycle. Activation of the purine cycle leads to the consumption of glycine in a critical step mediated by *phosphoribosylglycinamide transformylase (Gart)* potentially countering the increase of glycine production. In WT and *Sk2* flies, we did not observe pronounced activation of the genes associated with the purine cycle (Fig. 5 and Supplementary Fig. 7a)

To test our hypothesis about purine cycle upregulation countering the effects of glycine production in HFD-TRF, we performed *Gart* KD and subjected those flies to HFD-TRF. Without the *Gart* KD, we found similar results where glycine was not increased under TRF in HFD. Upon *Gart* KD, glycine was found to be upregulated in HFD-TRF compared to HFD-ALF (Supplementary Fig. 7c). Moreover, *Nmdmc*, a necessary gene shown to be critical for purine cycle activation (PMID: 26912861) was found to be significantly increased under TRF in HFD but not *Sk2* (Supplementary Fig. 7a). This supports the idea that purine cycles are activated under TRF mainly in HFD, but not *Sk2*. Regarding *Sk2*, *Gart* levels are relatively high, however, no significant differences are observed between *Sk2*-ALF and TRF (Supplementary Fig. 7a). In conjunction with having lower levels of *Nmdmc* expression, a critical gene for encoding a key enzyme in purine activation, this suggests that increases in glycine levels found in *Sk2* did not undergo the same masking effect as shown in HFD-TRF as a result of purine cycle activation. It is to note, however, in an attempt to show that *Gart* had no effects on *Sk2* TRF vs. ALF, we conducted simultaneous KD of *Gart2* and *Sk2* in the IFM using *Act88F* and found that TRF in *Sk2/Gart* KD no longer had increased glycine shown in Supplementary Fig. 7c. This is potentially because only ~ 20% KD of *Sk2* was achieved with the *Act88F* driver when combined with *Gart* KD, which might be not sufficient to alter glycine levels. Due to this limitation, we are not able to certainly conclude the independence of *Sk2* TRF from *Gart*.

Regarding the differences in glycine levels between HFD and *Sk2*, it will be difficult to assess the differences in glycine levels as different pathways of obesity are involved. For example, having higher glycine levels in *Sk2* may be a compensatory effect to help regulate genetic-induced metabolic dysfunction which may or may not occur in HFD conditions. From our data, we were at least able to observe that TRF leads to an increase in glycine level in both HFD with *Gart* KD and *Sk2* mutants. Further, as glycine levels were originally found to be similar in ALF/TRF in HFD, this suggests that increased glycine resulting from TRF is used towards the activation of the purine cycle as upon introducing KD of *Gart*, glycine levels were then found to be significantly increased in HFD-TRF.

Fig. S7a

Fig. S7c

Fig. S7a) Log expression of genes involved in the purine cycle found to be increased in TRF under HFD conditions. Shown here, *Gart* is significantly increased under TRF in HFD compared to ALF and *Nmdmc* is relatively low in both ALF/TRF in *Sk2*. Fig. S7c) Relative glycine levels upon *Gart2* KD under HFD-TRF. Figure R2. Relative *Sardh* and *CG5955* expression levels in CS, *Sk2*, *Act88F*>Ctrl RNAi and *Act88F*>*Sk2* RNAi/Ctrl RNAi.

4. I appreciate that the authors have tested 3 independent RNAi lines in the Act88F induced RNAi experiments. However, the important Figure 2d-e presents only of one? And it is not specified which one. Obviously, testing 3 different ones means the results of each of these needs to be shown at least in one of the functional assays. Only these functional assays can rule out off-target effects (and not the assay testing knock-down efficiency of the on-target).

Response: We tested 3 independent RNAi lines on flight performance (Fig. 2d) and results from independent RNAi lines were indicated with the symbol circle, triangle, or square, which has been described in the figure legend. We understand that combining data points from 3 independent lines might mask the shape differences, therefore, we have added supplementary Fig. 3c to show results from the 3 independent lines separately. The RNAi lines used in Fig. 2d-e have been specified in the source data file and now they are also specified in the method section.

Fig. S3c) Flight index of *Gnmt*, *Sardh* and *CG5955* with each individual line plotted out under 1, 3, and 5 weeks of age. *Gnmt* RNAi #1: VDRC25983; #2: BDSC42637; #3: VDRC110623; *Sardh* RNAi #1: VDRC108873; #2: VDRC27601; #3: BDSC51883; *CG5955* RNAi #1: BDSC64566; #2: VDRC15838; #3: VDRC102243.

5. Line 250 “To our knowledge, we report for the first time the muscle-specific requirement of *Gnmt*, *Sardh*, and *CG5955*.” A developmental flight muscle function for *CG5955* had been reported in PMID 20220848. Developmental knock-down resulted in flightlessness.

Response: Thank you for pointing this out. We have reworded this statement to indicate that *Gnmt*, *Sardh* and *CG5955* are required for muscle maintenance using *DJ694* driver (Fig. S3f) in addition to validating their roles during muscle development via *Act88F* driver (Fig. 2d). The statement has been modified as “... suggesting the important roles of *Gnmt*, *Sardh*, and *CG5955* in muscle function and maintenance”. It is to note that a *Mef2-Gal4* driver was used in PMID 20220848 which is not an IFM-specific driver but a driver for all muscle cell lineages including the heart, body wall, and other muscles.

6. Flight performance differences for *Gnmt*, *Sardh*, and *CG5955* RNAi reported here with the supposedly adult specific *DJ694* driver appear very minor to me, despite significance (Fig 3A). Is this biologically relevant?

Response: Overall, it was observed that ~11% reduction in muscle performance resulted from adult IFM-specific KD in *Gnmt*, *Sardh* and *CG5955* in 3-week-old female flies. In measuring flight performance, this magnitude of reduction in muscle performance is clearly observed and mimics muscle performance seen in older adult flies. In humans, considering that muscle strength declines around 1.5% between age 50-60 and by 3% thereafter (PMID: 23160774) 11% reduction in muscle performance seems to fall within biological relevance.

7. line 282: “The expression levels of *Dgat2* were reduced under TRF versus ALF at ≥ 4 time points from our transcriptomic data in WT and obesity models (Fig. 4b and Supplementary Fig. 2b).” I do not see this reduction for wild type in Fig 4b.

Were the assays done in Fig4d-j with flies raised on normal food of fatty food? If the first is the case and give the above not difference in wild type I question the usefulness of the finding.

Response: We have now replotted expression levels without the 2 outliers and their corresponding ALF/TRF counterparts, which represents more appropriate comparisons of expression levels (Fig. 4b and Supplementary Fig. 2b). When comparing *Dgat2* expression under ALF and TRF, the comparisons were made at each specific time point. We understand that *Dgat2* expression levels at certain time points are close between ALF and TRF, therefore, we have now modified the statement that “The expression levels of *Dgat2* were downregulated under TRF versus ALF from our transcriptomic data in WT and obesity models (Fig. 4b and Supplementary Fig. 2b).” As DESeq analysis showed *Dgat2* expression levels were reduced under TRF in all WT, HFD and *Sk2* flies (Fig. 4b), the majority of assays were done under regular food. We agree that performing the assays under HFD would provide valuable information. We now have examined the flight performance of 5-week-old female flies under TRF upon *Dgat2* KD in both RD and HFD (Supplementary Fig 4g). Consistent with the observation in RD, *DJ694-driven* KD of *Dgat2* improved flight performance in HFD under TRF.

Fig. 4b) Expression level (normalized read count, log₂) of *Dgat2* under ALF and TRF in WT, HFD, and *Sk2* flies. Figure S4g. Flight performance of 5-week-old female flies upon *DJ694* driven *Dgat2* KD under ALF and TRF in RD or HFD.

8. In the response to the reviewers' comments the authors write that they now used gene-switch Act88F-GAL4 to induce knock-down in the adult only. It seems for the most interesting experiment, the adult-specific knock-down of Dgat2, only the less characterized and likely not muscle-specific DJ694-GAL4 line was used. Figures 5 and 6 were only done with developmental knock-down, so no need to comment from me in detail. The obvious developmental effects cannot be ruled out, see next point.

Response: As indicated above, the *Act88F-GS* is a viable driver for IFM-specific gene modulation in the adult phase (PMID: 31123597; PMID: 32537848), and *DJ694* has also been established as a robust driver for gene modulation in adults in IFM (PMID: 12882353; PMID: 22008792; PMID: 30239736.) without the need for induction of drugs (RU486). Moreover, pursuing RU486-mediated induction along with TRF implementation became challenging, therefore we used the *DJ694* driver for the follow-up studies.

We appreciate reviewer feedback on testing the KD of *Nmdmc*, *AdSL*, *Ampka*, *mAcon1*, *Ogdh*, and *SdhD* during the adult phase. Unlike the *Act88F* driver, muscle function wasn't severely affected upon KD of those genes using *DJ694* driver (except *Ogdh*). However, TRF failed to improve muscle performance (except in *mAcon1* KD flies), suggesting the potential involvement of *Nmdmc*, *AdSL*, *Ampka*, *Ogdh*, and *SdhD* for TRF beneficial effects on muscle. Future studies will further test the roles of these genes under TRF in diet and genetic obesity conditions on muscle performance.

Fig. S7f

Fig.S8h

Fig. S7e & S8h) Flight index of adult IFM-specific KD from indicated genes at 5-week-old female flies. A: ALF; T: TRF.

9. *Nmdmc* and *AdSL* were already described to be essential for normal muscle function (PMID 20220848). This means both genes are essential for muscle function in general. Hence, it is trivial that a possible improvement induced by TRF does not happen upon knock-down of *Nmdmc* and *AdSL*. The same is also true for *Ampka* (SNF1A), *mAcon1*, *Ogdh* (Nc73EF) and *SdhD*. These gene were all shown to be developmentally required for muscle function (PMID 20220848).

Response: The literary source cited a study that used the *Mef2-Gal4* driver for inducing RNAi knockdown of genes, of which *Nmdmc*, *AdSL* and *Ampka* related genes were found to be essential for muscle function (also cited in the revised manuscript). It is to note that *Mef-2* is a driver for multiple muscle lineages and is not IFM-specific. When we performed KD using the *Act88F* driver, our results aligned with the idea that these genes were developmentally required for muscle function. As the focus of this manuscript is on the mechanism of TRF's effect on IFM, we performed IFM-specific knockdown using

DJ694. Muscle function wasn't severely affected by the KD of those genes using the *DJ694* driver (except *Ogdh*). However, TRF failed to improve muscle performance (except in *mAcon1* KD flies), suggesting the potential involvement of *Nmdmc*, *AdSL*, *Ampka*, *Ogdh*, and *SdhD* for TRF beneficial effects on muscle.

Reviewer #3 (Remarks to the Author):

The authors present a carefully and comprehensively revised version of this interesting study, which satisfies almost all of my initial concerns.

In view of the remarkable extra experimental evidence added by the authors, it remains unclear, why they refrain from overexpressing human *Dgat2* under TRF (and not under ALF) conditions to demonstrate that reverting the (endogenous) *Dgat2* down-regulation erases the beneficial effects of TRF on flight performance.

Response: Thank you for the comment. We now have performed TRF on flies with h*Dgat2* overexpression driven by *DJ694* in both RD and HFD. Interestingly, improved flight performance was still observed under TRF compared to age-matched ALF (Supplementary Fig. 4i). It is not completely surprising as *Dgat2* is not the only gene modulated under TRF, and reduction of endogenous *Dgat2* from TRF may counter the effects of h*Dgat2* overexpression. The possibility of h*Dgat2* overexpression not disturbing other TRF-modulated changes, and those changes leading to improvement of muscle performance during aging or obesogenic challenges, however, cannot be ruled out. Although it will be an interesting topic to decipher the extent of *Dgat2* modulation or other modulations on TRF beneficial effects, we feel that it is not within the scope of this manuscript.

Fig. S4h) Relative expression levels of *Dgat2* under ALF and TRF upon h*Dgat2* OE from 3-week-old female flies. Fig. S4i) Flight performance of 3-week-old female flies upon *Dgat2* O under ALF and TRF in RD or HFD. A: ALF; T: TRF.

The use of an additional, adult-specific driver for the indirect fly muscles is a real gain and strengthens the conclusion of the authors. Still, it is unclear to me, why the authors determined the knockdown efficiency under ALF and not under TRF to directly address the extent of the compensatory regulation.

Response: To address the extent of the compensatory regulation, we performed knockdown using adult-specific muscle driver *DJ694*. Our qPCR validation has shown ~40-60% knockdown level for *Gnmt*, *Sardh*, and *CG5955*. While mild increases in expression levels of *Gnmt*, *Sardh*, and *CG5955* were observed under TRF in their corresponding KD flies (Fig. S3g), no muscle improvements were observed.

Fig. S3g) Relative mRNA expressions of *Gnmt*, *Sardh*, and *CG5955* upon corresponding *DJ694* driven KD in 3-week-old female ALF and TRF flies were quantified with qRT-PCR. A: ALF; T: TRF.

“Abdomen lipid staining was also performed in fat body to ensure the Nile red quantification of lipids represent the true physiology (Fig. S5a-c).

The rationale of this argument is unclear to me as Act88F is introduced as an IFM-specific driver. Showing that manipulation of any of the genes of interest in the flight muscles does not cause changes in the abdominal fat body LD population argues in favor of the absence of non-cell autonomous effects. But it fail to prove the analytical value of the method as suggested by this statement. Any control causing a change in LD size/density is missing here. This needs to be corrected. All in all, I congratulate the authors to an insightful study, which broadens our understanding of the mechanisms underlying the beneficial effects of TRF.

Response: Thank you for the encouragement and the comment. The lipid staining and imaging method have been applied in our previous publication (PMID: 31221967) where we demonstrated increased numbers of lipid droplets in IFMs of HFD and *Sk2* flies compared to WT flies. We have imaged HFD as a positive control along with the experimental flies (Genotype: Act88F>Ctrl RNAi). Increased numbers of bigger lipid droplets were shown in Fig. R3. We have now modified the statement to “Abdomen lipid staining was also performed in the fat body and no significant differences were observed (Supplementary Fig. 5a-c).”

Fig. R1

Fig. R1) Fluorescence images of the IFMs from 3-week-old female flies with Ctrl RNAi driven by *Act88F-Gal4* under RD and HFD. Phalloidin (green) and Nile Red (red puncta). Scale bar is 20 μ m.

Reviewer #4 (Remarks to the Author):

I thank the authors for their answers and edits in response to my remarks. There's only one remaining issue. In response to my remarks about Suppl Figure S1b the authors adapted the figure itself, but they did not respond to the question that was also in the remark, i.e. why is in the SK2 experiment the number of rhythmic genes (much) higher in the ALF group than in the TRF group? Which is contrary to what is expected and what is found in the WT and HF experiments.

Response: Thank you for the comment. We are aware that TRF is known to increase the number of rhythmic genes in most of the independent TRF studies on varied tissues under aging or high-fat diet conditions. One possible explanation for fewer rhythmic genes in the TRF group compared to the ALF group is that the *Sk2* mutant fly is a genetic-induced obesity model, which might have more rhythmic genes. Moreover, TRF induces beneficial effects that are pleiotropic, and not limited to maintaining expression rhythmicity. One recent study has shown similar observations with fewer rhythmic genes under TRF compared to ALF upon knock-out of *Clk* in a mice model (PMID: 30174302). At this stage, we did not explore why fewer rhythmic genes were found under TRF in *Sk2* mutants, we feel that a detailed delineation of the underlying mechanisms would be important but beyond the scope of the current study. This explanation has been added to the manuscript text under results as well.

REVIEWERS' COMMENTS

Reviewer #2 (Remarks to the Author):

I appreciate that the authors have further improved their manuscript according to some of my earlier comments.

I just want to come back to my point former point 9 that stated: "Nmdmc and AdSL were already described to be essential for normal muscle functon(PMID 20220848). This means both genes are essential for muscle function in general. Hence, it is trivial that a possible improvement induced by TRF does not happen upon knock-down of Nmdmc and AdSL. The same is also true for Ampk α (SNF1A), mAcon1, Ogdh (Nc73EF) and SdhD. These gene were all shown to be developmentally required for muscle functon(PMID 20220848)."

Response: The literary source cited a study that used the Mef2-Gal4 driver for inducing RNAi knockdown of genes, of which Nmdmc, AdSL and Ampk α related genes were found to be essential for muscle functon(also cited in the revised manuscript). It is to note that Mef-2 is a driver for multiple muscle lineages and is not IFM-specific. When we performed KD using the Act88F driver, our results aligned with the idea that these genes were developmentally required for muscle function. As the focus of this manuscript is on the mechanism of TRF's effect on IFM, we performed IFM-specific knockdown using DJ694. Muscle function wasn't severely affected by the KD of those genes using the DJ694 driver (except Ogdh). However, TRF failed to improve muscle performance (except in mAcon1 KD flies), suggesting the potential involvement of Nmdmc, AdSL, Ampk α , Ogdh, and SdhD for TRF beneficial effects on muscle.

I have looked at the initial reference of the here used DJ694 driver (PMID: 12882353). This paper shows that DJ694 is NOT IFM-specific (in contrast to Act88F), there is expression in many cells in the head and in the abdomen. Furthermore, this initial reference does NOT show that this driver is off in the developing IFMs.

Despite the statement above, this revised manuscript does NOT cite PMID 20220848.

Reviewer #3 (Remarks to the Author):

My remaining concerns have been satisfyingly addressed and I congratulate the authors to a carefully revised and interesting study.

Reviewer #4 (Remarks to the Author):

I have no further comments.

REVIEWERS' COMMENTS

Reviewer #2 (Remarks to the Author):

I appreciate that the authors have further improved their manuscript according to some of my earlier comments.

I just want to come back to my point former point 9 that stated: "Nmdmc and AdSL were already described to be essential for normal muscle function(PMID 20220848). This means both genes are essential for muscle function in general. Hence, it is trivial that a possible improvement induced by TRF does not happen upon knock-down of Nmdmc and AdSL. The same is also true for Ampk α (SNF1A), mAcon1, Ogdh (Nc73EF) and SdhD. These gene were all shown to be developmentally required for muscle functon(PMID 20220848)."

I have looked at the initial reference of the here used DJ694 driver (PMID: 12882353). This paper shows that DJ694 is NOT IFM-specific (in contrast to Act88F), there is expression in many cells in the head and in the abdomen. Furthermore, this initial reference does NOT show that this driver is off in the developing IFMs.

Despite the statement above, this revised manuscript does NOT cite PMID 20220848.

Response: We sincerely apologize for not including PMID 20220848 in the revised manuscript, and the manuscript has now been added to the reference. We agree with the reviewer that, in principle, the loss of TRF-mediated benefit upon knockdown of Nmdmc, AdSL, Ampk α (SNF1A), mAcon1, Ogdh (Nc73EF), and SdhD, could be potentially due to the knockdown of genes essential for muscle function but not necessarily due to the absence of upregulation induced by TRF. However, the results from the above-mentioned experiments, together with our transcriptomic and metabolomics data, complementarily support our main finding of this manuscript that TRF improves muscle function through modulation of purine cycle (Nmdmc, AdSL) in HFD, and Ampk signaling (Ampk α (SNF1A)), glycogen metabolism, glycolysis, TCA (mAcon1, Ogdh (Nc73EF)) and ETC (SdhD) in *Sk2*. Our transcriptomic data showed significant upregulation of genes involved in the purine cycle under HFD-TRF but not *Sk2*-TRF, while significant upregulation of genes involved in AMPK-associated pathways in *Sk2*-TRF but not HFD-TRF. Moreover, the metabolomic changes also aligned with the upregulation of distinct pathways in two obese models.

Upon further evaluation of the PMID: 12882353 article on the *DJ694* driver, *DJ694* is a muscle-specific driver that expresses thoracic flight muscle and abdominal muscle, as well as some cells in the head. PMID: 12882353 demonstrated that the expression level of the *DJ694* driver increases rapidly upon eclosion (Fig 3d in PMID: 12882353), and in the longitudinal flight muscle, the *DJ694* reporter increases monotonically across all ages (Fig 5a in PMID: 12882353). Although *DJ694* expresses during the larval and pupal phase, it is not expressed in the larval and pupal muscles but instead in the oenocytes and salivary glands (Figure 2 in PMID: 29259847). Another study also indicated that the *DJ694*-Gal4 driver becomes initiated in adults, especially in IFMs, upon eclosion (PMID: 22008792). They utilized *DJ694* to examine the role of MEF2 in the adult IFM (PMID: 22008792). Therefore, the goal of modestly modulating candidate genes in the adult stage would be greatly facilitated by using the *DJ694* driver because of the absence of expression in muscle before the adult stage.

In our study, we observed modest, or, no significant differences in flight performance upon *DJ694*-driven KD of *Gnmt*, *Sardh* and *CG5955* at day 4 (Figure 3a and Supplementary Figure 3f). In addition, *DJ694*-driven KD of *AdSL*, *Nmdmc*, *Ampka* and *SdhD* didn't lead to any significant decline in flight performance at the age of week 5 (Supplementary Figure 7f and 8h).

We acknowledge that *DJ694*-driven KD also occurs at oenocytes and salivary glands during the developmental phase and at abdominal muscle during the adult phase. We now have carefully stated our rationale for choosing the *DJ694* driver in our study and noted in the discussion that the potential effects on flight performance from *DJ694*-driven KD occurring outside of IFM are not assessed by our methods.

In the "results" section:

In order to examine the role of *Gnmt*, *Sardh* and *CG5955* in IFM during the adult phase, we utilized *DJ694-Gal4* driver, an adult-muscle driver that initiates upon eclosion and remains active during the whole adult life span.

In the "discussion" section:

Our results have shown that TRF-mediated benefits in IFM were abrogated upon KD of *AdSL*, *Nmdmc*, *Ampka*, *Ogdh*, and *SdhD* (**Supplementary Figure 7f, g and 8h**). It is to note that *AdSL*, *Nmdmc*, *Ampka*, *Ogdh*, and *SdhD* have been demonstrated to be important for muscle development using *Mef2-Gal4* driver. Therefore, the abrogation of TRF benefits could be potentially due to the knockdown of genes essential for muscle function and not necessarily due to the absence of upregulation induced by TRF. In an attempt to rule out this possibility, we used *DJ694-Gal4*, an adult-muscle driver, to induce modest suppression of target gene expression. Interestingly, *DJ694*-driven KD of *AdSL*, *Nmdmc*, *Ampka* and *SdhD* didn't lead to any significant decline in flight performance at the age of week 5 (**Supplementary Figure 7f and 8h**), suggesting the abrogation of TRF benefits was not caused by any muscle function decline from suppression of tested genes. Although the *DJ694* driver expresses in the oenocytes and salivary glands during the developmental phase and abdominal muscle during the adult phase, the *DJ694* driver allows modestly manipulating candidate genes in the adult IFM because of the absence of expression in muscle before the adult stage. However, potential effects on flight performance from *DJ694*-driven KD occurring outside of IFM cannot be assessed by our methods. Further assessment will also be needed to investigate the individual contribution of candidate genes in TRF-mediated benefits.

Fig 3
a

Fig S3
f

Fig. 3a and Fig. S3f) Flight index of 4-day-old and 3-week-old female (Fig. 3a) and male (Fig. S3f) flies upon *DJ694*-driven KD of *Gnm1*, *Sardh*, and *CG5955*.

f

h

Fig. S7f & S8h) Flight index of KD of indicated genes using *DJ694* driver in 5-week-old female flies. A: ALF; T: TRF.

Reviewer #3 (Remarks to the Author):

My remaining concerns have been satisfyingly addressed and I congratulate the authors to a carefully revised and interesting study.

Response: Thank you. We greatly appreciated positive feedback from reviewer 3.

Reviewer #4 (Remarks to the Author):

I have no further comments.

Response: Thank you very much for your evaluation and time.